# CERTIFYING THE FULL YOLO PIPELINE: A PROBABILISTIC VERIFICATION APPROACH

**Zongxin Liu**[1,2,†]**, Zhiming Chi**[1,2,†]**, Lijia Yu**[3]**, Tao Lin**[1,2]**, Lijun Zhang**[1,2,3,∗]

[1]Key Laboratory of System Software (Chinese Academy of Sciences), Institute of Software,
Chinese Academy of Sciences
[2]University of Chinese Academy of Sciences
[3]Institute of AI for Industries, Chinese Academy of Sciences
{liuzx, chizm}@ios.ac.cn, ljyu@iaii.ac.cn, {lintao,zhanglj}@ios.ac.cn

## ABSTRACT

Object detection systems are essential in safety-critical applications, but they are vulnerable to object disappearance (OD) threats, in which valid objects become undetected under small input perturbations, creating serious risks. This paper addresses the problem of verifying the robustness of YOLO (You Only Look Once) networks against OD by proposing a three-step probabilistic verification framework: (1) estimating output ranges under a distribution of input perturbations, (2) formally verifying the Non-Maximum Suppression (NMS) process within these ranges, and (3) iteratively refining the results to reduce over-approximation. The framework scales to practical YOLO models. Both theoretical analysis and experimental results demonstrate that our method achieves comparable probabilistic guarantees and provides tighter Intersection-over-Union (IoU) lower bounds while requiring significantly fewer samples than existing methods.

## 1 INTRODUCTION

Object detection (Zhao et al., 2019; Zou et al., 2023) is a fundamental computer vision task that combines object localization and classification. Neural network architectures, including YOLO (You Only Look Once) (Redmon, 2016; Redmon & Farhadi, 2017; Farhadi & Redmon, 2018; Bochkovskiy et al., 2020), Fast R-CNN (Girshick, 2015), and SSD (Liu et al., 2016; Li et al., 2017), have achieved significant progress in both accuracy and computational efficiency, enabling their widespread deployment in real-world applications. Despite these advances, neural network-based detection systems remain vulnerable to minute, often imperceptible, input perturbations (e.g., sensor noise) (Im Choi & Tian, 2022; Lin et al., 2025; Goodfellow et al., 2015; Madry et al., 2018; Dong et al., 2018; Carlini & Wagner, 2017). Of particular concern is the *object disappearance (OD) problem*, in which minor input perturbations suppress the detection of valid objects. Such perturbations pose substantial risks in safety-critical domains, potentially leading to catastrophic consequences due to detection failures. Consequently, verifying the safety of object detection systems is crucial for their reliable deployment.

To measure network robustness, verification methods are commonly employed. For a given network F, an input $\boldsymbol{x}$, and a property function $\phi$, verification methods can be grouped into three categories:

1. **Formal Verification.** The goal is to verify whether the property is preserved for all perturbed inputs satisfying a given input constraint set $\mathcal{C}$: $\forall \boldsymbol{x}' \in \mathcal{C}, \phi(\mathrm{F}(\boldsymbol{x}')) = \phi(\mathrm{F}(\boldsymbol{x}))$. Here $\mathcal{C}$ can encode norm-bounded perturbations, polyhedral constraints, or application-specific semantic constraints. Equivalently, one can search for a counterexample $\boldsymbol{x}' \in \mathcal{C}$ that violates the property. However, exact formal verification for ReLU networks is NP-complete (Katz et al., 2017), making it infeasible for large-scale networks.

2. **Probabilistic Verification.** Given a radius $\varepsilon$ and a tolerance $\alpha$, the goal is to verify whether $\mathbb{P}_{\boldsymbol{x}' \sim \mathcal{D}}(\phi(\mathrm{F}(\boldsymbol{x}')) = \phi(\mathrm{F}(\boldsymbol{x}))) \geq 1 - \alpha$, where $\mathcal{D}$ is a distribution over inputs in $\mathcal{C}$. Although this

---

∗Corresponding author.
†Equal contribution.

approach leverages probabilistic guarantees to reduce verification time and memory, its reliance on processing internal network nodes prevents it from scaling to larger network architectures. Representative works include (Weng et al., 2019; Boetius et al., 2025).

3. **Probably Approximately Correct (PAC) Verification.** Given $\varepsilon$, $\alpha$, and $\beta$, the goal is to verify whether $\mathbb{P}_{\boldsymbol{x}' \sim \mathcal{D}}(\phi(\mathrm{F}(\boldsymbol{x}')) = \phi(\mathrm{F}(\boldsymbol{x}))) \geq 1 - \alpha$ holds with confidence at least $1 - \beta$. PAC methods rely on sampling and do not require access to internal network nodes, which allows them to scale further to larger models and datasets. Representative works include (Tran et al., 2023; Park et al., 2020; Li et al., 2022; Blohm et al., 2025).

Verifying object detection networks with these methods, however, presents additional challenges beyond the large parameter scales:

1. **Post-Processing Stage**: Critical post-processing steps, such as Non-Maximum Suppression (NMS) (Neubeck & Van Gool, 2006), generally fall outside the scope of current formal verification methods (Cohen et al., 2024; Elboher et al., 2024);

2. **Large Input-Output Spaces**: The dimensionality of the detection inputs and outputs renders dimension-dependent PAC-based methods (Li et al., 2022; Blohm et al., 2025; Haussler & Welzl, 1987) computationally infeasible.

Due to these limitations, even recent verification methods specifically designed for object detection (Cohen et al., 2024; Elboher et al., 2024) are restricted to simplified models or do not account for complex operations such as NMS. To address this gap, we propose a PAC-based **O**bject **D**etection **P**robabilistic **V**erification (ODPV) framework for YOLO networks under OD threats. To our knowledge, **this is the first framework that effectively verifies the robustness of original object detection networks at a practical scale**. Although PAC-based methods cannot provide deterministic guarantees, they currently offer the most practical means to verify YOLO in a reasonable time.

Our methodology includes three main components: (1) estimating output ranges under input perturbations, (2) formally verifying NMS within the estimated output space, and (3) iteratively refining verification results. We implement our approach and evaluate it on standard benchmarks. Our main contributions are as follows:

- We formally define the PAC verification problem of the OD threat in object detection and propose a novel verification approach to address it.

- We implement a complete verification process that includes the NMS step, which has been underexplored in previous work, and provide probabilistic guarantees for each step.

- We conduct experiments on widely used networks and datasets to evaluate our proposed method. We demonstrate that our method requires fewer samples to achieve comparable probabilistic guarantees and tighter certified Intersection-over-Union (IoU) bounds.

In summary, we are the first to address the challenges of verifying large-scale detection networks and to provide an efficient probabilistic verification method.

**Remark 1.** *We emphasize an important distinction: Our work differs from randomized smoothing in the type of guarantee it provides (Cohen et al., 2019; Yang et al., 2020). Randomized smoothing establishes robustness for modified, "smoothed" classifiers, not the original detector. In contrast, we leave the network unchanged and provide statistical guarantees for the original model.*

## 2 RELATED WORK

**Object detection.** Early detectors relied on hand-crafted features such as Histogram of Oriented Gradients (HOG) (Dalal & Triggs, 2005) and sliding windows (Viola & Jones, 2001), but lacked adaptability. Convolutional Neural Network based approaches transformed feature extraction; R-CNN variants (Girshick et al., 2014; Ren, 2015) combined region proposals with deep learning methods. More recent approaches such as YOLO (Redmon, 2016; Redmon & Farhadi, 2017; Farhadi & Redmon, 2018; Bochkovskiy et al., 2020) and SSD (Liu et al., 2016; 2017) achieved real-time detection in complex scenarios.

**Verification techniques for Neural Networks.** Formal verification determines whether a property holds under given input constraints. State-of-the-art tools (Katz et al., 2017; 2019; Zhang et al., 2022; 2018) employ Branch-and-Bound, combining relaxations (Singh et al., 2019; Bak, 2021),

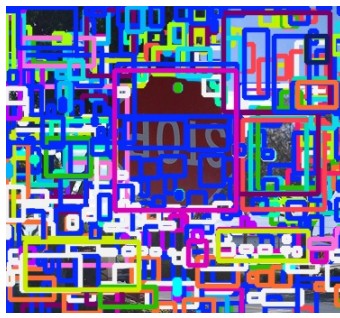 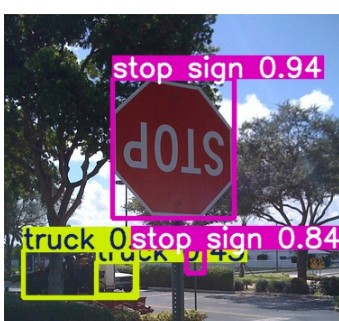 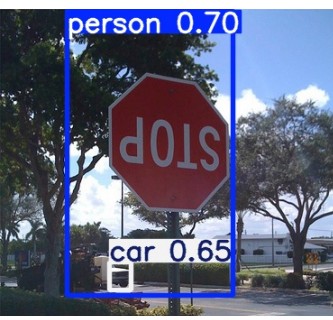

Figure 1: (First Stage) The network tries to find all boxes that may contain objects. A subset of these boxes is shown here.

Figure 2: (Second Stage) Final output boxes selected by NMS along with their corresponding labels and confidence scores.

Figure 3: Under imperceptible input perturbations, YOLO can no longer recognize these objects.

bound propagation (Wang et al., 2018b; Weng et al., 2018; Wang et al., 2018a; Gowal et al., 2019), and constraint solving (Khedr et al., 2021; Ehlers, 2017; Henriksen & Lomuscio, 2020; Kouvaros & Lomuscio, 2021). However, for large networks such as YOLO (with $640 \times 640 \times 3$ inputs), even basic bound propagation may require more than 5000 GB of memory, rendering formal verification infeasible in practice. To address scalability, probabilistic verification estimates the likelihood of property satisfaction. Sampling-based methods (Webb et al., 2019; Cardelli et al., 2019; Mangal et al., 2019; Anderson & Sojoudi, 2023) provide probabilistic estimates, but may miss rare cases, thereby creating gaps between analysis and actual robustness. DeepPAC (Li et al., 2022) approximates local network behavior with linear equations and high-confidence error bounds, but it requires prohibitively large sample sizes for models such as YOLO. Techniques like median smoothing (Chiang et al., 2020) certify robustness for a modified, "smoothed" detector, whereas our approach directly verifies the original network.

**Verification of Object Detection.** Current efforts mainly focus on small or simplified detectors. Cohen et al. (2024) propagates bounds to certify IoU, while Elboher et al. (2024) encodes IoU into networks for existing verifiers. Both approaches ignore the NMS step and fail to scale to real-world detectors. Comprehensive verification of complete detection pipelines remains an open problem.

## 3 PRELIMINARIES

This section outlines the key stages of YOLO object detection, as shown in Figs. 1–3 with an image from the COCO validation dataset (Lin et al., 2014) and defines the threat of OD.

### 3.1 KEY STAGES OF YOLO OBJECT DETECTION

**Bounding Box Prediction (First Stage).** The YOLO network $F : \mathbb{R}^{d_0} \to \mathbb{R}^{d_L}$ processes an input $\boldsymbol{x}$ (with dimension $d_0$) to generate an output $\boldsymbol{y} = F(\boldsymbol{x})$ (with dimension $d_L$). The output $\boldsymbol{y}$ can be reformulated as a set of bounding boxes $\{box_i\}_{i=1}^{n_{\boldsymbol{x}}}$, where $n_{\boldsymbol{x}}$ is a constant determined by the fixed input dimension. Each bounding box $box_i$ is represented as $(x_i, y_i, w_i, h_i, c_i, p_{i_1}, p_{i_2}, \ldots, p_{i_{n_{\text{cls}}}})$. Here, $(x_i, y_i)$ denotes the box's center coordinates, $(w_i, h_i)$ its width and height, $c_i$ its confidence score, and $p_{i_j}$ the probability of the object belonging to class $j$ (for $j \in [n_{\text{cls}}]$, where $n_{\text{cls}}$ is the total number of classes). The class of $box_i$ is assigned as $\text{Class}(box_i) = \arg\max_{j \in [n_{\text{cls}}]} p_{i_j}$. These boxes collectively identify possible object locations in the input image, as Figure 1 illustrates.

**Non-Maximum Suppression (Second Stage).** Let $\boldsymbol{y} = F(\boldsymbol{x})$ be the first-stage output tensor. The second stage applies an operator N to $\boldsymbol{y}$, which yields an index set $S(\boldsymbol{y}) \subseteq [n_{\boldsymbol{x}}]$. The selected boxes are $\{box_{i_j}\}_{i_j \in S(\boldsymbol{y})} \subseteq \{box_i\}_{i=1}^{n_{\boldsymbol{x}}}$, which form the final YOLO output (Figure 2). The standard operator N is NMS (Neubeck & Van Gool, 2006) in YOLO, which uses $\boldsymbol{y}$ and predefined thresholds $\eta, \iota \in (0, 1)$ to select the final output. For simplicity, we denote this as $N(\boldsymbol{y})$; since $\eta$ and $\iota$ are fixed, we omit them. NMS selects boxes based on the following three rules:

(n1) If $i_j \in [n_{\boldsymbol{x}}]$ and $box_{i_j} \in N(\boldsymbol{y})$, then it must satisfy $c_{i_j} \geq \iota$;

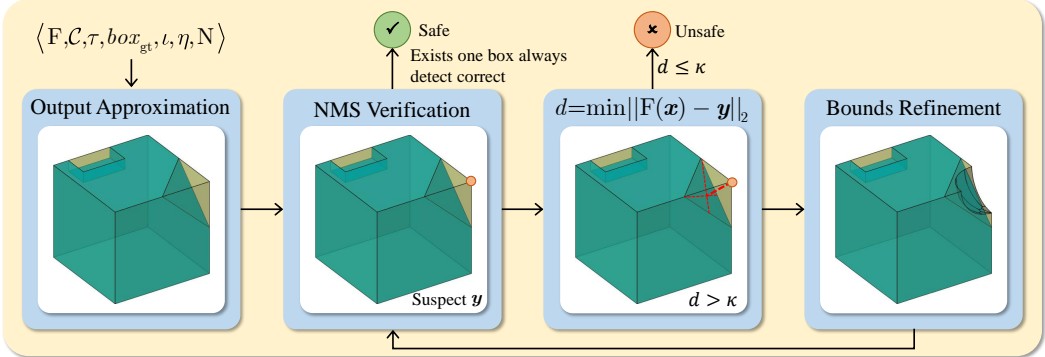

Figure 4: Overview of our verification framework for object detection. The green cube denotes the true but unknown output set under input constraints. The yellow cube is the probabilistically guaranteed over-approximation computed by our method. Refinement iteratively removes regions that do not intersect the true output set, progressively tightening the bound.

(n2) If $i_j \in [n_{\boldsymbol{x}}]$ satisfies $box_{i_j} \notin N(\boldsymbol{y})$ and $c_{i_j} \geq \iota$, then there must exist a $box_{i_k} \in N(\boldsymbol{y})$ such that $\text{Class}(box_{i_j}) = \text{Class}(box_{i_k})$ and $c_{i_j} \leq c_{i_k}$, $\text{IoU}(box_{i_j}, box_{i_k}) \geq \eta$;

(n3) If $i_j, i_k \in [n_{\boldsymbol{x}}]$ such that $box_{i_j}, box_{i_k} \in N(\boldsymbol{y})$ and $\text{Class}(box_{i_j}) = \text{Class}(box_{i_k})$, then it must satisfy $\text{IoU}(box_{i_j}, box_{i_k}) < \eta$.

The $\text{IoU}(box_1, box_2) = \frac{\text{Area}(box_1 \cap box_2)}{\text{Area}(box_1 \cup box_2)}$ measures overlap between two boxes, where $\text{Area}(box_1 \cap box_2)$ and $\text{Area}(box_1 \cup box_2)$ denote the areas of the intersection and union, respectively. We focus on the properties of the unique NMS-selected subset, omitting implementation details.

## 3.2 OBJECT DISAPPEARANCE THREAT IN OBJECT DETECTION

An object detection model successfully detects object $O$ in the image $\boldsymbol{x}$ if there exists at least one $box_i \in N(F(\boldsymbol{x}))$ satisfying: $\text{Class}(box_i) = \text{Class}(box_{\text{gt}})$ and $\text{IoU}(box_i, box_{\text{gt}}) \geq \tau$, where $\tau$ is a predefined IoU threshold and $box_{\text{gt}}$ is $O$'s ground truth box. We define the OD threat as follows:

**OD Threat Definition.** Given ground truth box $box_{\text{gt}}$, input constraints $\mathcal{C}$, IoU threshold $\tau$, and class $\text{Class}(box_{\text{gt}})$, OD occurs if there exists an input $\boldsymbol{x} \in \mathcal{C}$ such that:

$$\max_{box_i \in N(F(\boldsymbol{x}))} \left[ \text{IoU}(box_i, box_{\text{gt}}) \cdot \mathbb{I}\left(\text{Class}(box_i) = \text{Class}(box_{\text{gt}})\right) \right] < \tau,$$

where $\mathbb{I}(\cdot)$ denotes an indicator function (returns 1 if true, 0 otherwise). We assume $\max(\emptyset) = 0$.

## 4 VERIFICATION FRAMEWORK FOR OBJECT DETECTION

In this section, we introduce the verification target and our verification approach.

First, we formally define the OD PAC-Verification problem.

**Definition 1** (OD PAC-Verification Problem). Given input constraints $\mathcal{C}$, IoU threshold $\tau$, error rate $\alpha \in [0, 1]$, significance level $\beta \in [0, 1]$, and ground truth box $box_{\text{gt}}$, verify whether with confidence at least $1 - \beta$, the following holds:

$$\mathbb{P}_{\boldsymbol{x} \sim \mathcal{C}} \left( \exists box_i \in N(F(\boldsymbol{x})) \text{ s.t. } \left[ \text{IoU}(box_i, box_{\text{gt}}) \geq \tau \wedge \text{Class}(box_i) = \text{Class}(box_{\text{gt}}) \right] \right) \geq 1 - \alpha$$

If true, the system is deemed PAC-safe in $\mathcal{C}$ under $\tau$. Here we use $\boldsymbol{x} \sim \mathcal{C}$ to denote that $\boldsymbol{x}$ is sampled from a distribution over the inputs in the constraint set $\mathcal{C}$.

Then, we propose a three-part verification framework (see Alg. 1 and Fig. 4) to solve this problem:

**Part 1: Network Output Approximation.** For input $\boldsymbol{x}^{(0)}$ and constraint $\mathcal{C}$, approximate the output set $\{F(\boldsymbol{x})\}_{\boldsymbol{x} \in \mathcal{C}}$ of the network F with a regular region $\mathcal{Z}$ (initially a hyperrectangle; refined by removing hyperspherical exclusion zones) satisfying $\{F(\boldsymbol{x})\}_{\boldsymbol{x} \in \mathcal{C}} \subseteq \mathcal{Z}$.

---

**Algorithm 1** Verification framework for the OD PAC-Verification problem

---

**Require:**
   The network F; the input constraints $\mathcal{C}$; the threshold in OD verification problem $\tau$; the threshold for Part Three $\kappa$; the number of refinement steps $T$; the ground truth bounding box $box_{\text{gt}}$.

**Ensure:**
   Whether YOLO is safe under OD threat.
 1: Compute $\mathcal{Z}$ that probabilistically over-approximates $\{F(\boldsymbol{x})\}_{\boldsymbol{x} \in \mathcal{C}}$  ▷ Part 1
 2: **repeat**
 3:    **if** $\forall \boldsymbol{y} \in \mathcal{Z}, \exists box_i \in N(\boldsymbol{y})$ such that $\text{IoU}(box_i, box_{\text{gt}}) \geq \tau \wedge \text{Class}(box_i) = q$ **then**
 4:       **return** Safe.                               ▷ $q = \text{Class}(box_{\text{gt}})$, Part Two
 5:    **else**
 6:       Find a $\boldsymbol{y}' \in \mathcal{Z}$ violating the specified property  ▷ Part 2
 7:       $d_{\min} = \min_{\boldsymbol{x} \in \mathcal{C}} \|F(\boldsymbol{x}) - \boldsymbol{y}'\|_2$  ▷ Part 3
 8:       $\mathcal{Z} = \mathcal{Z} \setminus \mathcal{B}_2(\boldsymbol{y}', d_{\min})$  ▷ Part 3
 9:    **end if**
10: **until** $d_{\min} \leq \kappa$ or $T$ refinement steps have been performed
11: **return** Unsafe  ▷ Part 3

---

**Part 2: NMS Verification.** Verify whether, for all $\boldsymbol{y} \in \mathcal{Z}$, there exists a $box_i \in N(\boldsymbol{y})$ that satisfies the OD safety property (Definition 1). If this holds, the detector is safe. Otherwise, identify a $\boldsymbol{y}$ that violates the IoU or class-matching condition.

**Part 3: Counterexample Validation and Refinement.** Compute $d_{\min} = \min_{\boldsymbol{x} \in \mathcal{C}} \|F(\boldsymbol{x}) - \boldsymbol{y}\|_2$. If $d_{\min} \leq \kappa$ (with $\kappa \geq 0$ as a tolerance), the system is unsafe. Otherwise, refine $\mathcal{Z}$ by excluding $\mathcal{B}_2(\boldsymbol{y}, d_{\min}) = \{\boldsymbol{y}' : \|\boldsymbol{y} - \boldsymbol{y}'\|_2 < d_{\min}\}$ because the regular region we obtained may be larger than the actual output space $\{F(\boldsymbol{x})\}_{\boldsymbol{x} \in \mathcal{C}}$, and then go back to Part 2. Note that we limit the number of Part 3 iterations for high-dimensional outputs to prevent computational overload.

**Remark 2.** *Since our goal is PAC verification, Part 1 and Part 3 are implemented using probabilistic methods with guarantees, rather than exact computation, as detailed in the next section.*

## 5   VERIFICATION METHOD FOR YOLO OBJECT DETECTION

We illustrate the application of the verification framework from Section 4 to YOLO object detection. Because of YOLO's complexity and scale, formal verification becomes intractable; therefore, we adopt PAC verification, i.e., black-box verification via sampling. Proofs for Propositions, Lemmas, and Theorems are provided in the Appendix.

We define the input constraint as $\mathcal{C} = \{\boldsymbol{x} : \|\boldsymbol{x} - \boldsymbol{x}^{(0)}\|_p \leq \varepsilon\}$ for a given sample $\boldsymbol{x}^{(0)}$, norm $p \in \mathbb{Z}^+ \cup \{\infty\}$, and perturbation radius $\varepsilon \in (0, 1)$. We consider a probability distribution over the input set $\mathcal{C}$, and write $\boldsymbol{x} \sim \mathcal{C}$ to denote that $\boldsymbol{x}$ is a sample drawn from this distribution. For convenience, we define the comparison $\boldsymbol{a} \leq \boldsymbol{b}$ for vectors $\boldsymbol{a}, \boldsymbol{b} \in \mathbb{R}^k$ to mean $\forall j \in [k] : \boldsymbol{a}_j \leq \boldsymbol{b}_j$, where $\boldsymbol{a}_j$ is the $j$-th component of $\boldsymbol{a}$. Similarly, scalar-vector multiplication is defined element-wise.

### 5.1   NETWORK OUTPUT APPROXIMATION (PART 1)

Consider a network $F : \mathbb{R}^{d_0} \to \mathbb{R}^{d_L}$ and an input constraint $\mathcal{C}$. In Part 1 of our approach, we aim to determine the range of $\{F(\boldsymbol{x})\}_{\boldsymbol{x} \in \mathcal{C}}$ with a probabilistic guarantee. We first find a constant $c_1 \in \mathbb{R}^+$ and a vector $\boldsymbol{v}_{\max} \in \mathbb{R}^{d_L}$ such that $\forall \boldsymbol{x} \in \mathcal{C}, c_1 \boldsymbol{v}_{\max} \geq |F(\boldsymbol{x}) - F(\boldsymbol{x}^{(0)})|$ holds element-wise. Then let $\mathcal{Z} = \{F(\boldsymbol{x}^{(0)}) + \boldsymbol{\epsilon} : |\boldsymbol{\epsilon}| \leq c_1 \boldsymbol{v}_{\max}\}$, which implies $\{F(\boldsymbol{x})\}_{\boldsymbol{x} \in \mathcal{C}} \subseteq \mathcal{Z}$.

As shown in Algorithm 2, we first randomly select $N_1$ samples from $\mathcal{C}$, and define $(\boldsymbol{v}_{\max})_j = \max\{\max_i\{|F(\boldsymbol{x}^{(i)})_j - F(\boldsymbol{x}^{(0)})_j|\}, \zeta\}$, where $\zeta > 0$ is a small constant to ensure all components are positive. When finding $c_1$, directly solving the problem $c_1 = \min_{c \geq 0} c$   s.t.   $|F(\boldsymbol{x}) - F(\boldsymbol{x}^{(0)})| \leq c \boldsymbol{v}_{\max}, \forall \boldsymbol{x} \in \mathcal{C}$ is infeasible. Since each constraint is convex for $c$, by the $\text{RCP}_N$ (Campi et al., 2009), we can get $c_1$ by randomly selecting $N_2$ samples $\{\boldsymbol{x}^{(i)}\}_{i=1}^{N_2}$ from $\mathcal{C}$, then we calculate $c_1$ by

---

**Algorithm 2** Network Output Approximation (Part 1)

---

**Require:**
   The neural network F; the input constraints $\mathcal{C}$, $N_1, N_2 \in \mathbb{Z}^+$, a threshold $\zeta$, original input $\boldsymbol{x}^{(0)}$.
**Ensure:**
   The output over-approximation region $\mathcal{Z}$.
1: $\{\boldsymbol{x}^{(i)}\}_{i=1}^{N_1} \leftarrow$ Randomly select $N_1$ points in $\mathcal{C}$.                      ▷ Find the $\boldsymbol{v}_{\max}$
2: **for** $j \in [d_L]$ **do**                                                                      ▷ Find the $\boldsymbol{v}_{\max}$
3:     $(\boldsymbol{v}_{\max})_j \leftarrow \max\{\max_i\{|\mathrm{F}(\boldsymbol{x}^{(i)})_j - \mathrm{F}(\boldsymbol{x}^{(0)})_j|\}, \zeta\}$    ▷ Use $\zeta$ to prevent division by zero
4: **end for**
5: $\{\boldsymbol{x}^{(i)}\}_{i=1}^{N_2} \leftarrow$ Randomly select $N_2$ points in $\mathcal{C}$.                      ▷ Find the $c_1$
6: $c_1 \leftarrow \max_{i \in [N_2], j \in [d_L]} \frac{\left|\mathrm{F}(\boldsymbol{x}^{(i)})_j - \mathrm{F}(\boldsymbol{x}^{(0)})_j\right|}{(\boldsymbol{v}_{\max})_j}$.                      ▷ Find the $c_1$
7: **return** $\mathcal{Z} \leftarrow \{\mathrm{F}(\boldsymbol{x}^{(0)}) + \boldsymbol{\epsilon} : |\boldsymbol{\epsilon}| \leq c_1 \boldsymbol{v}_{\max}\}$.

---

**Algorithm 3** NMS Verification (Part 2)

---

**Require:**
   $\{\{box_i^k\}_{i=1}^{n_{\boldsymbol{x}}}\}_{k \in \Delta}$ derived from $\mathcal{Z}$; IoU threshold $\tau$; ground truth bounding box $box_{\mathrm{gt}}$.
**Ensure:**
   Either a non-empty safe set $Q(\mathcal{Z}, \tau, box_{\mathrm{gt}})$, or an unsafe witness $\boldsymbol{z} \in \mathcal{Z}$.
1: $Q \leftarrow \emptyset$.
2: **for** $i \in [n_{\boldsymbol{x}}]$ **do**
3:     Calculate $\tau_1(i, \mathcal{Z}, box_{\mathrm{gt}})$ and $\tau_2(i, \mathcal{Z}, box_{\mathrm{gt}})$                      ▷ Appendix I
4:     $\tau(i, \mathcal{Z}, box_{\mathrm{gt}}) \leftarrow \min(\tau_1(i, \mathcal{Z}, box_{\mathrm{gt}}), \tau_2(i, \mathcal{Z}, box_{\mathrm{gt}}))$                      ▷ Lemma 2
5:     **if** $\tau(i, \mathcal{Z}, box_{\mathrm{gt}}) > \tau$ **then** $Q \leftarrow Q \cup \{i\}$                      ▷ Lemma 1
6:     **end if**
7: **end for**
8: **if** $Q \neq \emptyset$ **then return** (Safe, $Q$)
9: **else** $j \leftarrow \arg\max_{j' \in [n_{\boldsymbol{x}}]}\{\tau(j', \mathcal{Z}, box_{\mathrm{gt}})\}$; $\boldsymbol{z} \leftarrow$ unsafe witness that attains $\tau(j, \mathcal{Z}, box_{\mathrm{gt}})$
10: **end if**
11: **return** (Unsafe, $\boldsymbol{z}$)

---

the following optimization problem:

$$c_1 = \min_{c \geq 0} c \quad \text{s.t.} \quad |\mathrm{F}(\boldsymbol{x}^{(i)}) - \mathrm{F}(\boldsymbol{x}^{(0)})| \leq c\boldsymbol{v}_{\max}, \quad \forall i \in [N_2]. \tag{1}$$

**Proposition 1** (probabilistic guarantee for Part 1). *For any $N_1 > 1$, let $\boldsymbol{v}_{\max}$ be a vector with positive components (e.g., as estimated from $N_1$ samples in Algorithm 2). If $c_1$ is computed based on this $\boldsymbol{v}_{\max}$ using $N_2 \geq \lceil \frac{2\ln(1/\beta)}{\alpha} + 2 + \frac{2\ln(2/\alpha)}{\alpha} \rceil$ samples as described in Algorithm 2, then with probability $1 - \beta$, we have: $\mathbb{P}_{\boldsymbol{x} \sim \mathcal{C}}\left(|\mathrm{F}(\boldsymbol{x}) - \mathrm{F}(\boldsymbol{x}^{(0)})| \leq c_1 \cdot \boldsymbol{v}_{\max}\right) \geq 1 - \alpha$, which implies $\mathbb{P}_{\boldsymbol{x} \sim \mathcal{C}}\left(\mathrm{F}(\boldsymbol{x}) \in \mathcal{Z}\right) \geq 1 - \alpha$.*

**Remark 3.** *The probabilistic guarantee imposes no special requirements on $N_1$. We select $\boldsymbol{v}_{\max}$ in this way because a larger $N_1$ yields a tighter approximation of the true output range (Appendix D).*

## 5.2 NMS Verification (Part 2)

To better illustrate the NMS verification, we use an infinite index set $\Delta$ to enumerate all possible values in $\mathcal{Z}$, i.e., $\mathcal{Z} = \{\boldsymbol{z}^k\}_{k \in \Delta}$, where each $\boldsymbol{z}^k \in \mathcal{Z}$ is a possible output vector. Each $\boldsymbol{z}^k$ can be interpreted as a set of boxes $\{box_i^k\}_{i=1}^{n_{\boldsymbol{x}}}$ according to the YOLO output format. We assume that $box_i^k$ can be written as $box_i^k = (x_i^k, y_i^k, w_i^k, h_i^k, c_i^k, p_{i_1}^k, p_{i_2}^k, \ldots, p_{i_{n_{\mathrm{cls}}}}^k)$. To verify the NMS, we first define the safe set $Q(\mathcal{Z}, \tau, box_{\mathrm{gt}})$, which contains indices of boxes that satisfy the NMS conditions.

**Definition 2** (Safe Set). The safe set $Q(\mathcal{Z}, \tau, box_{\mathrm{gt}}) \subseteq [n_{\boldsymbol{x}}]$ and $i \in Q(\mathcal{Z}, \tau, box_{\mathrm{gt}})$ if and only if:

(1) $\forall k \in \Delta$, $\mathrm{Class}(box_i^k) = \mathrm{Class}(box_{\mathrm{gt}})$, $c_i^k \geq \iota$ and $\mathrm{IoU}(box_i^k, box_{\mathrm{gt}}) \geq \tau$;

(2) $\nexists k \in \Delta, n \in [n_{\boldsymbol{x}}] \setminus \{i\}$ such that $c_n^k \geq \iota$, $\mathrm{Class}(box_n^k) = \mathrm{Class}(box_{\mathrm{gt}})$, $c_n^k \geq c_i^k$, $\mathrm{IoU}(box_i^k, box_n^k) \geq \eta$, and $\mathrm{IoU}(box_{\mathrm{gt}}, box_n^k) < \tau$.

Then we can soundly verify the NMS by checking whether the safe set is empty.

**Proposition 2** (NMS Soundness Verification). *For a given $\mathcal{Z}$, $\tau$, and $box_{\mathrm{gt}}$, if $Q(\mathcal{Z}, \tau, box_{\mathrm{gt}}) \neq \emptyset$, then $\forall k \in \Delta, \exists box_i \in \mathrm{N}(\boldsymbol{z}^k)$ such that $\mathrm{IoU}(box_i, box_{\mathrm{gt}}) \geq \tau \wedge \mathrm{Class}(box_i) = \mathrm{Class}(box_{\mathrm{gt}})$.*

According to this proposition, verification reduces to calculating the safe set. To calculate the safe set, we need the following key metric:

**Definition 3** (Safe IoU Threshold). The Safe IoU Threshold $\tau(i, \mathcal{Z}, box_{\mathrm{gt}}) := \inf\{\tau' \in [0,1] | i \notin Q(\mathcal{Z}, \tau', box_{\mathrm{gt}})\}$, where $\inf$ is the infimum operator.

The following lemmas about $\tau(i, \mathcal{Z}, box_{\mathrm{gt}})$ can help us compute the safe set $Q(\mathcal{Z}, \tau, box_{\mathrm{gt}})$.

**Lemma 1** (Threshold Properties). $\tau < \tau(i, \mathcal{Z}, box_{\mathrm{gt}}) \Rightarrow i \in Q(\mathcal{Z}, \tau, box_{\mathrm{gt}})$

We can obtain $\tau(i, \mathcal{Z}, box_{\mathrm{gt}})$ by solving the following optimization problem:

**Lemma 2** (Threshold Computation). *The threshold can be calculated as $\tau(i, \mathcal{Z}, box_{\mathrm{gt}}) = \min\{\tau_1(i, \mathcal{Z}, box_{\mathrm{gt}}), \tau_2(i, \mathcal{Z}, box_{\mathrm{gt}})\}$, where:*

$$\tau_1(i, \mathcal{Z}, box_{\mathrm{gt}}) = \min_{k \in \Delta} \mathrm{IoU}(box_i^k, box_{\mathrm{gt}}) \cdot \mathbb{I}(\mathrm{Class}(box_i^k) = q) \cdot \mathbb{I}(c_i^k \geq \iota), q = \mathrm{Class}(box_{\mathrm{gt}})$$

$$\tau_2(i, \mathcal{Z}, box_{\mathrm{gt}}) = \begin{cases} \min\limits_{k \in \Delta, n \neq i, \mathcal{C}_{kn} = 1} \mathrm{IoU}(box_n^k, box_{\mathrm{gt}}) & \text{if } \exists(k, n) \text{ s.t. } \mathcal{C}_{kn} = 1 \\ 1 & \text{otherwise} \end{cases}$$

*where constraint $\mathcal{C}_{kn} \equiv \mathbb{I}(c_n^k \geq \iota) \cdot \mathbb{I}(\mathrm{Class}(box_n^k) = q) \cdot \mathbb{I}(\mathrm{IoU}(box_i^k, box_n^k) \geq \eta) \cdot \mathbb{I}(c_n^k \geq c_i^k)$.*

Appendix I shows how we encode the optimization problem in line 3 of Algorithm 3 as a mixed-integer quadratically constrained program (MIQCP) and use the Gurobi solver to solve it.

## 5.3 Counterexample Validation and Refinement (Part 3)

Part 3 of our framework refines the initial output approximation $\mathcal{Z}$. When Part 2 detects a potential counterexample $\boldsymbol{y} \in \mathcal{Z}$, in Part 3, we need to check whether $\boldsymbol{y}$ is actually reachable by F for some $\boldsymbol{x} \in \mathcal{C}$. This is done by computing $d_{\min} = \min_{\boldsymbol{x} \in \mathcal{C}} \|\mathrm{F}(\boldsymbol{x}) - \boldsymbol{y}\|_2$.

Due to the high dimensionality, even if $\boldsymbol{y} \in \{\mathrm{F}(\boldsymbol{x})\}_{\boldsymbol{x} \in \mathcal{C}}$, the $d_{\min}$ derived from the sampled outputs converges to zero very slowly as the sample size increases, so directly estimating $d_{\min}$ simply by taking the minimum distance from a set of sampled outputs $\{\mathrm{F}(\boldsymbol{x}^{(i)})\}$ to $\boldsymbol{y}$ may be unreliable.

To address this, we introduce Algorithm 4, a two-step procedure for estimating $d_{\min}$ with probabilistic guarantees.

**Step One (Estimating C):** This step aims to characterize the local variability of the function F within the input constraint set $\mathcal{C}$. It computes a constant $C = \max_{i \in [N]} \frac{B_i'}{A_i' - B_i'}$ by repeatedly sampling pairs of points and observing the ratio $\frac{B_i'}{A_i' - B_i'}$, where $A_i' = \max_{j \in [M]} \|\mathrm{F}(\boldsymbol{x}^{(i,j)}) - \mathrm{F}(\boldsymbol{x}^{(i)})\|_2$ and $B_i' = \min_{j \in [M]} \|\mathrm{F}(\boldsymbol{x}^{(i,j)}) - \mathrm{F}(\boldsymbol{x}^{(i)})\|_2$.

**Step Two (Estimating $d_{\min}$ using C):** Using the constant $C$ and a new set of $M_2$ samples, this step estimates $d_{\min}$ for the specific target vector $\boldsymbol{y}$. The formula $d_{\min} \leftarrow \max\{\frac{B_m - C(A_m - B_m)}{1 + 2C}, 0\}$ leverages $C$ to provide a more conservative estimate of the minimum distance than $B_m$ (the minimum observed distance from the $M_2$ samples) alone.

Let $\mathrm{V}(\boldsymbol{y}, d_{\min}) = \mathbb{P}_{\boldsymbol{x} \sim \mathcal{C}}(\mathrm{F}(\boldsymbol{x}) \in \mathcal{B}_2(\boldsymbol{y}, d_{\min}))$, and $\mathrm{V}(\boldsymbol{y}, 0) = 0$. We use $\mathrm{V}(\boldsymbol{y}, d_{\min})$ to measure the intersection between $\mathcal{B}_2(\boldsymbol{y}, d_{\min})$ and $\{\mathrm{F}(\boldsymbol{x})\}_{\boldsymbol{x} \in \mathcal{C}}$. We show that with high probability, $\mathrm{V}(\boldsymbol{y}, d_{\min})$ is very small.

**Theorem 1** (Probabilistic Guarantee for Part 3). *Consider Algorithm 4. Here $N$ is the number of outer samples $\{\boldsymbol{x}^{(i)}\}_{i=1}^N$ in Step One, $M$ is the number of samples $\{\boldsymbol{x}^{(i,j)}\}_{j=1}^M$ drawn for each $\boldsymbol{x}^{(i)}$, and $M_2$ is the number of samples used in Step Two.*

*For any $\alpha, \beta, \epsilon \in (0, 1)$ satisfying $\epsilon < \frac{1}{2}$ and $N \cdot ((1 - 2\epsilon)^M) > \frac{1}{\alpha} \ln(\frac{1}{\beta})$, with Algorithm 4, for any $\boldsymbol{y}$, with probability at least $1 - \beta - 2(1 - \epsilon)^{M_2}$ over steps one and two, we have $\mathrm{V}(\boldsymbol{y}, d_{\min}) \leq \alpha$.*

---

**Algorithm 4** Counterexample Validation and Refinement (Part 3)

---

**Require:**
    The neural network F; the input constraints $\mathcal{C}$, $N, M, M_2 \in \mathbb{Z}^+$, a vector $\boldsymbol{y}$.

**Ensure:**
    Estimate $d_{\min} = \min_{\boldsymbol{x} \in \mathcal{C}} \|\boldsymbol{y} - \mathrm{F}(\boldsymbol{x})\|_2$.

1: $C \leftarrow 0$; $\{\boldsymbol{x}^{(i)}\}_{i=1}^N \leftarrow$ Randomly select $N$ samples from $\mathcal{C}$      ▷ Step One
2: **for** $i \in [N]$ **do**
3:      $\{\boldsymbol{x}^{(i,j)}\}_{j=1}^M \leftarrow$ Randomly select $M$ samples from $\mathcal{C}$ again      ▷ Step One
4:      $A_i' \leftarrow \max_{j \in [M]} \|\mathrm{F}(\boldsymbol{x}^{(i,j)}) - \mathrm{F}(\boldsymbol{x}^{(i)})\|_2$, $B_i' \leftarrow \min_{j \in [M]} \|\mathrm{F}(\boldsymbol{x}^{(i,j)}) - \mathrm{F}(\boldsymbol{x}^{(i)})\|_2$      ▷ Step One
5:      $C \leftarrow \max\{C, \frac{B_i'}{A_i' - B_i'}\}$      ▷ Step One
6: **end for**
7: $\{\boldsymbol{x}^{(i)}\}_{i=1}^{M_2} \leftarrow$ Randomly select $M_2$ samples from $\mathcal{C}$      ▷ Step Two
8: Let $A_m \leftarrow \max_{i \in [M_2]} \|\mathrm{F}(\boldsymbol{x}^{(i)}) - \boldsymbol{y}\|_2$; $B_m \leftarrow \min_{i \in [M_2]} \|\mathrm{F}(\boldsymbol{x}^{(i)}) - \boldsymbol{y}\|_2$      ▷ Step Two
9: **return** $d_{\min} \leftarrow \max\{\frac{B_m - C(A_m - B_m)}{1 + 2C}, 0\}$      ▷ Step Two

---

**Remark 4.** *Take $N = 3000$, $M = 10$, $M_2 = 2000$, and $\epsilon = 1/200$, $\alpha, \beta = 0.0099$, then $1 - \beta \geq 0.99$ and $1 - \beta - 2(1 - \epsilon)^{M_2} \geq 0.99$.*

We also provide a sound refinement algorithm for small networks, shown in Appendix K.

### 5.4 THE PROBABILISTIC GUARANTEE FOR THE ENTIRE ALGORITHM

We prove that the whole algorithm implemented above has a probabilistic guarantee by combining Proposition 1, Proposition 2, and Theorem 1:

**Theorem 2.** *(Probabilistic Guarantee for the Entire Algorithm) Using the notation and definitions from Algorithms 1–4. Given $\alpha, \beta, \epsilon \in (0, 1)$ satisfying $\epsilon < \frac{1}{2}$, $N \cdot ((1 - 2\epsilon)^M) > \frac{1}{\alpha} \ln(\frac{1}{\beta})$ and $N_2 \geq \lceil \frac{2\ln(1/\beta)}{\alpha} + 2 + \frac{2\ln(2/\alpha)}{\alpha} \rceil$. Then, after executing the algorithms defined above, for any $\kappa$ used in Algorithm 1, if these algorithms output "Safe" after at most $T$ refinement iterations, then with probability at least $1 - T(\beta + 2(1 - \epsilon)^{M_2}) - \beta$ over Algorithm 2 (Part 1) and Algorithm 4 (Part 3), we have $\mathbb{P}_{\boldsymbol{x} \sim \mathcal{C}}(\boldsymbol{x} \text{ is safe}) > 1 - (1 + T)\alpha$.*

If we take $N_1 = 30,000$, $N_2 = 5,000$, $N = 3,000$, $M = 10$, $M_2 = 2,000$, $\alpha = \beta = 0.0099$, $\epsilon = 1/200$, $T = 1$, we can achieve a 98% probabilistic guarantee with 98% confidence using only 37,000 samples (reuse Part 1 samples in Part 3 Step One), which means with at least 98% confidence, the probability of an OD threat occurring under the given perturbation distribution is at most 2%.

**Remark 5.** *Note all our theoretical guarantees depend only on the i.i.d. assumption and hold for any sampling distribution, not just uniform.*

## 6 EXPERIMENTS

Our experiments evaluate bound accuracy and safety guarantees. Detailed experimental settings and additional results are provided in Appendices N to T.

**Basic setting.** Our experiments use the medium and large versions of the YOLOv3, YOLOv5, YOLOv8 and YOLO11 models by Ultralytics (Jocher et al., 2023). We conduct verification on the COCO dataset (Lin et al., 2014), a widely used benchmark for object detection, and randomly select 100 validation images containing more than 520 objects in total. We use a uniform distribution for sampling, which is a common choice in the literature (Li et al., 2022; Cohen et al., 2024). The IoU threshold is set to $\tau \in \{0.5, 0.7\}$, and the NMS constants are $\eta = 0.45$ and $\iota = 0.25$, which are commonly used in object detection tasks. In Appendix X, we also evaluate our method under other perturbation distributions (e.g., Gaussian and salt-and-pepper noise) and different threat models (e.g., False Appearance). We set $\zeta = 0.001$ (Alg. 2) and $\kappa = 0.01$ (Alg. 1). The perturbation radius $\varepsilon$ is set to $\frac{1}{255}$ or $\frac{2}{255}$. Larger radii make the network overly fragile, enabling counterexamples to be found with very few samples, and thus eliminating meaningful differences between methods.

Table 1: Comparison with $\mathrm{RCP}_N$. $\Delta_{\mathrm{PGD}}$ denotes the mean absolute difference of IoU lower bounds relative to Projected Gradient Descent (PGD) attack. Bold values indicate the best performance.

| $\varepsilon$ | method | model | time | $\Delta_{\mathrm{PGD}}$ | | model | time | $\Delta_{\mathrm{PGD}}$ | |
| | | | | $\tau = 0.5$ | $\tau = 0.7$ | | | $\tau = 0.5$ | $\tau = 0.7$ |
|---|---|---|---|---|---|---|---|---|---|
| $\frac{1}{255}$ | Ours | | **109.0** | **0.49** | **0.45** | | **50.7** | **0.48** | **0.44** |
| | $\mathrm{RCP}_N$ | v3spp | 563.5 | 0.55 | 0.53 | v8m | 455.0 | 0.52 | 0.52 |
| $\frac{2}{255}$ | Ours | | **106.3** | **0.48** | **0.41** | | **49.6** | **0.53** | **0.45** |
| | $\mathrm{RCP}_N$ | | 562.9 | 0.58 | 0.54 | | 454.7 | 0.59 | 0.55 |
| $\frac{1}{255}$ | Ours | | **108.5** | **0.52** | **0.46** | | **132.3** | **0.49** | **0.46** |
| | $\mathrm{RCP}_N$ | v3 | 561.0 | 0.57 | 0.55 | v8x | 591.3 | 0.55 | 0.54 |
| $\frac{2}{255}$ | Ours | | **105.0** | **0.48** | **0.42** | | **129.0** | **0.51** | **0.44** |
| | $\mathrm{RCP}_N$ | | 560.3 | 0.60 | 0.55 | | 590.6 | 0.61 | 0.57 |
| $\frac{1}{255}$ | Ours | | **43.6** | **0.42** | **0.39** | | **59.1** | **0.48** | **0.43** |
| | $\mathrm{RCP}_N$ | v5m | 445.5 | 0.47 | 0.47 | 11m | 468.8 | 0.53 | 0.52 |
| $\frac{2}{255}$ | Ours | | **42.8** | **0.48** | **0.42** | | **58.0** | **0.50** | **0.43** |
| | $\mathrm{RCP}_N$ | | 444.4 | 0.55 | 0.50 | | 467.8 | 0.57 | 0.53 |
| $\frac{1}{255}$ | Ours | | **131.5** | **0.48** | **0.44** | | **147.2** | **0.49** | **0.45** |
| | $\mathrm{RCP}_N$ | v5x | 593.8 | 0.54 | 0.54 | 11x | 618.1 | 0.54 | 0.54 |
| $\frac{2}{255}$ | Ours | | **128.4** | **0.52** | **0.45** | | **141.9** | **0.50** | **0.44** |
| | $\mathrm{RCP}_N$ | | 593.2 | 0.60 | 0.57 | | 616.5 | 0.62 | 0.57 |

Table 2: Guarantee evaluation of our method with $\tau = 0.5$ and $\varepsilon \in \{\frac{1}{255}, \frac{2}{255}\}$ under $10^6$ uniform perturbations. TPR/FPR: True/False Positive Rate. TNR/FNR: True/False Negative Rate. A detection is considered positive if verified robust by our method, and negative otherwise. Certified Robust Accuracy (CRA): percentage of detections verified robust that are indeed robust. Average Bounds Improvement (ABI): mean improvement of certified IoU lower bounds without refinement.

| model | $\varepsilon$ | TPR (%) | FPR (%) | TNR (%) | FNR (%) | CRA (%) | ABI |
|---|---|---|---|---|---|---|---|
| yolo11x | 1/255 | 94.9 | 2.9 | 97.1 | 5.1 | 98.9 | 0.10 |
| | 2/255 | 85.2 | 1.4 | 98.6 | 14.8 | 99.4 | 0.17 |
| yolo11m | 1/255 | 95.0 | 3.0 | 97.0 | 5.0 | 98.6 | 0.10 |
| | 2/255 | 93.1 | 1.1 | 98.9 | 6.9 | 99.4 | 0.11 |
| yolov8x | 1/255 | 95.1 | 0.0 | 100.0 | 4.9 | 100.0 | 0.09 |
| | 2/255 | 89.7 | 0.6 | 99.4 | 10.3 | 99.7 | 0.14 |
| yolov8m | 1/255 | 97.3 | 2.6 | 97.4 | 2.7 | 98.9 | 0.08 |
| | 2/255 | 93.9 | 2.4 | 97.6 | 6.1 | 98.8 | 0.11 |
| yolov5xu | 1/255 | 94.8 | 0.7 | 99.3 | 5.2 | 99.7 | 0.09 |
| | 2/255 | 90.6 | 0.6 | 99.4 | 9.4 | 99.7 | 0.13 |
| yolov5mu | 1/255 | 96.0 | 2.9 | 97.1 | 4.0 | 98.6 | 0.10 |
| | 2/255 | 92.4 | 0.5 | 99.5 | 7.6 | 99.7 | 0.11 |
| yolov3-sppu | 1/255 | 95.6 | 0.0 | 100.0 | 4.4 | 100.0 | 0.08 |
| | 2/255 | 87.4 | 0.0 | 100.0 | 12.6 | 100.0 | 0.15 |
| yolov3u | 1/255 | 95.7 | 1.3 | 98.7 | 4.3 | 99.4 | 0.09 |
| | 2/255 | 85.8 | 0.0 | 100.0 | 14.2 | 100.0 | 0.16 |

**Baseline Selection.** By Theorem 2, our method achieves a 98% probabilistic guarantee with 98% confidence using only **37,000** samples. In contrast, $\mathrm{RCP}_N$ requires over **11,000,000** samples, while DeepPAC (Li et al., 2022) requires over **100,000,000** samples and needs to solve linear programs (LPs) with more than $10^{12}$ variables to achieve the same guarantee (see Appendix N), making both

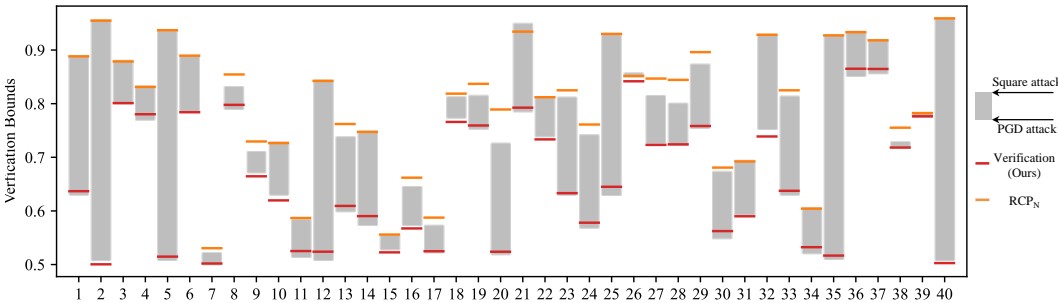

Figure 5: Lower bounds on IoU between detected boxes and their ground-truth counterparts under our method, $RCP_N$, the square attack, and PGD. Results are reported for YOLO11x with $\varepsilon = \frac{1}{255}$ and $\tau = 0.5$. Each x-axis tick represents one object in COCO.

approaches impractical. Formal verification methods are also infeasible: existing tools (Cohen et al., 2024; Elboher et al., 2024) handle only 2-3 convolutional layers with 2-3 linear layers, far below the scale of YOLO, and cannot address its complex architecture or NMS. Therefore, **full-guarantee direct comparisons** with DeepPAC, $RCP_N$, and formal verification are not feasible. Instead, we report $RCP_N$ with $10^6$ samples as a weaker-guarantee empirical baseline.

**Bounds Accuracy.** Table 1 compares our method with $RCP_N$, showing that our approach is both faster and more accurate. In particular, it achieves a smaller mean absolute difference between IoU lower bounds and the worst-case input found by the PGD attack ($\Delta_{\mathrm{PGD}}$), indicating tighter certified bounds. Figure 5 further confirms this, as our bounds remain consistently closer to those of PGD.

**Safety Guarantee.** Table 2 further shows results under $10^6$ uniform perturbations: the certified robust accuracy (CRA) exceeds 98%, and the false positive rate (FPR) remains very low, consistent with theory. The true positive rate (TPR) is lower, as expected since our certification is stricter than empirical robustness. Finally, the average bounds improvement (ABI) confirms that our method yields tighter certified IoU lower bounds.

**Additional Experiments.** We further evaluate (i) the effect of Part 3 (Appendix P), (ii) an ablation study on hyperparameters (Appendix T), (iii) real-world applications (Appendix R), and (iv) a comparison with median smoothing (Appendix S). All results confirm the effectiveness of our method and its strong probabilistic guarantees.

## 7 CONCLUSION

This paper presents a new verification framework that provides provable probabilistic guarantees for YOLO-based object detection systems under specific threat models (e.g., object disappearance, false appearance), across a range of perturbation distributions. The framework applies to any distribution from which we can draw independent and identically distributed (i.i.d.) samples, and it can be extended to other threat models by changing the definition and encoding of the safety property in the NMS verification stage. Our contributions are threefold: (i) we give a formal definition of the object detection verification problem; (ii) we propose a practical three-stage methodology that explicitly includes formal analysis of non-maximum suppression; and (iii) we derive strong probabilistic guarantees for the full detection pipeline. We further show that adapting the method to other threat models only requires modifying the NMS verification condition, while keeping the other stages unchanged. Experiments on multiple YOLO architectures, perturbation distributions, and threat models demonstrate that our approach provides reliable safety assurances and achieves tighter certified IoU bounds with much higher sample efficiency than prior methods.

**Limitations and Future Directions:** Our method relies on an assumed distribution of input perturbations, a limitation inherent to the PAC framework. Developing verification methods for other types of attacks remains an important direction for future work. Another valuable direction involves leveraging adversarial attack strategies to further refine Part 1 and Part 3, alongside investigating more efficient methods for interval estimation and refinement.

## ACKNOWLEDGMENTS

This work is supported by the Chinese Academy of Sciences (CAS) Project for Young Scientists in Basic Research, Grant No.YSBR-040, the Institute of Software, Chinese Academy of Sciences (ISCAS) New Cultivation Project ISCAS-PYFX-202201, and ISCAS Basic Research ISCAS-JCZD-202302. The authors thank anonymous referees for their valuable comments.

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

# A  APPENDIX

This appendix provides more related work, supplementary discussions, proofs, and experiments to support the main text. We organize the appendix as follows:

- **Section B**: Summarizes additional related work.
- **Section C**: Provides the proof of Proposition 1 (probabilistic guarantee for part 1).
- **Section D**: Explains the role of sample size in Part 1 Step 1 and its effect on $v_{\max}$.
- **Section E**: Gives the proof of Proposition 2 (NMS Soundness Verification).
- **Section F**: Proves Lemma 1 (threshold properties).
- **Section G**: Proves Lemma 2 (explicit threshold computation formulas).
- **Section H**: Defines and estimates the verification bound for object detection and provides proofs for Lemmas 4 and 5.
- **Section I**: Details the verification procedure for Non-Maximum Suppression, including abstract box construction and IoU bound computation.
- **Section J**: Proves Theorem 1 (probabilistic guarantee for Part 3).
- **Section K**: Proposes a strict sound gradient-based refinement algorithm to implement Part Three on small-scale networks (Theorem 3).
- **Section L**: Provides the proof of Theorem 3 based on dual formulations.
- **Section M**: Gives the proof of Theorem 2, combining probabilistic guarantees across all parts.
- **Section N**: Reports detailed experimental settings, sample number calculations, and server configuration.
- **Section O**: Discusses the efficiency of the NMS verification process.
- **Section P**: Shows the effectiveness of Part 3 refinement for YOLO and CNNs.
- **Section Q**: Shows verification time comparison between our method and $\text{RCP}_N$.
- **Section R**: Demonstrates our method's effectiveness on real-world images.
- **Section S**: Compares our method against median smoothing under Gaussian noise.
- **Section T**: Presents an ablation study on parameters $\eta$ and $\iota$.
- **Section U**: Discusses the broader impact of our verification method.
- **Section V**: LLM Usage Statement.
- **Section W**: Lists and explains all hyperparameters used in our algorithms.
- **Section X**: Provides details on how to adapt our method to other attacks beyond OD.

# B  ADDITIONAL RELATED WORK

**Adversarial attacks.**  Adversarial methods induce misclassification through imperceptible perturbations. White-box attacks exploit gradient information (Goodfellow et al., 2015; Madry et al., 2018; Carlini & Wagner, 2017) and can be adapted to OD attacks (Choi & Tian, 2022; Li et al., 2020). Black-box attacks use transferability (Chen & Liu, 2024) or query-based optimization (Li et al., 2020); these are more practical but may be computationally costly. Using adversarial attacks in isolation only demonstrates non-robustness if the attack is successful, whereas our method provides a rigorous probabilistic certificate of robustness. A network that is robust with high probability under common perturbations (e.g., sensor noise) is acceptable for practical deployment, whereas the strict requirement of complete robustness in a neighborhood often leads to a significant drop in network performance. Thus, our focus is on providing probabilistic robustness under realistic perturbations such as sensor noise rather than adversarial attacks (which often represent worst-case scenarios).

**Bound Estimation Methods.** There are several classical methods for estimating probabilistic output bounds of neural networks, including DKW-based confidence regions (Massart, 1990; Naaman, 2021), ERM-based hyper-rectangles, and $\epsilon$-nets (Haussler & Welzl, 1987; Blohm et al., 2025). While these methods are theoretically robust, their sample complexity typically scales with the output dimension $d_L$ (e.g., $\tilde{O}(d_L/\epsilon)$ for $\epsilon$-nets). Given the extremely high-dimensional output space of YOLO networks, dimension-dependent bounds like those from $\epsilon$-nets or DKW would be computationally infeasible. We therefore focus on dimension-independent PAC bounds, making estimation feasible even for high-dimensional outputs.

**PAC with Attack.** There are several PAC verification methods that incorporate adversarial attacks to refine the estimated output bounds (Blohm et al., 2025; Li et al., 2022; Baluta et al., 2021). For example, Blohm et al. (2025) evaluates the robustness of individual points via a local robustness oracle (such as PGD or LiRPA) and leverages $\epsilon$-net sampling methods to provide high-probability statistical guarantees for global robustness. Combining attacks with PAC verification is an interesting direction, and we will explore this in future work.

## C  THE PROOF OF PROPOSITION 1

> **Proposition** (probabilistic guarantee for part 1). *For any $N_1 > 1$, let $\boldsymbol{v}_{\max}$ be a vector with positive components (e.g., as estimated from $N_1$ samples in Algorithm 2). If $c_1$ is computed based on this $\boldsymbol{v}_{\max}$ using $N_2 \geq \lceil \frac{2\ln(1/\beta)}{\alpha} + 2 + \frac{2\ln(2/\alpha)}{\alpha} \rceil$ samples as described in Algorithm 2, then with probability $1 - \beta$, we have: $\mathbb{P}_{\boldsymbol{x}\sim\mathcal{C}}\left(|\mathrm{F}(\boldsymbol{x}) - \mathrm{F}(\boldsymbol{x}^{(0)})| \leq c_1 \cdot \boldsymbol{v}_{\max}\right) \geq 1 - \alpha$, which implies $\mathbb{P}_{\boldsymbol{x}\sim\mathcal{C}}\left(\mathrm{F}(\boldsymbol{x}) \in \mathcal{Z}\right) \geq 1 - \alpha$.*

This proposition can be directly obtained by the classic method $\mathrm{RCP}_N$, which is introduced below.

### C.1  CLASSIC METHOD FOR PROBABILISTIC SAMPLING.

We begin by introducing a well-known result from the $\mathrm{RCP}_N$ method (Campi et al., 2009), which forms the basis of our approach in this section. Consider the following optimization problem with an infinite number of constraints:

$$\min_{\boldsymbol{x}\in\mathbb{R}^d} \boldsymbol{a}^\top \boldsymbol{x} + b \quad \text{s.t.} \quad \boldsymbol{x} \in \bigcap_{\delta\in\Delta} \mathcal{X}_\delta, \tag{2}$$

where $\Delta$ is an index set, and $\mathcal{X}_\delta \subseteq \mathbb{R}^d$ denotes the constraint set corresponding to the index $\delta$.

Since there are infinitely many constraints, we cannot solve the problem directly. Thus we sample the constraints $\{\delta_i\}_{i=1}^N$ from $\Delta$ according to a probability distribution $\mathcal{Q}$ on $\Delta$, and consider the sampled problem equation 3. We are interested in quantifying how likely it is that the optimal solution of the sampled problem is (almost) feasible for the original problem.

$$\min_{\boldsymbol{x}\in\mathbb{R}^d} \boldsymbol{a}^\top \boldsymbol{x} + b \quad \text{s.t.} \quad \boldsymbol{x} \in \bigcap_{i\in[N]} \mathcal{X}_{\delta_i}. \tag{3}$$

When $\mathcal{X}_\delta$ is convex for every $\delta \in \Delta$, we have the following result. If $\mathcal{Q}$ is a distribution defined on $\Delta$, and

$$N \geq \left\lceil \frac{2\ln(1/\beta)}{\alpha} + 2d + \frac{2d\ln(2/\alpha)}{\alpha} \right\rceil,$$

then, with probability at least $1 - \beta$ over the i.i.d. samples $\{\delta_i\}_{i=1}^N \sim \mathcal{Q}^N$, if the optimization problem equation 3 has a unique solution $\boldsymbol{x}_{\min}$, this solution $\boldsymbol{x}_{\min}$ satisfies

$$\mathbb{P}_{\delta\sim\mathcal{Q}}\left(\boldsymbol{x}_{\min} \in \mathcal{X}_\delta\right) \geq 1 - \alpha.$$

A classic application of $\mathrm{RCP}_N$ is to compute a high-probability upper bound of a function $f(t)$ over a domain $\Delta$. This can be formulated as an optimization problem with a decision variable $x \in \mathbb{R}$ (i.e., dimension $d = 1$):

$$\min_{x\in\mathbb{R}} x \quad \text{s.t.} \quad x \geq f(t), \ \forall t \in \Delta. \tag{4}$$

By applying the $\mathrm{RCP}_N$ result with $d = 1$, we know that if we draw $N$ samples $\{t_i\}_{i=1}^N$ from $\Delta$ according to $\mathcal{Q}$, where

$$N \geq \left\lceil \frac{2\ln(1/\beta)}{\alpha} + 2 + \frac{2\ln(2/\alpha)}{\alpha} \right\rceil,$$

then, with probability at least $1 - \beta$ over the sampling, we have $\mathbb{P}_{t\sim\mathcal{Q}}\big(f(t) \leq \max_{i=1,\ldots,N} f(t_i)\big) \geq 1 - \alpha$.

# D  ANALYSIS OF THE SAMPLE SIZE EFFECT IN PART 1 STEP 1

In this section, we justify the choice of $\boldsymbol{v}_{\max}$ as presented in Algorithm 2. The main objective is to ensure that the range $\mathcal{Z}$ obtained by Algorithm 2 does not significantly exceed the actual range $\{\mathrm{F}(\boldsymbol{x})\}_{\boldsymbol{x}\in\mathcal{C}}$.

Algorithm 2 guarantees this property under certain assumptions, as detailed in the following proposition.

**Proposition 3.** *Let* $|(\mathrm{F}(\boldsymbol{x}))_i - (\mathrm{F}(\boldsymbol{x}^{(0)}))_i| \sim N_i$ *when* $\boldsymbol{x} \sim \mathcal{C}$, *and* $v_i \in \mathbb{R}_+$ *is the minimum value such that* $\mathbb{P}_{\boldsymbol{x}\sim N_i}(\boldsymbol{x} \leq v_i) = 1$.

*If* $\alpha_1^i \leq \alpha_2^i \leq 1$ *and* $\beta_1, \beta_2$ *satisfy that:* $\mathbb{P}_{x\sim N_i}(x \leq \alpha_1^i v_i) = \beta_1$ *and* $\mathbb{P}_{x\sim N_i}(x \leq \alpha_2^i v_i) = \beta_2$ *for any* $i \in [d_L]$.

*Let* $z_i = (c_1 \boldsymbol{v}_{\max})_i$ *where* $c_1$ *and* $\boldsymbol{v}_{\max}$ *are obtained by algorithm 2, then we have that:* $\frac{z_j}{v_j} \leq \max_i\{\frac{\alpha_2^i}{\alpha_1^i}\}\alpha_2^j$ *for any* $j \in [d_L]$ *with probability* $1 - d_L(1 + \beta_1^{N_1} - \beta_2^{N_1}) - d_L(1 + \beta_1^{N_2} - \beta_2^{N_2})$.

*Proof.* Observe that for any coordinate $i \in [d_L]$, given the $N_1$ randomly selected points $\{x_k\}$ from $\mathcal{C}$ in Algorithm 2, we have:

$$\mathbb{P}\left(\alpha_2^i v_i \geq \max_k (x_k)_i \geq \alpha_1^i v_i\right) \geq \beta_2^{N_1} - \beta_1^{N_1}.$$

Applying the union bound, with probability $1 - d_L(1 - (\beta_2^{N_1} - \beta_1^{N_1})) = 1 - d_L(1 + \beta_1^{N_1} - \beta_2^{N_1})$, the condition $\alpha_2^i v_i \geq \max_k (x_k)_i \geq \alpha_1^i v_i$ holds for all $i \in [d_L]$.

Similarly, for the $N_2$ points $\{x_k'\}$ randomly selected in Algorithm 2, with probability $1 - d_L(1 + \beta_1^{N_2} - \beta_2^{N_2})$, the condition $\alpha_2^i v_i \geq \max_k (x_k')_i \geq \alpha_1^i v_i$ holds for all $i \in [d_L]$.

Consequently, based on the construction in Algorithm 2, if the above events occur, we have $c_1 \leq \max_i\{\alpha_2^i/\alpha_1^i\}$. Therefore, for any $j \in [d_L]$, implies:

$$(c_1 \boldsymbol{v}_{\max})_j \leq \max_i \left\{\frac{\alpha_2^i}{\alpha_1^i}\right\} \alpha_2^j v_j,$$

which completes the proof. $\square$

Based on our observations of numerous neural network outputs, we find that the output values are highly likely to be concentrated within a specific region. An example is provided in Figure 6. Based on this observation, it is possible that $\alpha_1 \approx \alpha_2$ while $\beta_2 \approx 1$ and $\beta_1 < 1$. Consequently, since $\alpha_1 \approx \alpha_2$, we can infer that each dimension of $\mathcal{Z}$ will not extend significantly beyond $v_i$. Furthermore, since $\beta_2 \approx 1$ and $\beta_1 < 1$, the probability term satisfies:

$$1 - d_L(1 + \beta_1^{N_1} - \beta_2^{N_1}) - d_L(1 + \beta_1^{N_2} - \beta_2^{N_2}) \approx 1.$$

However, in practice, we cannot accurately estimate $\alpha_i$ and $\beta_i$. Therefore, the choice of $N_1$ is primarily determined empirically. The proposition above serves to theoretically justify the accuracy of the region $\mathcal{Z}$ obtained when $N_1$ is sufficiently large.

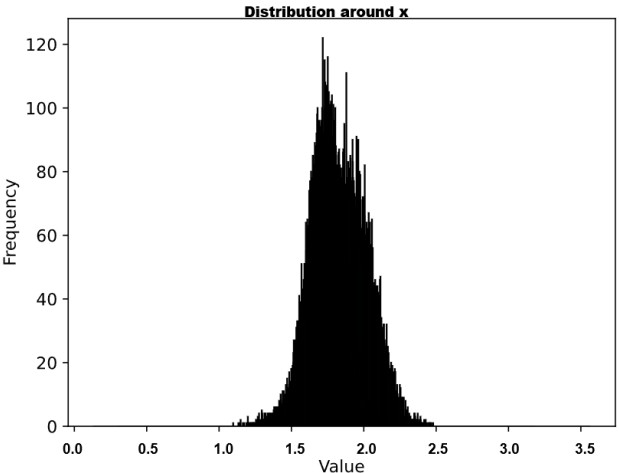

Figure 6: This figure shows the distribution around a sample $x$. When we take $\alpha_1 = 2.3/3.5$ and $\alpha_2 = 2.8/3.5$, from this figure, we have that $\beta_1 \leq 0.99$ but $\beta_2 \approx 1$. Thus $(\alpha_2/\alpha_1)\alpha_2 \approx 0.97$. Hence, if $N_1 = N_2 = 3000$, then $1 - d_L(1 + \beta_1^{N_1} - \beta_2^{N_1}) - d_L(1 + \beta_1^{N_2} - \beta_2^{N_2}) \geq 0.99$ when $d_L \leq 10^7$.

# E   THE PROOF OF PROPOSITION 2

**Proposition** (NMS Soundness Verification). *For a given $\mathcal{Z}$, $\tau$, and $box_{\text{gt}}$, if $Q(\mathcal{Z}, \tau, box_{\text{gt}}) \neq \emptyset$, then for all $k \in \Delta$, there exists $box_i \in \mathrm{N}(\boldsymbol{z}^k)$ such that $\mathrm{IoU}(box_i, box_{\text{gt}}) \geq \tau$ and $\mathrm{Class}(box_i) = \mathrm{Class}(box_{\text{gt}})$.*

We first present a lemma.

**Lemma 3.** *For a given $\boldsymbol{z} \in \mathcal{Z}$, if there is no $box_i \in \mathrm{N}(\boldsymbol{z})$ that satisfies $\mathrm{IoU}(box_i, box_{\text{gt}}) \geq \tau$ and $\mathrm{Class}(box_i) = \mathrm{Class}(box_{\text{gt}})$, then:*
*For any $box_i \in \boldsymbol{z}$ with $c_i \geq \iota$, $\mathrm{Class}(box_i) = \mathrm{Class}(box_{\text{gt}})$, and $\mathrm{IoU}(box_i, box_{\text{gt}}) \geq \tau$, there exists another $box_j \in \boldsymbol{z}$ such that $c_j \geq \iota$, $\mathrm{Class}(box_j) = \mathrm{Class}(box_i)$, $c_i \leq c_j$, $\mathrm{IoU}(box_j, box_{\text{gt}}) < \tau$, and $\mathrm{IoU}(box_i, box_j) \geq \eta$.*

*Proof.* Assume that there exists a $box_i \in \boldsymbol{z}$ with $c_i \geq \iota$, $\mathrm{Class}(box_i) = \mathrm{Class}(box_{\text{gt}})$, and $\mathrm{IoU}(box_i, box_{\text{gt}}) \geq \tau$. By the assumption of the lemma, it implies that $box_i \notin \mathrm{N}(\boldsymbol{z})$.

According to non-maximum suppression condition (n2) in Section 3.1, there must exist a $box_j \in \mathrm{N}(\boldsymbol{z})$ such that $\mathrm{IoU}(box_i, box_j) \geq \eta$, $c_i \leq c_j$, and $\mathrm{Class}(box_i) = \mathrm{Class}(box_j) = \mathrm{Class}(box_{\text{gt}})$. By condition (n1) in Section 3.1, $box_j \in \mathrm{N}(\boldsymbol{z})$ implies that $c_j \geq \iota$.

However, by the assumptions, there is no $box_j \in \mathrm{N}(\boldsymbol{z})$ that satisfies $\mathrm{Class}(box_j) = \mathrm{Class}(box_{\text{gt}})$ and $\mathrm{IoU}(box_j, box_{\text{gt}}) \geq \tau$. Since we have shown that $\mathrm{Class}(box_j) = \mathrm{Class}(box_{\text{gt}})$ above, it follows that $\mathrm{IoU}(box_j, box_{\text{gt}}) < \tau$. This completes the proof. $\square$

Using this lemma, we can directly prove Proposition 2.

*Proof.* Assume that $Q(\mathcal{Z}, \tau, box_{\text{gt}}) \neq \emptyset$, but for some $k \in \Delta$, the stated result does not hold.

Based on condition (1) of the definition of $Q(\mathcal{Z}, \tau, box_{\text{gt}})$, let $i \in Q(\mathcal{Z}, \tau, box_{\text{gt}})$. Then, based on Lemma 3, we know that there exists another $box_j^k \in \boldsymbol{z}^k$ such that $c_j^k \geq \iota$, $\mathrm{Class}(box_j^k) = \mathrm{Class}(box_i^k)$, $c_i^k \leq c_j^k$, $\mathrm{IoU}(box_j^k, box_{\text{gt}}) < \tau$, and $\mathrm{IoU}(box_i^k, box_j^k) \geq \eta$. This contradicts condition (2) of the definition of $Q(\mathcal{Z}, \tau, box_{\text{gt}})$.

Thus, the assumption leads to a contradiction, and the result follows. $\square$

## F  THE PROOF OF LEMMA 1

**Lemma** (Threshold Properties). $\tau < \tau(i, \mathcal{Z}, box_{\mathrm{gt}}) \Rightarrow i \in Q(\mathcal{Z}, \tau, box_{\mathrm{gt}})$

*Proof.* First, we demonstrate that for any $\tau' > \tau$, the inclusion $Q(\mathcal{Z}, \tau', box_{\mathrm{gt}}) \subseteq Q(\mathcal{Z}, \tau, box_{\mathrm{gt}})$ holds. Indeed, if an index $i$ satisfies Condition (1) in Definition 2 with a threshold $\tau'$, it necessarily satisfies Condition (1) with any lower threshold $\tau < \tau'$, assuming the input constraints and ground truth remain unchanged. A similar argument applies to Condition (2) in Definition 2. This establishes the monotonicity result.

Now, suppose for the sake of contradiction that $\tau < \tau(i, \mathcal{Z}, box_{\mathrm{gt}})$ and $i \notin Q(\mathcal{Z}, \tau, box_{\mathrm{gt}})$. By the monotonicity established above, this implies that $i \notin Q(\mathcal{Z}, \tau', box_{\mathrm{gt}})$ for any $\tau' > \tau$. According to the definition of $\tau(i, \mathcal{Z}, box_{\mathrm{gt}})$, this entails $\tau \geq \tau(i, \mathcal{Z}, box_{\mathrm{gt}})$, which contradicts the initial hypothesis that $\tau < \tau(i, \mathcal{Z}, box_{\mathrm{gt}})$. Thus, the assumption leads to a contradiction, which completes the proof. □

## G  THE PROOF OF LEMMA 2

**Lemma** (Threshold Computation).
*The threshold can be calculated as* $\tau(i, \mathcal{Z}, box_{\mathrm{gt}}) = \min\{\tau_1(i, \mathcal{Z}, box_{\mathrm{gt}}), \tau_2(i, \mathcal{Z}, box_{\mathrm{gt}})\}$*, where:*

$$\tau_1(i, \mathcal{Z}, box_{\mathrm{gt}}) = \min_{k \in \Delta} \mathrm{IoU}(box_i^k, box_{\mathrm{gt}}) \cdot \mathbb{I}(\mathrm{Class}(box_i^k) = q) \cdot \mathbb{I}(c_i^k \geq \iota), q = \mathrm{Class}(box_{\mathrm{gt}})$$

$$\tau_2(i, \mathcal{Z}, box_{\mathrm{gt}}) = \begin{cases} \min\limits_{k \in \Delta, n \neq i, \mathcal{C}_{kn}=1} \mathrm{IoU}(box_n^k, box_{\mathrm{gt}}) & \text{if } \exists(k, n) \text{ s.t. } \mathcal{C}_{kn} = 1 \\ 1 & \text{otherwise} \end{cases}$$

*where constraint* $\mathcal{C}_{kn} \equiv \mathbb{I}(c_n^k \geq \iota) \cdot \mathbb{I}(\mathrm{Class}(box_n^k) = q) \cdot \mathbb{I}(\mathrm{IoU}(box_i^k, box_n^k) \geq \eta) \cdot \mathbb{I}(c_n^k \geq c_i^k)$.

*Proof.* We prove the equality by showing both directions of the inequality.

**Part 1: Proof of** $\tau(i, \mathcal{Z}, box_{\mathrm{gt}}) \leq \min\{\tau_1, \tau_2\}$**.**

Observe that if $\tau' > \tau_1(i, \mathcal{Z}, box_{\mathrm{gt}})$, index $i$ is not contained in the safe set $Q(\mathcal{Z}, \tau', box_{\mathrm{gt}})$ because it violates Condition (1) in Definition 2. Similarly, if $\tau' > \tau_2(i, \mathcal{Z}, box_{\mathrm{gt}})$, index $i$ is excluded from the safe set $Q(\mathcal{Z}, \tau', box_{\mathrm{gt}})$ because it violates Condition (2), even if Condition (1) is satisfied. Consequently, the threshold must satisfy:

$$\tau(i, \mathcal{Z}, box_{\mathrm{gt}}) \leq \min\{\tau_1(i, \mathcal{Z}, box_{\mathrm{gt}}), \tau_2(i, \mathcal{Z}, box_{\mathrm{gt}})\}.$$

**Part 2: Proof of** $\tau(i, \mathcal{Z}, box_{\mathrm{gt}}) \geq \min\{\tau_1, \tau_2\}$**.**

Note that when $\tau' < \tau_1(i, \mathcal{Z}, box_{\mathrm{gt}})$, index $i$ must satisfy Condition (1) in Definition 2. Likewise, when $\tau' < \tau_2(i, \mathcal{Z}, box_{\mathrm{gt}})$, index $i$ must satisfy Condition (2). Therefore, if we choose a threshold $\tau < \min\{\tau_1(i, \mathcal{Z}, box_{\mathrm{gt}}), \tau_2(i, \mathcal{Z}, box_{\mathrm{gt}})\}$, index $i$ satisfies both conditions, which implies $i \in Q(\mathcal{Z}, \tau, box_{\mathrm{gt}})$. By the definition of the threshold $\tau(i, \mathcal{Z}, box_{\mathrm{gt}})$, this implies:

$$\tau(i, \mathcal{Z}, box_{\mathrm{gt}}) \geq \min\{\tau_1(i, \mathcal{Z}, box_{\mathrm{gt}}), \tau_2(i, \mathcal{Z}, box_{\mathrm{gt}})\}.$$

Combining the results from Part 1 and Part 2, the equality holds. □

## H  COMPUTING THE VERIFICATION BOUND

In this section, we show how to **Calculate the verification bound** for NMS.

We first define the verification bounds:

**Definition 4** (OD Verification Bound). For constraints $\mathcal{C}$ and $box_{\mathrm{gt}}$, define:

$$\min_{\boldsymbol{x} \in \mathcal{C}} \max_{box_i \in \mathrm{N}(\mathrm{F}(\boldsymbol{x}))} \mathrm{IoU}(box_i, box_{\mathrm{gt}}) \cdot \mathbb{I}(\mathrm{Class}(box_i) = \mathrm{Class}(box_{\mathrm{gt}})),$$

as the OD verification bound. This quantifies robustness against OD attacks.

We need to estimate the verification bound $\min_{z \in \mathcal{Z}} \max_{box_i \in N(z)} \text{IoU}(box_i, box_{\text{gt}}) \mathbb{I}(\text{Class}(box_i) = \text{Class}(box_{\text{gt}}))$ under output over-approximation region $\mathcal{Z}$ and ground truth $box_{\text{gt}}$, according to definition 4.

To estimate such a bound, we first need the following lemma:

**Lemma 4.** *For any $\mathcal{Z}, \tau, box_{\text{gt}}$, there is:*

$$\min_{z \in \mathcal{Z}} \max_{box_i \in N(z)} \text{IoU}(box_i, box_{\text{gt}}) \mathbb{I}(\text{Class}(box_i) = \text{Class}(box_{\text{gt}})) \geq \tau \mathbb{I}(\{|Q(\mathcal{Z}, \tau, box_{\text{gt}})| \geq 1\})$$

So we try to find the maximum $\tau$ such that $|Q(\mathcal{Z}, \tau, box_{\text{gt}})| > 0$, but this maximum may not exist in some cases (it may only be approached arbitrarily), so we consider the following value instead:

$$\min_{\tau \in [0,1]} \tau \text{ s.t. } |Q(\mathcal{Z}, \tau', box_{\text{gt}})| = 0, \; \forall 1 \geq \tau' > \tau \tag{5}$$

We use the following lemma to calculate such minimum value:

**Lemma 5.** *The solution of problem 5 is equal to:* $\max_{i \in [n_x]} \tau(i, \mathcal{Z}, box_{\text{gt}})$.

Using this lemma, we only need to calculate $\tau(i, \mathcal{Z}, box_{\text{gt}})$ as described above, and then we can estimate the verification bound.

### H.1 THE PROOF OF LEMMA 4

*Proof.* By Proposition 2, if $|Q(\mathcal{Z}, \tau, box_{\text{gt}})| \geq 1$, then for any $z \in \mathcal{Z}$, there exists a $box_i \in N(z)$ such that $\text{IoU}(box_i, box_{\text{gt}}) \geq \tau$ and $\text{Class}(box_i) = \text{Class}(box_{\text{gt}})$. This implies that the value of $\min_{z \in \mathcal{Z}} \max_{box_i \in N(z)} \text{IoU}(box_i, box_{\text{gt}}) \cdot \mathbb{I}(\text{Class}(box_i) = \text{Class}(box_{\text{gt}}))$ is greater than or equal to $\tau$, which yields the desired result. $\qquad\square$

### H.2 THE PROOF OF LEMMA 5

*Proof.* We have shown that when $\tau' > \tau$, we have $Q(\mathcal{Z}, \tau', box_{\text{gt}}) \subset Q(\mathcal{Z}, \tau, box_{\text{gt}})$ in Lemma 1.

By the definition of $\tau(i, \mathcal{Z}, box_{\text{gt}})$ and above result, we know that the safe set $Q(\mathcal{Z}, \tau', box_{\text{gt}})$ is $\emptyset$ for any $\tau' > \max_{i \in [n_x]} \tau(i, \mathcal{Z}, box_{\text{gt}})$, so the solution of problem 5 is at most $\max_{i \in [n_x]} \tau(i, \mathcal{Z}, box_{\text{gt}})$.

If $\tau$ is the solution of problem 5, then for any $i \in [n_x]$, there must be $i \notin Q(\mathcal{Z}, \tau', box_{\text{gt}})$ for any $\tau' > \tau$, so $\tau \geq \max_{i \in [n_x]} \tau(i, \mathcal{Z}, box_{\text{gt}})$.

This completes the proof of the lemma.

$\qquad\square$

## I VERIFICATION PROCESS OF NMS

To illustrate Non-Maximum Suppression (NMS) verification, we define constraints as $\mathcal{Z} = \mathcal{H} \setminus \mathcal{S}$. Here, $\mathcal{H}$ constrains the neural network output $\boldsymbol{y}$, and $\mathcal{S}$ constrains bounding box parameters. This formulation is equivalent to the original. Algorithm 5 details the NMS verification process.

Next we show how to construct the abstract box $\overline{box}$ and how to calculate the lower and upper bounds of IoU.

Let $\mathbb{B}$ be the 'box space' (space of individual box structures). An interpretation function $G : \mathbb{R}^{d_L} \to \{\mathcal{S} \subseteq \mathbb{B} \mid |\mathcal{S}| = n_x\}$ maps $\boldsymbol{y}$ to a set $\{box_i\}_{i=1}^{n_x}$ of $n_x$ bounding boxes, where $n_x$ is a constant determined by the fixed input dimension. A function $S : \mathbb{B} \to \mathcal{P}(\{1, \ldots, d_L\})$ (where $\mathcal{P}$ is the power set) maps each distinct $box_k \in \mathbb{B}$ (that could form part of an output set) to its source indices in $\boldsymbol{y}$. We then define the regular region $\mathcal{Z}$ as follows:

**Definition 5** (Regular Region). A regular region is a subset $\mathcal{Z} \subseteq \mathbb{R}^{d_L}$ defined as $\mathcal{Z} = \mathcal{H} \setminus \mathcal{S}$, where:

- $\mathcal{H}$ is an axis-aligned hyperrectangle centered at $F(\boldsymbol{x}^{(0)})$ with component-wise perturbations bounded by $c_1 \boldsymbol{v}_{\max}$:

$$\mathcal{H} = \left\{ F(\boldsymbol{x}^{(0)}) + \boldsymbol{\epsilon} \in \mathbb{R}^{d_L} \mid \forall j \in \{1, \ldots, d_L\}, \; |\epsilon_j| \leq (c_1 \boldsymbol{v}_{\max})_j \right\}$$

---

**Algorithm 5** Sound Object-Disappearance Threat Verification for NMS

---

**Require:** Constraints $\mathcal{Z} = \mathcal{H} \setminus \mathcal{S}$; ground truth $box_{\text{gt}}$; confidence threshold $\iota$; IoU threshold $\tau$; IoU suppression threshold $\eta$.
**Ensure:** Calculate $\tau_1, \tau_2$.
1: $\mathcal{B}_{\text{cand}}, \mathcal{B}_{\text{other}} \leftarrow \emptyset, \emptyset$
2: $\{\underline{\overline{box}}_i\}_{i=1}^{n_x} = \text{CONSTRUCT\_ABSTRACT\_BOX}(\mathcal{Z})$
3: **for all** $\underline{\overline{box}}_i \in \{\underline{\overline{box}}_k\}_{k=1}^{n_x}$ **do**
4: $\quad \tau_1(i) \leftarrow 0, \tau_2(i) \leftarrow 1$
5: $\quad$ **if** $\forall k \in [n_{\text{cls}}] \setminus \{\text{Class}(box_{\text{gt}})\}, \overline{p}_{i,\text{Class}(box_{\text{gt}})} \leq \underline{p}_{i,k}$ **then**
6: $\quad\quad$ **continue** $\qquad\qquad\qquad\qquad\qquad$ $\triangleright$ Skip boxes that cannot match $box_{\text{gt}}$ class
7: $\quad$ **end if**
8: $\quad$ **if** $\underline{c}_i \geq \iota$ **then** $\qquad\qquad\qquad\qquad\qquad$ $\triangleright$ Ensure indicator $\mathbb{I}(c_i \geq \iota) = 1$
9: $\quad\quad \tau_1(i) \leftarrow \text{IOU\_LOWER\_BOUNDS}(\underline{\overline{box}}_i, box_{\text{gt}})$
10: $\quad\quad$ **if** $\forall k \in [n_{\text{cls}}] \setminus \{\text{Class}(box_{\text{gt}})\}, \underline{p}_{i,\text{Class}(box_{\text{gt}})} \geq \overline{p}_{i,k}$ **then**
11: $\quad\quad\quad \mathcal{B}_{\text{cand}} \leftarrow \mathcal{B}_{\text{cand}} \cup \{\underline{\overline{box}}_i\}$ $\qquad\qquad$ $\triangleright$ Box must match $box_{\text{gt}}$ class
12: $\quad\quad$ **else**
13: $\quad\quad\quad \mathcal{B}_{\text{other}} \leftarrow \mathcal{B}_{\text{other}} \cup \{\underline{\overline{box}}_i\}$ $\qquad\qquad$ $\triangleright$ Box may match $box_{\text{gt}}$ class
14: $\quad\quad$ **end if**
15: $\quad$ **end if**
16: **end for**
17: **for all** $\underline{\overline{box}}_i \in \mathcal{B}_{\text{cand}}$ **do**
18: $\quad$ **for all** $\underline{\overline{box}}_j \in \mathcal{B}_{\text{other}}$ **do**
19: $\quad\quad$ **if** $\overline{c}_j < \underline{c}_i$ **then** $\qquad\qquad\qquad\qquad$ $\triangleright$ Ensure $box_j$ may suppress $box_i$
20: $\quad\quad\quad$ **continue** $\qquad\qquad\qquad\qquad$ $\triangleright$ Skip boxes that cannot suppress $box_i$
21: $\quad\quad$ **end if**
22: $\quad\quad ub \leftarrow \text{IOU\_UPPER\_BOUNDS}(\underline{\overline{box}}_i, \underline{\overline{box}}_j)$
23: $\quad\quad$ **if** $ub \geq \eta$ **then** $\qquad\qquad\qquad\qquad$ $\triangleright$ Ensure $box_i$ may be suppressed by $box_j$
24: $\quad\quad\quad lb \leftarrow \text{IOU\_LOWER\_BOUNDS}(\underline{\overline{box}}_j, box_{\text{gt}})$
25: $\quad\quad\quad \tau_2(i) \leftarrow \min(\tau_2(i), lb)$
26: $\quad\quad$ **end if**
27: $\quad$ **end for**
28: **end for**
29: **return** $\{\tau_1(i)\}_{i=1}^{n_x}, \{\tau_2(i)\}_{i=1}^{n_x}$

---

- $\mathcal{S}$ is a union of $k$ hyperspherical zones. Each zone $\mathcal{S}_i = \text{B}(\boldsymbol{c}_i, d_i)$ is defined by a center $\boldsymbol{c}_i$ and radius $d_i$, where $\boldsymbol{c}_i$ is the center of the exclusion zone and $d_i$ is the radius. The union of these zones is given by:

$$\mathcal{S} = \bigcup_{i=1}^k \mathcal{S}_i = \bigcup_{i=1}^k \text{B}(\boldsymbol{c}_i, d_i), \text{B}(\boldsymbol{c}_i, d_i) = \left\{ \boldsymbol{y} \in \mathbb{R}^{d_L} \mid \|\boldsymbol{y} - \boldsymbol{c}_i\|_2^2 \leq d_i^2 \right\}$$

Note, sometimes we just need a subset of dimensions of $\mathcal{S}_i$. Thus we can extend B to $\text{B}(\boldsymbol{c}_i, d_i, \mathcal{I}_i)$, where $\mathcal{I}_i$ is the index set of dimensions in $\mathcal{S}_i$. Then we have:

$$\text{B}(\boldsymbol{c}_i, d_i, \mathcal{I}_i) = \left\{ \boldsymbol{y} \in \mathbb{R}^{d_L} \mid \|\boldsymbol{y}_{\mathcal{I}_i} - \boldsymbol{c}_{i,\mathcal{I}_i}\|_2^2 \leq d_i^2 \right\}$$

where $\boldsymbol{c}_{i,\mathcal{I}_i}$ is the component of $\boldsymbol{c}_i$ for index set $\mathcal{I}_i$.

## I.1 ABSTRACT BOUNDING BOX CONSTRUCTION AND IOU BOUND COMPUTATION

We use $\tilde{x}$ to represent that $x$ is a Gurobi variable, $\tilde{\mathbb{R}}$ to represent real number Gurobi variable space. Let $\tilde{\boldsymbol{y}} \in \tilde{\mathbb{R}}^{d_L}$ be the Gurobi variable vector.

Then we encode the regular region $\mathcal{Z}$ as a set of constraints. To encode $\mathcal{H}$, we need to obtain the lower and upper bounds of each parameter of each bounding box. Suppose $\mathcal{H} = \left\{ \text{F}(\boldsymbol{x}^{(0)}) + \boldsymbol{\epsilon} \in \mathbb{R}^{d_L} \mid \forall j \in \{1, \ldots, d_L\}, |\epsilon_j| \leq (c_1 \boldsymbol{v}_{\max})_j \right\}$, where $c_1$ and $\boldsymbol{v}_{\max}$ (Note $\boldsymbol{v}_{\max} > 0$)

are obtained from Part 1. Let $\overline{\boldsymbol{y}} = \mathrm{F}(\boldsymbol{x}^{(0)}) + c_1 \boldsymbol{v}_{\max}$ and $\underline{\boldsymbol{y}} = \mathrm{F}(\boldsymbol{x}^{(0)}) - c_1 \boldsymbol{v}_{\max}$. Then we add the following constraints to the Gurobi model:

$$\underline{\boldsymbol{y}} \le \tilde{\boldsymbol{y}} \le \overline{\boldsymbol{y}}$$

Here, $\le$ is the component-wise less than or equal to operator.

Next, we need to add the exclusion zone constraints. For each exclusion zone $\mathcal{S}_i$, we need to add the following constraints:

$$\sum_{j \in \mathcal{I}_i} (\tilde{\boldsymbol{y}}_j - (\boldsymbol{c}_i)_j)^2 \ge d_i^2$$

where $(\boldsymbol{c}_i)_j$ is the component of $\boldsymbol{c}_i$ for index $j$.

CONSTRUCT_ABSTRACT_BOX($\mathcal{Z}$) adds these constraints to the Gurobi model, then reorganizes $\tilde{\boldsymbol{y}}$ as an abstract bounding box set $\{\overline{box}_i\}_{i=1}^{n_{\boldsymbol{x}}}$, where $n_{\boldsymbol{x}}$ is the number of bounding boxes. Each bounding box $\overline{box}_i = (\tilde{x}_i, \tilde{y}_i, \tilde{w}_i, \tilde{h}_i, \tilde{c}_i, \{\tilde{p}_{i,j}\}_{j=1}^{n_{\mathrm{cls}}})$, where $n_{\mathrm{cls}}$ is the number of classes. Note all the parameters of $\overline{box}_i$ are from $\tilde{\boldsymbol{y}}$ thus have constraints on them.

IoU bounds for abstract boxes involve:

1. **Geometric Constraints**: Box $i$ coordinates are:

$$\tilde{x}^i_{\min} = \tilde{x}_i - \frac{\tilde{w}_i}{2}, \quad \tilde{x}^i_{\max} = \tilde{x}_i + \frac{\tilde{w}_i}{2},$$
$$\tilde{y}^i_{\min} = \tilde{y}_i - \frac{\tilde{h}_i}{2}, \quad \tilde{y}^i_{\max} = \tilde{y}_i + \frac{\tilde{h}_i}{2}.$$

2. **Intersection/Union Area**

$$I_w = \max(0, \min(\tilde{x}^i_{\max}, \tilde{x}^j_{\max}) - \max(\tilde{x}^i_{\min}, \tilde{x}^j_{\min})),$$
$$I_h = \max(0, \min(\tilde{y}^i_{\max}, \tilde{y}^j_{\max}) - \max(\tilde{y}^i_{\min}, \tilde{y}^j_{\min})),$$
$$A_{\mathrm{int}} = I_w \cdot I_h,$$
$$A_{\mathrm{union}} = (\tilde{x}^i_{\max} - \tilde{x}^i_{\min}) \cdot (\tilde{y}^i_{\max} - \tilde{y}^i_{\min}) + (\tilde{x}^j_{\max} - \tilde{x}^j_{\min}) \cdot (\tilde{y}^j_{\max} - \tilde{y}^j_{\min}) - A_{\mathrm{int}}.$$

We use big-M constraints to encode $\max(0, \cdot)$ and $\min(\cdot, \cdot)$ operations in Gurobi. For example, to encode $\tilde{A}_{\mathrm{expr}} = \max(0, E)$, where $E$ is an expression, we introduce an auxiliary binary variable $\tilde{b} \in \{0, 1\}$ and a sufficiently large constant $M$. The variable $\tilde{A}_{\mathrm{expr}}$ is then constrained by:

$$\tilde{A}_{\mathrm{expr}} \ge E$$
$$\tilde{A}_{\mathrm{expr}} \ge 0$$
$$\tilde{A}_{\mathrm{expr}} \le E + M \cdot \tilde{b}$$
$$\tilde{A}_{\mathrm{expr}} \le M \cdot (1 - \tilde{b})$$

where $\tilde{A}_{\mathrm{expr}}$ is a Gurobi variable representing the maximum. The inner terms of $I_w$ and $I_h$, such as $\min(\tilde{x}^i_{\max}, \tilde{x}^j_{\max})$, are handled similarly using appropriate big-M formulations or Gurobi's built-in functions. The constant $M$ must be chosen such that $M \ge \max(U_E, -L_E)$, where $U_E$ and $L_E$ are known upper and lower bounds for the expression $E$.

- If $E \ge 0$, the objective of minimizing $\tilde{A}_{\mathrm{int}}$ (or other constraints) will force $\tilde{b} = 0$. The constraints become $\tilde{A}_{\mathrm{int}} \ge E$, $\tilde{A}_{\mathrm{int}} \ge 0$, $\tilde{A}_{\mathrm{int}} \le E$, and $\tilde{A}_{\mathrm{int}} \le M$. This correctly sets $\tilde{A}_{\mathrm{int}} = E$.
- If $E < 0$, the objective will force $\tilde{b} = 1$. The constraints become $\tilde{A}_{\mathrm{int}} \ge E$, $\tilde{A}_{\mathrm{int}} \ge 0$, $\tilde{A}_{\mathrm{int}} \le E + M$, and $\tilde{A}_{\mathrm{int}} \le 0$. This correctly sets $\tilde{A}_{\mathrm{int}} = 0$.

3. **Binary Search for IoU bounds**: To find $\underline{\mathrm{IoU}}$ (lower bound of IoU):

1: $\tau_{lb} \leftarrow 0, \tau_{ub} \leftarrow 1, \epsilon_{\mathrm{search}} \leftarrow 10^{-5}$
2: **while** $|\tau_{ub} - \tau_{lb}| > \epsilon_{\mathrm{search}}$ **do**

3:  $\tau_{\text{mid}} \leftarrow (\tau_{lb} + \tau_{ub})/2$
4:  Solve $L_{\text{check}} = \min(A_{\text{int}} - \tau_{\text{mid}} A_{\text{union}})$ by Gurobi
5:  **if** $L_{\text{check}} \geq 0$ **then** $\tau_{lb} \leftarrow \tau_{\text{mid}}$
6:  **else**$\tau_{ub} \leftarrow \tau_{\text{mid}}$
7:  **end if**
8: **end while**
9: **return** $\tau_{lb}$

Checking IoU bounds against a value $\tau$ is often done via optimizing $A_{\text{int}} - \tau A_{\text{union}}$, as direct interval ratio optimization $A_{\text{int}}/A_{\text{union}}$ is complex for solvers (e.g., Gurobi) without reformulation. For instance, to check if $\text{IoU}(box_i, box_{\text{gt}}) \geq \tau$ is possible, one can check if $\max(A_{\text{int}} - \tau A_{\text{union}}) \geq 0$.

It is straightforward to extend the above algorithm to find $\tau_{ub}$ (the upper bound of IoU).

The Gurobi solver, when minimizing or maximizing an objective subject to these constraints, returns an assignment for $\tilde{\boldsymbol{y}}$ (which corresponds to a specific point $\boldsymbol{z} \in \mathcal{Z}$). If a particular optimization (e.g., a step in the binary search showing $L_{\text{check}} < 0$) demonstrates a property violation, the returned $\tilde{\boldsymbol{y}}$ is the concrete instance $\boldsymbol{z}$ that exhibits this violation.

## J THE PROOF OF THEOREM 1

**Theorem** (probabilistic guarantee for Part 3). *Consider Algorithm 4. Here $N$ is the number of outer samples $\{\boldsymbol{x}^{(i)}\}_{i=1}^{N}$ in Step One, $M$ is the number of samples $\{\boldsymbol{x}^{(i,j)}\}_{j=1}^{M}$ drawn for each $\boldsymbol{x}^{(i)}$, and $M_2$ is the number of samples used in Step Two.*
*For any $\alpha, \beta, \epsilon \in (0, 1)$ satisfying $\epsilon < \frac{1}{2}$ and $N \cdot ((1 - 2\epsilon)^M) > \frac{1}{\alpha} \ln(\frac{1}{\beta})$, with Alg. 4, for any $\boldsymbol{y}$, with probability at least $1 - \beta - 2(1 - \epsilon)^{M_2}$ over steps one and two, we have $\text{V}(\boldsymbol{y}, d_{\min}) \leq \alpha$.*

*Proof.* For any given $\boldsymbol{y}' = \text{F}(\boldsymbol{x}')$, let $A_{\boldsymbol{y}'}$ satisfy that $\mathbb{P}_{\boldsymbol{x} \sim \mathcal{C}}(\|\text{F}(\boldsymbol{x}) - \boldsymbol{y}'\|_2 \leq A_{\boldsymbol{y}'}) = 1 - \epsilon$, let $B_{\boldsymbol{y}'}$ satisfy that $\mathbb{P}_{\boldsymbol{x} \sim \mathcal{C}}(\|\text{F}(\boldsymbol{x}) - \boldsymbol{y}'\|_2 \geq B_{\boldsymbol{y}'}) = 1 - \epsilon$. Let $S_{\boldsymbol{y}'} = \frac{B_{\boldsymbol{y}'}}{A_{\boldsymbol{y}'} - B_{\boldsymbol{y}'}}$.

For convenience, we denote Step 1 in Algorithm 4 by **P1**, and Step 2 by **P2**.

**Part One: For any $i$, with probability** $(1 - 2\epsilon)^M$ **of P1, we have** $\|\text{F}(\boldsymbol{x}^{(i,j)}) - \text{F}(\boldsymbol{x}^{(i)})\|_2 \in [B_{\text{F}(\boldsymbol{x}^{(i)})}, A_{\text{F}(\boldsymbol{x}^{(i)})}]$ **for all** $j \in [M]$**.**

Based on the definition of $B_{\text{F}(\boldsymbol{x}^{(i)})}$ and $A_{\text{F}(\boldsymbol{x}^{(i)})}$, we know that:

$$\mathbb{P}_{\boldsymbol{x}^{(i,j)} \sim \mathcal{C}}(\|\text{F}(\boldsymbol{x}^{(i,j)}) - \text{F}(\boldsymbol{x}^{(i)})\|_2 \in [B_{\text{F}(\boldsymbol{x}^{(i)})}, A_{\text{F}(\boldsymbol{x}^{(i)})}]) \geq 1 - 2\epsilon \tag{6}$$

by union bound. Because each selection of $\boldsymbol{x}^{(i,j)}$ is independent, we have:

$$\mathbb{P}(\forall j \in [M], \|\text{F}(\boldsymbol{x}^{(i,j)}) - \text{F}(\boldsymbol{x}^{(i)})\|_2 \in [B_{\text{F}(\boldsymbol{x}^{(i)})}, A_{\text{F}(\boldsymbol{x}^{(i)})}]) \geq (1 - 2\epsilon)^M. \tag{7}$$

**Part Two: With probability** $1 - \beta$ **of P1, we have** $\mathbb{P}_{\boldsymbol{x} \sim \mathcal{C}}(S_{\text{F}(\boldsymbol{x})} \leq C) \geq 1 - \alpha$**.**

Firstly, if $i \in [N]$ such that $\|\text{F}(\boldsymbol{x}^{(i,j)}) - \text{F}(\boldsymbol{x}^{(i)})\|_2 \in [B_{\text{F}(\boldsymbol{x}^{(i)})}, A_{\text{F}(\boldsymbol{x}^{(i)})}]$ for all $j \in [M]$, then there must be:

$$C \geq \frac{B'_i}{A'_i - B'_i} \geq S_{\text{F}(\boldsymbol{x}^{(i)})}. \tag{8}$$

Here, $B'_i = \min_{j \in [M]} \|\text{F}(\boldsymbol{x}^{(i,j)}) - \text{F}(\boldsymbol{x}^{(i)})\|_2$ and $A'_i = \max_{j \in [M]} \|\text{F}(\boldsymbol{x}^{(i,j)}) - \text{F}(\boldsymbol{x}^{(i)})\|_2$.

Let $p$ satisfy that $\mathbb{P}_{\boldsymbol{x} \sim \mathcal{C}}(S_{\text{F}(\boldsymbol{x})} \leq p) = 1 - \alpha$, then we have $\mathbb{P}_{\boldsymbol{x} \sim \mathcal{C}}(S_{\text{F}(\boldsymbol{x})} \geq p) \geq \alpha$.

Then, we define the witness event $W_i$ as: $W_i = \{S_{\text{F}(\boldsymbol{x}^{(i)})} \geq p\} \cap \{\|\text{F}(\boldsymbol{x}^{(i,j)}) - \text{F}(\boldsymbol{x}^{(i)})\|_2 \in [B_{\text{F}(\boldsymbol{x}^{(i)})}, A_{\text{F}(\boldsymbol{x}^{(i)})}]$ for all $j \in [M]\}$.

Using equation 7, we have that:

$$
\begin{aligned}
\mathbb{P}(W_i) &= \mathbb{E}\left[\mathbf{1}_{S_{\mathrm{F}(\boldsymbol{x}^{(i)})} \geq p} \cdot \mathbb{P}(\|\mathrm{F}(\boldsymbol{x}^{(i,j)}) - \mathrm{F}(\boldsymbol{x}^{(i)})\|_2 \in [B_{\mathrm{F}(\boldsymbol{x}^{(i)})}, A_{\mathrm{F}(\boldsymbol{x}^{(i)})}] \text{ for all } j \in [M] \mid \boldsymbol{x}^{(i)})\right] \\
&\geq \mathbb{E}[\mathbf{1}_{S_{\mathrm{F}(\boldsymbol{x}^{(i)})} \geq p} \cdot (1 - 2\epsilon)^M] \\
&= \alpha \cdot (1 - 2\epsilon)^M.
\end{aligned}
$$

$$(9)$$

Since each sample $\boldsymbol{x}^{(i)}$ is independent, thus each event $W_i$ is independent, then we have that:

$$
\mathbb{P}(\forall i \in [N], W_i \text{ does not occur}) \leq (1 - \alpha \cdot (1 - 2\epsilon)^M)^N \leq e^{-\alpha \cdot (1-2\epsilon)^M \cdot N}. \tag{10}
$$

From the theorem's sample-size condition, we have $N \cdot ((1-2\epsilon)^M) > \frac{1}{\alpha}\ln(\frac{1}{\beta})$, hence $N(1-2\epsilon)^M > \frac{1}{\alpha}\ln(\frac{1}{\beta})$, so

$$
\alpha \cdot (1 - 2\epsilon)^M \cdot N > \ln\left(\frac{1}{\beta}\right) \implies e^{-\alpha \cdot (1-2\epsilon)^M \cdot N} < \beta.
$$

This means that with probability at least $1 - \beta$, there exists $i$ such that $W_i$ occurs, which means we have $\|\mathrm{F}(\boldsymbol{x}^{(i,j)}) - \mathrm{F}(\boldsymbol{x}^{(i)})\|_2 \in [B_{\mathrm{F}(\boldsymbol{x}^{(i)})}, A_{\mathrm{F}(\boldsymbol{x}^{(i)})}]$ for all $j \in [M]$ and $S_{\mathrm{F}(\boldsymbol{x}^{(i)})} \geq p$. Fix such an index $i$, from equation 8, we have

$$
C = \max_{k \in [N]} \frac{B'_k}{A'_k - B'_k} \geq \frac{B'_i}{A'_i - B'_i} \geq S_{\mathrm{F}(\boldsymbol{x}^{(i)})} = \frac{B_{\mathrm{F}(\boldsymbol{x}^{(i)})}}{A_{\mathrm{F}(\boldsymbol{x}^{(i)})} - B_{\mathrm{F}(\boldsymbol{x}^{(i)})}} \geq p.
$$

So, with probability at least $1 - \beta$, we obtain $C \geq p$, according to the definition of $p$, we have:

$$
\mathbb{P}_{\boldsymbol{x} \sim \mathcal{C}}(S_{\mathrm{F}(\boldsymbol{x})} \leq C) \geq \mathbb{P}_{\boldsymbol{x} \sim \mathcal{C}}(S_{\mathrm{F}(\boldsymbol{x})} \leq p) = 1 - \alpha. \tag{11}
$$

**Part Three: For any $y$, with probability at least $1 - \beta - 2(1 - \epsilon)^{M_2}$ over steps one and two, we have $\mathrm{V}(\boldsymbol{y}, d_{\min}) \leq \alpha$.**

Let $\mathcal{Q} = \{\boldsymbol{x} \in \mathcal{C} : S_{\mathrm{F}(\boldsymbol{x})} \leq p\}$. By the definition of $p$, we have $\mathbb{P}_{\boldsymbol{x} \sim \mathcal{C}}(\boldsymbol{x} \in \mathcal{Q}) \geq 1 - \alpha$.

Consider the $M_2$ i.i.d. samples $\{\boldsymbol{x}^{(i)}\}_{i=1}^{M_2} \sim \mathcal{C}$ in Step Two of the algorithm. Fix any $\boldsymbol{x}' \in \mathcal{Q}$, by the definition of $B_{\mathrm{F}(\boldsymbol{x}')}$ and $A_{\mathrm{F}(\boldsymbol{x}')}$, we have $\mathbb{P}_{\boldsymbol{x}^{(i)} \sim \mathcal{C}}(\|\mathrm{F}(\boldsymbol{x}^{(i)}) - \mathrm{F}(\boldsymbol{x}')\|_2 \leq B_{\mathrm{F}(\boldsymbol{x}')}) \geq \epsilon$, so the probability that *none* of the $M_2$ samples satisfies this is at most $(1 - \epsilon)^{M_2}$. Similarly, each draw satisfies $\mathbb{P}_{\boldsymbol{x}^{(i)} \sim \mathcal{C}}(\|\mathrm{F}(\boldsymbol{x}^{(i)}) - \mathrm{F}(\boldsymbol{x}')\|_2 \geq A_{\mathrm{F}(\boldsymbol{x}')}) \geq \epsilon$, so the probability that none satisfies it is at most $(1 - \epsilon)^{M_2}$. By a union bound, with probability at least $1 - 2(1 - \epsilon)^{M_2}$, there exist:

$$
\begin{aligned}
\hat{B}(\boldsymbol{x}') &= \min_{i \in [M_2]} \|\mathrm{F}(\boldsymbol{x}') - \mathrm{F}(\boldsymbol{x}^{(i)})\|_2 \leq B_{\mathrm{F}(\boldsymbol{x}')}, \\
\hat{A}(\boldsymbol{x}') &= \max_{i \in [M_2]} \|\mathrm{F}(\boldsymbol{x}') - \mathrm{F}(\boldsymbol{x}^{(i)})\|_2 \geq A_{\mathrm{F}(\boldsymbol{x}')}.
\end{aligned}
\tag{12}
$$

Then for a given $\boldsymbol{y}$, let

$$
\boldsymbol{x}^{(q)} = \arg\min_{\boldsymbol{x} \in \mathcal{Q}} \|\boldsymbol{y} - \mathrm{F}(\boldsymbol{x})\|_2, \quad t = \|\boldsymbol{y} - \mathrm{F}(\boldsymbol{x}^{(q)})\|_2. \tag{13}
$$

Then, we have $A_m = \max_{i \in [M_2]} \|\boldsymbol{y} - \mathrm{F}(\boldsymbol{x}^{(i)})\|_2 \geq \max_{i \in [M_2]} \|\mathrm{F}(\boldsymbol{x}^{(q)}) - \mathrm{F}(\boldsymbol{x}^{(i)})\|_2 - \|\boldsymbol{y} - \mathrm{F}(\boldsymbol{x}^{(q)})\|_2 = \hat{A}(\boldsymbol{x}^{(q)}) - t$, similar, we have $B_m \leq \hat{B}(\boldsymbol{x}^{(q)}) + t$. So, we have $B_m - C(A_m - B_m) \leq \hat{B}(\boldsymbol{x}^{(q)}) - C(\hat{A}(\boldsymbol{x}^{(q)}) - \hat{B}(\boldsymbol{x}^{(q)})) + t(1 + 2C)$.

As shown in the Part Two, with probability $1 - \beta$ of P1, there is $C \geq p$, and then, using $\boldsymbol{x}^{(q)} \in \mathcal{Q}$, we have $\mathbb{P}_{\mathrm{P2}}(\hat{B}(\boldsymbol{x}^{(q)}) - C(\hat{A}(\boldsymbol{x}^{(q)}) - \hat{B}(\boldsymbol{x}^{(q)})) \leq 0) \geq 1 - 2(1 - \epsilon)^{M_2}$, which can imply that $\mathbb{P}_{\mathrm{P2}}(B_m - C(A_m - B_m) \leq t(1 + 2C)) \geq 1 - 2(1 - \epsilon)^{M_2}$ by the above result.

And when $B_m - C(A_m - B_m) \leq t(1 + 2C)$ holds, we have $t \geq \max(\frac{B_m - C(A_m - B_m)}{(1+2C)}, 0) = d_{\min}$. Because $\mathcal{B}_2(\boldsymbol{y}, r)$ is an *open* ball, hence $\mathcal{B}_2(\boldsymbol{y}, d_{\min}) \cap \mathrm{F}(\mathcal{Q}) = \emptyset$ by the definition of $t$, which implies that:

$$
\mathrm{V}(\boldsymbol{y}, d_{\min}) = \mathbb{P}_{\boldsymbol{x} \sim \mathcal{C}}\big(\mathrm{F}(\boldsymbol{x}) \in \mathcal{B}_2(\boldsymbol{y}, d_{\min})\big) \leq \mathbb{P}_{\boldsymbol{x} \sim \mathcal{C}}(\boldsymbol{x} \notin \mathcal{Q}) \leq \alpha. \tag{14}
$$

**Putting probabilities together.** From equation 11, we have $1 - \alpha$ percent of $\boldsymbol{x} \in \mathcal{C}$ such that $S_{\mathrm{F}(\boldsymbol{x})} \leq C$. For a fixed $\boldsymbol{x}^{(q)}$ chosen above, the algorithm succeeds with probability at least $1 - 2(1 - \epsilon)^{M_2}$. A union bound gives

$$\mathbb{P}\big(\mathrm{V}(\boldsymbol{y}, d_{\min}) \leq \alpha\big) \geq 1 - \beta - 2(1 - \epsilon)^{M_2}.$$

This completes the proof. $\qquad\square$

## K   PART THREE FOR SMALL NETWORKS

To demonstrate the superiority of our method in Section 4 and compare it with existing approaches, we show how we can apply Part Three on small-scale object detection based on the feed-forward ReLU neural networks. In this case, the problem in Part Three can be formulated as follows:

$$
\begin{aligned}
\min_{\boldsymbol{x}, \hat{\boldsymbol{x}}} \quad & \|\boldsymbol{y} - \boldsymbol{x}^{(L)}\|_2 \\
\text{s.t. (i)} \quad & \boldsymbol{x} \in \mathcal{C}, \quad \hat{\boldsymbol{x}}^{(0)} = \boldsymbol{x}; \\
\text{(ii)} \quad & \boldsymbol{x}^{(i)} = \mathbf{W}^{(i)} \hat{\boldsymbol{x}}^{(i-1)} + \mathbf{b}^{(i)}, \quad i \in [L]; \\
\text{(iii)} \quad & \hat{\boldsymbol{x}}^{(i)} = \max\big(0, \boldsymbol{x}^{(i)}\big), \quad i \in [L-1]
\end{aligned}
$$

where $\mathbf{W}^{(i)}$ is the weight matrix of $i$-th layer in the network, $\mathbf{b}^{(i)}$ is the bias vector of $i$-th layer in the network. Unfortunately, finding an exact solution to this problem is NP-complete (Katz et al., 2017). So, we consider bounds on the outputs of hidden-layers: $l_j^{(i)} \leq x_j^{(i)} \leq u_j^{(i)}$. Based on these bounds, let $\mathcal{I}^{-(i)} = \{j : u_j^{(i)} \leq 0\}$, $\mathcal{I}^{+(i)} = \{j : l_j^{(i)} \geq 0\}$ and $\mathcal{I}^{\pm(i)} = \{j : l_j^{(i)} < 0 < u_j^{(i)}\}$, the condition (iii) can be replaced by the following constraints: $\hat{x}_j^{(i)} \geq 0, \hat{x}_j^{(i)} \geq x_j^{(i)}, (u_j^{(i)} - l_j^{(i)})\hat{x}_j^{(i)} \leq u_j^{(i)} x_j^{(i)} - l_j^{(i)} u_j^{(i)}, j \in \mathcal{I}^{\pm(i)}; \hat{x}_j^{(i)} = x_j^{(i)}, j \in \mathcal{I}^{+(i)}; \hat{x}_j^{(i)} = 0, j \in \mathcal{I}^{-(i)}$. To solve this problem, we prove that such minimum value is equal to the maximum value of the square root of $d(\boldsymbol{\alpha}, \boldsymbol{\nu})$ which is defined in the following theorem. So, we can use the gradient to estimate the maximum of $d(\boldsymbol{\alpha}, \boldsymbol{\nu})$, and use it to be the minimum distance in Part Three.

**Theorem 3.** *Under the setting in this section, we have that:*

$$\min_{\boldsymbol{x} \in \mathcal{C}} ||\mathrm{F}(\boldsymbol{x}) - \boldsymbol{y}||_2 = \max_{\boldsymbol{\alpha} \in [0,1]} \sqrt{d(\boldsymbol{\alpha}, \boldsymbol{\nu})}$$

*, which is defined as below:*

$$
\begin{aligned}
d(\boldsymbol{\alpha}, \boldsymbol{\nu}) = & \boldsymbol{y}^\top \boldsymbol{\nu}^{(L)} - \frac{\boldsymbol{\nu}^{(L)\top} \boldsymbol{\nu}^{(L)}}{4} - [\boldsymbol{\nu}^{(1)\top} \mathbf{W}^{(1)}]_+ \boldsymbol{u}^{(0)} + [\boldsymbol{\nu}^{(1)\top} \mathbf{W}^{(1)}]_- \boldsymbol{l}^{(0)} \\
& - \sum_{i=1}^{L} \boldsymbol{\nu}^{(i)\top} \boldsymbol{b}^{(i)} + \sum_{i=1}^{L-1} \sum_{j \in \mathcal{I}^{\pm(i)}} \left[ \frac{u_j^{(i)} l_j^{(i)} [\hat{\nu}_j^{(i)}]_+}{u_j^{(i)} - l_j^{(i)}} \right]
\end{aligned}
\tag{15}
$$

*where*

$$\hat{\nu}_j^{(i)} = \boldsymbol{\nu}^{(i+1)\top} \mathbf{W}_{:,j}^{(i+1)}, \hat{\nu}_j^{(i)} = [\hat{\nu}_j^{(i)}]_+ - [\hat{\nu}_j^{(i)}]_-, \forall i \in \{1, 2, ..., L-1\}$$

$$
\nu_j^{(i)} = 
\begin{cases}
\hat{\nu}_j^{(i)} & , j \in \mathcal{I}^{+(i)} \\
0 & , j \in \mathcal{I}^{-(i)} \\
\frac{u_j^{(i)}}{u_j^{(i)} - l_j^{(i)}} [\hat{\nu}_j^{(i)}]_+ - \alpha_j^{(i)} [\hat{\nu}_j^{(i)}]_- & , j \in \mathcal{I}^{\pm(i)}
\end{cases}
\quad \forall i \in \{1, 2, ..., L-1\}
$$

## L   THE PROOF OF THEOREM 3

We mainly follow the proof idea in (Kotha et al., 2023).

*Proof.* Consider the $\mathcal{I}^{-(i)} = \{j : u_j^{(i)} \leq 0\}$, $\mathcal{I}^{+(i)} = \{j : l_j^{(i)} \geq 0\}$ and $\mathcal{I}^{\pm(i)} = \{j : l_j^{(i)} < 0 < u_j^{(i)}\}$, such optimization problem tends to the:

$$
\begin{aligned}
\min_{\boldsymbol{x},\hat{\boldsymbol{x}}} \quad & (\boldsymbol{x}^{(L)} - \boldsymbol{y})^\top (\boldsymbol{x}^{(L)} - \boldsymbol{y}) \\
\text{s.t.} \quad & \boldsymbol{l}^{(0)} \leq \hat{\boldsymbol{x}}^{(0)} \leq \boldsymbol{u}^{(0)} \\
& \boldsymbol{x}^{(i)} = \mathbf{W}^{(i)}\hat{\boldsymbol{x}}^{(i-1)} + \boldsymbol{b}^{(i)}; i = \{1, 2, 3, ..., L\} \\
& \hat{x}_j^{(i)} \geq 0;\ j \in \mathcal{I}^{\pm(i)} \\
& \hat{x}_j^{(i)} \geq x_j^{(i)};\ j \in \mathcal{I}^{\pm(i)} \\
& (u_j^{(i)} - l_j^{(i)})\hat{x}_j^{(i)} \leq u_j^{(i)} x_j^{(i)} - l_j^{(i)} u_j^{(i)},\ j \in \mathcal{I}^{\pm(i)} \\
& \hat{x}_j^{(i)} = x_j^{(i)},\ j \in \mathcal{I}^{+(i)} \\
& \hat{x}_j^{(i)} = 0,\ j \in \mathcal{I}^{-(i)}
\end{aligned} \tag{16}
$$

Then we use the Lagrange dual of the optimization problem to solve it, the Lagrange dual of the optimization problem is given by:

$$
\begin{aligned}
\min_{\boldsymbol{x},\hat{\boldsymbol{x}}} \max_{\boldsymbol{\lambda},\boldsymbol{\mu},\boldsymbol{\nu},\boldsymbol{\tau}} \quad & (\boldsymbol{x}^{(L)} - \boldsymbol{y})^\top (\boldsymbol{x}^{(L)} - \boldsymbol{y}) \\
& + \sum_{i=1}^{L} \boldsymbol{\nu}^{(i)\top} \left( \boldsymbol{x}^{(i)} - \mathbf{W}^{(i)}\hat{\boldsymbol{x}}^{(i-1)} - \boldsymbol{b}^{(i)} \right) \\
& + \sum_{i=1}^{L-1} \sum_{j \in \mathcal{I}^{\pm(i)}} \mu_j^{(i)}(-\hat{x}_j^{(i)}) \\
& + \sum_{i=1}^{L-1} \sum_{j \in \mathcal{I}^{\pm(i)}} \tau_j^{(i)}(x_j^{(i)} - \hat{x}_j^{(i)}) \\
& + \sum_{i=1}^{L-1} \sum_{j \in \mathcal{I}^{\pm(i)}} \left[ \lambda_j^{(i)} \left( (u_j^{(i)} - l_j^{(i)})\hat{x}_j^{(i)} - u_j^{(i)} x_j^{(i)} + l_j^{(i)} u_j^{(i)} \right) \right] \\
\text{s.t.} \quad & \boldsymbol{l}^{(0)} \leq \hat{\boldsymbol{x}}^{(0)} \leq \boldsymbol{u}^{(0)}; \\
& \hat{x}_j^{(i)} = 0, j \in \mathcal{I}^{-(i)}; \hat{x}_j^{(i)} = x_j^{(i)}, j \in \mathcal{I}^{+(i)} \\
& \boldsymbol{\mu} \geq 0;\ \boldsymbol{\tau} \geq 0;\ \boldsymbol{\lambda} \geq 0;
\end{aligned} \tag{17}
$$

According to its strong duality, the solution of the dual problem is the same as that of the primal problem. The dual problem is given by:

$$
\begin{aligned}
\max_{\boldsymbol{\lambda},\boldsymbol{\mu},\boldsymbol{\nu},\boldsymbol{\tau}} \min_{\boldsymbol{x},\hat{\boldsymbol{x}}} \quad & (\boldsymbol{x}^{(L)} - 2\boldsymbol{y} + \boldsymbol{\nu}^{(L)})^{\top}\boldsymbol{x}^{(L)} + \boldsymbol{y}^{\top}\boldsymbol{y} - (\boldsymbol{\nu}^{(1)})^{\top}\mathbf{W}^{(1)}\hat{x}^{(0)} \\
& + \sum_{i=1}^{L-1}\sum_{j\in\mathcal{I}^{+(i)}}(\nu_j^{(i)} - \boldsymbol{\nu}^{(i+1)\top}\mathbf{W}_{:,j}^{(i+1)})x_j^{(i)} \\
& + \sum_{i=1}^{L-1}\sum_{j\in\mathcal{I}^{-(i)}}\nu_j^{(i)}x_j^{(i)} \\
& + \sum_{i=1}^{L-1}\sum_{j\in\mathcal{I}^{\pm(i)}}\left[\left(\nu_j^{(i)} + \tau_j^{(i)} - \lambda_j^{(i)}u_j^{(i)}\right)x_j^{(i)} \right. \\
& \left. + (-\boldsymbol{\nu}^{(i+1)\top}\mathbf{W}_{:,j}^{(i+1)} - \mu_j^{(i)} - \tau_j^{(i)} + (u_j^{(i)} - l_j^{(i)})\lambda_j^{(i)})\hat{x}_j^{(i)}\right] \\
& - \sum_{i=1}^{L}\boldsymbol{\nu}^{(i)\top}\boldsymbol{b}^{(i)} + \sum_{i=1}^{L-1}\sum_{j\in\mathcal{I}^{\pm(i)}}\lambda_j^{(i)}u_j^{(i)}l_j^{(i)} \\
\text{s.t.} \quad & \boldsymbol{l}^{(0)} \le \hat{\boldsymbol{x}}^{(0)} \le \boldsymbol{u}^{(0)};\ \hat{x}_j^{(i)} = 0, j\in\mathcal{I}^{-(i)};\ \hat{x}_j^{(i)} = x_j^{(i)}, j\in\mathcal{I}^{+(i)} \\
& \boldsymbol{\mu} \ge 0;\ \boldsymbol{\tau} \ge 0;\ \boldsymbol{\lambda} \ge 0;
\end{aligned}
\tag{18}
$$

Here, we adjust the order of each term in the formula and directly incorporate the constraints $\hat{x}_j^{(i)} = 0, j\in\mathcal{I}^{-(i)}; \hat{x}_j^{(i)} = x_j^{(i)}, j\in\mathcal{I}^{+(i)}$ into the objective function. Then we can minimize $-(\boldsymbol{\nu}^{(1)})^{\top}\mathbf{W}^{(1)}\hat{x}^{(0)}$ subject to $\boldsymbol{l}^{(0)} \le \hat{\boldsymbol{x}}^{(0)} \le \boldsymbol{u}^{(0)}$ according to the each dimension of $\boldsymbol{\nu}^{(1)\top}\mathbf{W}^{(1)}$. If $(\boldsymbol{\nu}^{(1)\top}\mathbf{W}^{(1)})_j > 0$, then $\hat{x}_j^{(0)} = u_j^{(0)}$; Otherwise, $\hat{x}_j^{(0)} = l_j^{(0)}$. Since no additional constraints applied to $\boldsymbol{x}^{(L)}$, we can minimize $(\boldsymbol{x}^{(L)} - 2\boldsymbol{y} + \boldsymbol{\nu}^{(L)})^{\top}\boldsymbol{x}^{(L)}$ by setting $\boldsymbol{x}^{(L)} = \boldsymbol{y} - \frac{1}{2}\boldsymbol{\nu}^{(L)}$. Then we get:

$$
\begin{aligned}
\max_{\boldsymbol{\lambda},\boldsymbol{\mu},\boldsymbol{\nu},\boldsymbol{\tau}} \min_{\boldsymbol{x},\hat{\boldsymbol{x}}} \quad & \boldsymbol{y}^{\top}\boldsymbol{\nu}^{(L)} - \frac{\boldsymbol{\nu}^{(L)\top}\boldsymbol{\nu}^{(L)}}{4} - [\boldsymbol{\nu}^{(1)\top}\mathbf{W}^{(1)}]_+\boldsymbol{u}^{(0)} + [\boldsymbol{\nu}^{(1)\top}\mathbf{W}^{(1)}]_-\boldsymbol{l}^{(0)} \\
& + \sum_{i=1}^{L-1}\sum_{j\in\mathcal{I}^{+(i)}}(\nu_j^{(i)} - \boldsymbol{\nu}^{(i+1)\top}\mathbf{W}_{:,j}^{(i+1)})x_j^{(i)} \\
& + \sum_{i=1}^{L-1}\sum_{j\in\mathcal{I}^{-(i)}}\nu_j^{(i)}x_j^{(i)} \\
& + \sum_{i=1}^{L-1}\sum_{j\in\mathcal{I}^{\pm(i)}}\left[\left(\nu_j^{(i)} + \tau_j^{(i)} - \lambda_j^{(i)}u_j^{(i)}\right)x_j^{(i)} \right. \\
& \left. + \left(-\boldsymbol{\nu}^{(i+1)\top}\mathbf{W}_{:,j}^{(i+1)} - \mu_j^{(i)} - \tau_j^{(i)} + (u_j^{(i)} - l_j^{(i)})\lambda_j^{(i)}\right)\hat{x}_j^{(i)}\right] \\
& - \sum_{i=1}^{L}\boldsymbol{\nu}^{(i)\top}\boldsymbol{b}^{(i)} + \sum_{i=1}^{L-1}\sum_{j\in\mathcal{I}^{\pm(i)}}\lambda_j^{(i)}u_j^{(i)}l_j^{(i)} \\
\text{s.t.} \quad & \boldsymbol{\mu} \ge 0;\ \boldsymbol{\tau} \ge 0;\ \boldsymbol{\lambda} \ge 0;
\end{aligned}
\tag{19}
$$

Since variables $\boldsymbol{x}, \hat{\boldsymbol{x}}$ are unconstrained, any nonzero coefficients would drive their values to $-\infty$ in the inner minimization. Therefore, the outer maximization requires setting all coefficients of $\boldsymbol{x}, \hat{\boldsymbol{x}}$ to zero. Then we get:

$$\max_{\boldsymbol{\lambda},\boldsymbol{\mu},\boldsymbol{\tau},\boldsymbol{\nu}} \quad \boldsymbol{y}^\top \boldsymbol{\nu}^{(L)} - \frac{\boldsymbol{\nu}^{(L)\top} \boldsymbol{\nu}^{(L)}}{4}$$
$$- [\boldsymbol{\nu}^{(1)\top} \mathbf{W}^{(1)}]_+ \boldsymbol{u}^{(0)} + [\boldsymbol{\nu}^{(1)\top} \mathbf{W}^{(1)}]_- \boldsymbol{l}^{(0)}$$
$$- \sum_{i=1}^{L} \boldsymbol{\nu}^{(i)\top} \boldsymbol{b}^{(i)} + \sum_{i=1}^{L-1} \sum_{j \in \mathcal{I}^{\pm(i)}} \lambda_j^{(i)} u_j^{(i)} l_j^{(i)}$$

$$\text{s.t.} \quad \boldsymbol{\mu} \geq 0; \boldsymbol{\tau} \geq 0; \boldsymbol{\lambda} \geq 0; \tag{20}$$
$$\nu_j^{(i)} - \boldsymbol{\nu}^{(i+1)\top} \mathbf{W}_{:,j}^{(i+1)} = 0, j \in \mathcal{I}^{+(i)}$$
$$\nu_j^{(i)} = 0, j \in \mathcal{I}^{-(i)}$$
$$\nu_j^{(i)} + \tau_j^{(i)} - \lambda_j^{(i)} u_j^{(i)} = 0; j \in \mathcal{I}^{\pm(i)}$$
$$\boldsymbol{\nu}^{(i+1)\top} \mathbf{W}_{:,j}^{(i+1)} = (u_j^{(i)} - l_j^{(i)}) \lambda_j^{(i)} - (\mu_j^{(i)} + \tau_j^{(i)}), j \in \mathcal{I}^{\pm(i)}$$

Define $\hat{\nu}_j^{(i)} = \boldsymbol{\nu}^{(i+1)\top} \mathbf{W}_{:,j}^{(i+1)}$. Then, we define $(u_j^{(i)} - l_j^{(i)}) \lambda_j^{(i)} = [\hat{\nu}_j^{(i)}]_+$ and $\mu_j^{(i)} + \tau_j^{(i)} = [\hat{\nu}_j^{(i)}]_-$, we get the following bound propagation procedure:

$$\max_{\boldsymbol{\alpha},\boldsymbol{\nu}} \quad \boldsymbol{y}^\top \boldsymbol{\nu}^{(L)} - \frac{\boldsymbol{\nu}^{(L)\top} \boldsymbol{\nu}^{(L)}}{4} - [\boldsymbol{\nu}^{(1)\top} \mathbf{W}^{(1)}]_+ \boldsymbol{u}^{(0)} + [\boldsymbol{\nu}^{(1)\top} \mathbf{W}^{(1)}]_- \boldsymbol{l}^{(0)}$$
$$- \sum_{i=1}^{L} \boldsymbol{\nu}^{(i)\top} \boldsymbol{b}^{(i)} + \sum_{i=1}^{L-1} \sum_{j \in \mathcal{I}^{\pm(i)}} \left[ \frac{u_j^{(i)} l_j^{(i)} [\hat{\nu}_j^{(i)}]_+}{u_j^{(i)} - l_j^{(i)}} \right]$$

$$\text{s.t.} \quad \hat{\nu}_j^{(i)} = \boldsymbol{\nu}^{(i+1)\top} \mathbf{W}_{:,j}^{(i+1)}, \hat{\nu}_j^{(i)} = [\hat{\nu}_j^{(i)}]_+ - [\hat{\nu}_j^{(i)}]_-, \forall i \in \{1, 2, ..., L-1\} \tag{21}$$
$$\nu_j^{(i)} = \begin{cases} \hat{\nu}_j^{(i)} & , j \in \mathcal{I}^{+(i)} \\ 0 & , j \in \mathcal{I}^{-(i)} \\ \frac{u_j^{(i)}}{u_j^{(i)} - l_j^{(i)}} [\hat{\nu}_j^{(i)}]_+ - \alpha_j^{(i)} [\hat{\nu}_j^{(i)}]_- & , j \in \mathcal{I}^{\pm(i)} \end{cases} \quad \forall i \in \{1, 2, ..., L-1\}$$
$$0 \leq \alpha_j^{(i)} \leq 1, j \in \mathcal{I}^{\pm(i)}$$

Here $\alpha_j^{(i)}$ is the optimizable parameter controlling the relaxation of neuron $j$ in layer $i$ introduced in (Xu et al., 2021). This completes the proof. $\square$

## M  THE PROOF OF THEOREM 2

**Theorem.** *Using the notation and definitions from Algorithms 1–4. Given $\alpha, \beta, \epsilon \in (0, 1)$ satisfying $\epsilon < \frac{1}{2}$, $N \cdot ((1 - 2\epsilon)^M) > \frac{1}{\alpha} \ln(\frac{1}{\beta})$ and $N_2 \geq \lceil \frac{2\ln(1/\beta)}{\alpha} + 2 + \frac{2\ln(2/\alpha)}{\alpha} \rceil$. Then, after executing the algorithms defined above, for any $\kappa$ used in Algorithm 1, if these algorithms output "Safe" after at most $T$ refinement iterations, then with probability at least $1 - T(\beta + 2(1 - \epsilon)^{M_2}) - \beta$ over Algorithm 2 (Part 1) and Algorithm 4 (Part 3), we have $\mathbb{P}_{\boldsymbol{x} \sim \mathcal{C}}(\boldsymbol{x} \text{ is safe}) > 1 - (1 + T)\alpha$.*

*Proof.* We use the simple fact that $\mathbb{P}(A \cap B) \geq \mathbb{P}(A) + \mathbb{P}(B) - 1$ to obtain the following result:

From Theorem 1, after T refinement steps, the reduction in the probability of the system being certifiably safe due to these refinements is at most $T\alpha$. This statement holds with a confidence of at least $T\left(1 - (\beta + 2(1 - \epsilon)^{M_2})\right) - (T - 1) = 1 - T\left(\beta + 2(1 - \epsilon)^{M_2}\right)$.

According to Proposition 1, we obtain that the probability of the box being safe is $1 - T\alpha - \alpha$ with probability $1 - T(\beta + 2(1 - \epsilon)^{M_2}) + (1 - \beta) - 1 = 1 - T(\beta + 2(1 - \epsilon)^{M_2}) - \beta$.

Then we obtain $\mathbb{P}(\boldsymbol{x} \text{ is safe}) > 1 - (1 + T)\alpha$ with probability $1 - T(\beta + 2(1 - \epsilon)^{M_2}) - \beta$.

This yields the final result. $\square$

**Remark 6.** *Note we do not assume that Part 1 and Part 3 are independent, so we can reuse the samples in Part 1 to Part 3.*

## N    EXPERIMENT DETAILS

**Detailed Setting**: We use $N_1 = 30000$, $N_2 = 5000$, $N = 3000$, $M = 10$, $M_2 = 2000$, $\alpha = 0.0099$, $\beta = 0.0099$ and $\epsilon = 1/200$ as our default setting. As we do not assume that Part 1 and Part 3 are independent, samples from Part 1 (Algorithm 2) can also be used in Part 3 (Algorithm 4), so we only need to sample $N_1 + N_2 + M_2 = 37000$ samples in total ($N_1 + N_2$ can cover the sample number $N + N \cdot M$). We use one refinement turn ($T = 1$) for all experiments.

**Attack Setting**: We use the square attack with 5000 iterations for each box. For PGD, we use 20 iterations with step sizes $1/255$ and $2/255$. In each iteration, we find the box with the highest IoU and the same class as the ground-truth box, then use its GIoU as the loss function. Each step may attack a different box. We find this to be a strong attack for YOLO networks.

**Server Setting**: We use a server with 8 NVIDIA V100 32G GPUs, 40 Intel(R) Xeon(R) Gold 5215 CPUs at 2.50GHz and 503GB memory. The code is implemented in Python with Gurobi and PyTorch.

**Sample Calculation**: According to Theorem 2, the sample number is $N_1 + N_2 + M_2 = 37,000$ for each box. Note that this is the sample number of the full algorithm.

Our work (Part 1) aims to find a tight hyper-rectangle bound $\mathcal{Z}$ for the $d_L$-dimensional vector output space $\{F(x)\}_{x \in \mathcal{C}}$. In this context, a more appropriate application of $\text{RCP}_N$ as a baseline for this task would be to directly estimate an upper bound for each dimension in this $d_L$-dimensional space. The decision-variable dimension for this problem would be $d = d_L$ (i.e., the network output dimension), corresponding to finding a vector $u \in \mathbb{R}^{d_L}$ such that, for all inputs $x \in \mathcal{C}$, the network output $F(x) \leq u$ element-wise. Based on this, the required sample count would be:

$$N \geq \left\lceil \frac{2 \ln(1/\beta)}{\alpha} + 2d_L + \frac{2d_L \ln(2/\alpha)}{\alpha} \right\rceil \approx 10^9$$

Alternatively, if one were to use the $\text{RCP}_N$ method to directly verify the entire problem, this could be framed as computing an IoU lower bound for each predicted box against its corresponding ground-truth (GT) box. The decision variable dimension in this case would be $d = (80 \times 80 + 40 \times 40 + 20 \times 20) \times 3 = 25,200$ (i.e., the number of bounding boxes). Based on this, the required sample count would be:

$$N \geq \left\lceil \frac{2 \ln(1/\beta)}{\alpha} + 2d + \frac{2d \ln(2/\alpha)}{\alpha} \right\rceil \approx 10^7$$

The order of magnitude of the sample size under either estimation is still prohibitively large for any practical application.

For the PAC-based methods, according to (Li et al., 2022), to achieve $1 - \alpha$ probability with $1 - \beta$ confidence, we need that $N$ satisfies:

$$N \geq \frac{2}{\beta} \times \left( \log(\frac{1}{\alpha}) + d_0 \right) \geq 122880391.$$

Here we also use $\alpha = 0.02, \beta = 0.02$.

In terms of randomized smoothing(Cohen et al., 2019), the number of samples required is strongly correlated with the standard deviation ($\sigma$) and the certified radius. For example, to certify a radius of $2/255$ against noise with a standard deviation of $\sigma = \frac{1}{255 \times 3}$, the randomized smoothing method would require an impractical number of samples (approximately $3.9 \times 10^9$). This severely limits its application in real-world scenarios.

Our method, by introducing $v_{\max}$ and the scalar $c_1$, reduces the entire verification problem to a $d = 1$ scalar optimization (for $c_1$) plus a constrained optimization (MIQCP). Thus the required sample size is significantly reduced to a manageable level, making it feasible for practical applications.

Table 3: Ground Truth Information for Table 2. #RB: Number of robust boxes; #NRB: Number of non-robust boxes

| Model | $\varepsilon$ | #RB ($\tau = 0.5$) | #NRB($\tau = 0.5$) | #RB ($\tau = 0.7$) | #NRB($\tau = 0.7$) |
|---|---|---|---|---|---|
| yolo11m | $\frac{1}{255}$ | 363 | 164 | 335 | 192 |
| yolo11m | $\frac{2}{255}$ | 346 | 181 | 315 | 212 |
| yolo11x | $\frac{1}{255}$ | 390 | 137 | 348 | 179 |
| yolo11x | $\frac{2}{255}$ | 379 | 148 | 335 | 192 |
| yolov3-sppu | $\frac{1}{255}$ | 364 | 163 | 332 | 195 |
| yolov3-sppu | $\frac{2}{255}$ | 356 | 171 | 320 | 207 |
| yolov3u | $\frac{1}{255}$ | 374 | 153 | 338 | 189 |
| yolov3u | $\frac{2}{255}$ | 365 | 162 | 327 | 200 |
| yolov5mu | $\frac{1}{255}$ | 354 | 173 | 316 | 211 |
| yolov5mu | $\frac{2}{255}$ | 344 | 183 | 301 | 226 |
| yolov5xu | $\frac{1}{255}$ | 387 | 140 | 354 | 173 |
| yolov5xu | $\frac{2}{255}$ | 372 | 155 | 337 | 190 |
| yolov8m | $\frac{1}{255}$ | 376 | 151 | 342 | 185 |
| yolov8m | $\frac{2}{255}$ | 361 | 166 | 327 | 200 |
| yolov8x | $\frac{1}{255}$ | 385 | 142 | 348 | 179 |
| yolov8x | $\frac{2}{255}$ | 370 | 157 | 337 | 190 |

**Remark 7.** *Note that without refinement (only Part 1 and Part 2), our method can achieve 99% confidence and 99% probability with $\alpha = 0.0099, \beta = 0.0099$. After refinement, the true confidence and probability are both 98%. For the other two methods, we use an error rate of $0.02$ and a significance level of $0.02$ to obtain the same confidence and probability.*

**Clarification on Sample Complexity** There are two distinct sampling procedures used in our experiments, which serve different purposes:

- **Verification Samples (37,000):** This is the sample budget used by our algorithm to issue the certificate. For each ground-truth box, we use 37,000 samples ($N_1 + N_2 + M_2$) to construct $\mathcal{Z}$ and the refinement constant $C$, as prescribed by Theorem 2.

- $\text{RCP}_N$ **Baseline Samples** ($10^6$ **in Table 1):** For the $\text{RCP}_N$ baseline in Table 1, we also use $10^6$ samples to compute empirical robustness, but this does not yield comparable theoretical guarantees. To achieve our target (98% robustness, 98% confidence), $\text{RCP}_N$ would theoretically require about 11.6 million samples. With only $10^6$ samples, $\text{RCP}_N$ can only provide an 86% robustness guarantee at 98% confidence.

- **Evaluation Samples** ($10^6$ **in Table 2):** Independently of the verification algorithm, we draw $10^6$ additional uniform perturbations to empirically estimate the "ground truth" robustness. These samples are only used to calculate evaluation metrics (TPR, FPR, TNR, FNR, and CRA) and do not inform the certificate itself.

**Ground truth information of Table 2**: Table 3 shows the ground truth information of Table 2. Here, #RB means the number of robust boxes, and #NRB means the number of non-robust boxes. The ground truth is calculated by $10^6$ uniform perturbations for each box.

## O  THE EFFICIENCY OF NMS VERIFICATION

The NMS optimization step is highly efficient. Verifying one ground-truth box against all candidate boxes takes an average of only 4.9 seconds. This efficiency arises because we only perform sound verification: boxes that could potentially cause unsafe behavior are filtered out in advance, leaving only a small number of boxes to be verified.

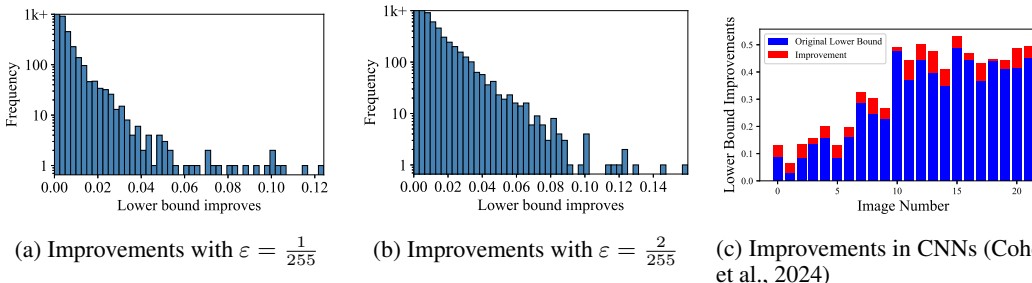

(a) Improvements with $\varepsilon = \frac{1}{255}$  (b) Improvements with $\varepsilon = \frac{2}{255}$  (c) Improvements in CNNs (Cohen et al., 2024)

Figure 7: Verification bound improvement after Part 3

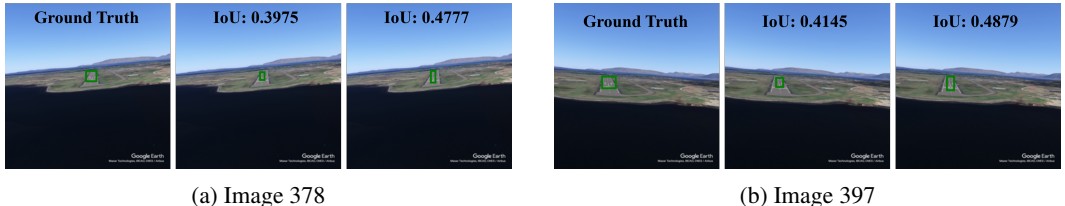

(a) Image 378            (b) Image 397

Figure 8: The effect of Part 3 in small object detection. The middle panel shows the bound from prior methods; the right panel shows our improved bound.

## P  THE EFFECT OF PART 3

This section shows that Part 3 in Section 4 and Appendix K is effective for both large-scale networks (YOLO) and small-scale CNNs. For the LARD dataset, we use a six-layer CNN network provided in (Cohen et al., 2024). As seen in Figure 7c, with Part 3, the IoU lower bound we obtain is higher than the bound obtained without using Part 3. This improvement occurs because Part 3 reduces the over-approximation of the network's output, implying our bound is closer to the true bound. Note that we use sound formal verification on small object detection, instead of probabilistic verification, so that the network can indeed achieve such a bound. For YOLO, we also show the effect of Part 3 in Fig. 7a and Fig. 7b. After refinement, most of the lower bounds are improved, which means Part 3 is also helpful in YOLO networks.

We observe that the first iteration yields the most significant gains. Since further refinement linearly accumulates confidence loss as described in Theorem 2, employing $T = 1$ represents a favorable trade-off in most cases.

## Q  TIME COMPARISON

Fig. 9 shows the detailed time comparison of our method and $\text{RCP}_N$ on different YOLO models with $\varepsilon = \frac{1}{255}$ and $\varepsilon = \frac{2}{255}$. The results show that our method is significantly faster than $\text{RCP}_N$ in all images.

## R  REAL-WORLD EXAMPLES

We take 40 images from sensors on our autonomous vehicles and annotate the ground truth manually. Table 4 shows the results of our method on these images. The results show that our method also works well in real-world scenarios. Our method still uses less time and achieves better bounds than $\text{RCP}_N$. CRA is high on these images, which means most boxes verified as robust by our method are indeed robust.

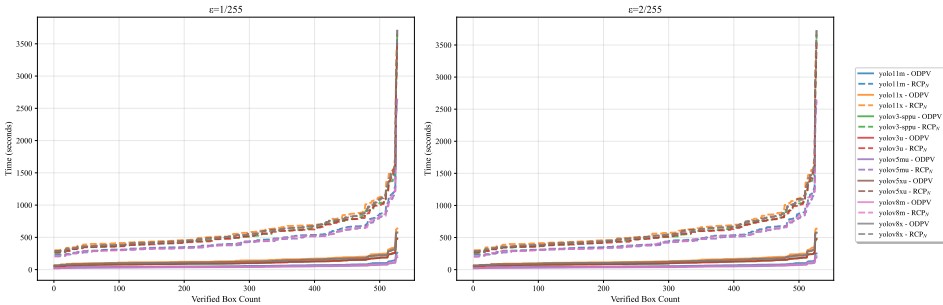

Figure 9: Verification time comparison

Table 4: Real-world examples of our method.

| model | $\varepsilon$ | method | time | $\Delta_{\mathrm{PGD}}$ | | CRA | |
| | | | | $\tau = 0.5$ | $\tau = 0.7$ | $\tau = 0.5$ | $\tau = 0.7$ |
|---|---|---|---|---|---|---|---|
| yolo11x | 1/(3 * 255) | $\mathrm{RCP}_N$ | 571.0 | 0.53 | 0.52 | **1.00** | **1.00** |
| | | ODPV | **32.1** | **0.47** | **0.44** | **1.00** | **1.00** |
| | 2/(3 * 255) | $\mathrm{RCP}_N$ | 569.6 | 0.57 | 0.55 | **1.00** | **1.00** |
| | | ODPV | **32.1** | **0.43** | **0.37** | **1.00** | **1.00** |

## S    COMPARED WITH MEDIAN SMOOTHING

Table 5 compares our method with median smoothing (MS) (Chiang et al., 2020) under a normal distribution on 50 images from the COCO dataset. We set the standard deviation as $\sigma = \frac{1}{255 \times 3}$ and $\sigma = \frac{2}{255 \times 3}$. The results show that our method achieves a smaller mean absolute difference of IoU lower bounds relative to the worst-case input found by the PGD attack ($\Delta_{\mathrm{PGD}}$), indicating more precise IoU lower bounds. In most cases, the CRA of our method is higher than that of median smoothing. This also proves that our method works well across different distributions.

## T    ABLATION STUDY OF PARAMETERS

Fig. 10 shows the ablation study of parameters $\eta$ and $\iota$. The results show that our method maintains more accurate bounds than $\mathrm{RCP}_N$ under different parameters.

## U    BROADER IMPACT

Our method focuses on verifying the safety of the object detection model, which may help people to better understand the model and give a safety metric. We do not think there is a negative social impact of our method as our method is not used to attack the model.

## V    LLM USAGE STATEMENT

In the preparation of this manuscript, a Large Language Model (LLM) was utilized as a writing assistance tool. Its use was strictly confined to language polishing, which includes proofreading for grammatical errors, improving sentence structure, and enhancing the overall clarity and readability of the text.

All core intellectual contributions—including the research ideation, paper structure, and the initial drafting of the content—are the original work of the authors. The LLM did not contribute to the formulation of any hypotheses, experimental results, or conclusions presented herein. The authors have reviewed all AI-generated suggestions and take full responsibility for the final content of this paper.

Table 5: Comparison of our method with median smoothing. $\Delta_{\mathrm{PGD}}$ denotes the mean absolute difference of IoU lower bounds relative to the PGD attack. Bold values indicate the best performance.

| model | $\varepsilon$ | method | $\Delta_{\mathrm{PGD}}$ | | CRA | |
|---|---|---|---|---|---|---|
| | | | $\tau = 0.5$ | $\tau = 0.7$ | $\tau = 0.5$ | $\tau = 0.7$ |
| yolo11x | 1/255 | $\mathrm{RCP}_N$ | 0.45 | 0.47 | **1.00** | **1.00** |
| | | ODPV | **0.42** | **0.40** | 0.99 | 0.98 |
| | | MS | 0.44 | 0.45 | 0.99 | 0.98 |
| | 2/255 | $\mathrm{RCP}_N$ | 0.59 | 0.56 | **1.00** | **1.00** |
| | | ODPV | **0.54** | **0.47** | **1.00** | **1.00** |
| | | MS | 0.59 | 0.53 | 0.96 | 0.97 |

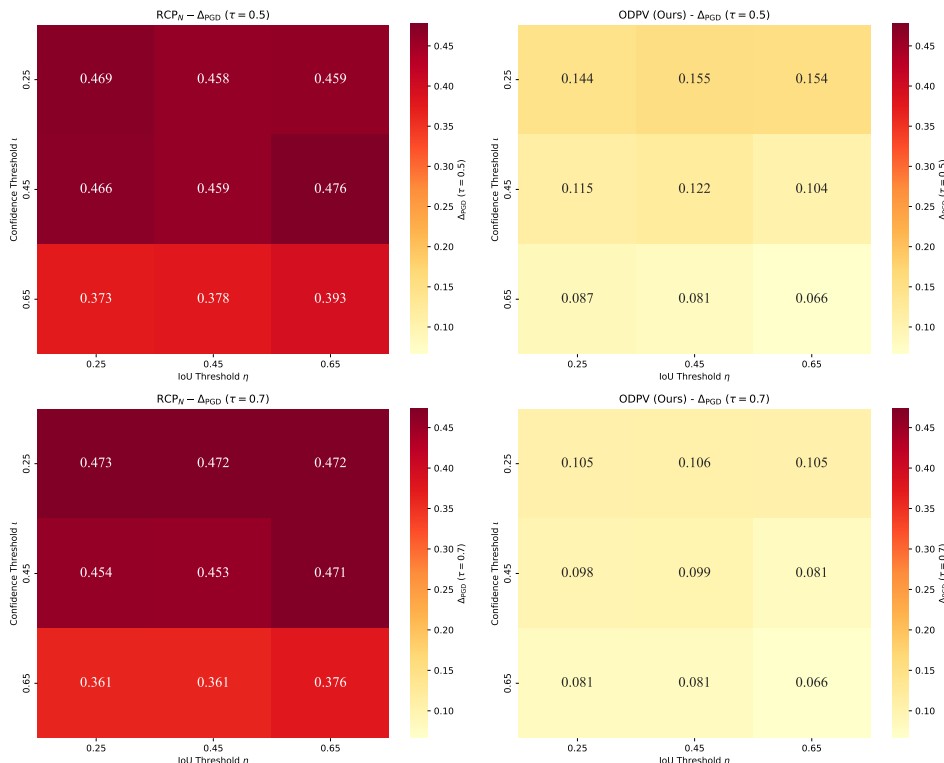

Figure 10: Ablation study of parameters $\eta$, $\iota$

# W  DESCRIPTION OF HYPERPARAMETERS

Our framework consists of three parts, implemented by Algorithms 1–4: Part 1 (Output Approximation), Part 2 (NMS Verification), and Part 3 (Counterexample-Based Refinement). Only Parts 1 and 3 are probabilistic; Part 2 is based on sound MIQCP constraints and introduces no additional probabilistic error.

## W.1  ROLE OF PARAMETERS AND FAILURE EVENTS

We now describe the role of the three key parameters $\alpha, \beta, \epsilon$ used in our theoretical guarantees and experiments.

- $\alpha$ **(Error Rate)**
  - Appears in Definition 1 (OD PAC-Verification), Proposition 1 (Part 1), Theorem 1 (Part 3), and Theorem 2 (Overall Guarantee).

- Controls the *allowable violation probability* of the OD property under the input distribution: In Theorem 2, if the algorithm returns "safe" after at most $T$ refinement steps, we guarantee

$$\mathbb{P}_{\boldsymbol{x} \sim \mathcal{C}}(\boldsymbol{x} \text{ is safe}) > 1 - (1 + T)\,\alpha.$$

- In our experiments, we set $\alpha = 0.0099$, so with $T = 1$ we obtain a lower bound on the safety probability for each certified box of approximately $1 - 0.0198 \approx 0.98$.

- $\beta$ **(Significance / Confidence)**
  - Used for the scenario-type bounds in Part 1 (Prop. 1) and Part 3 (Thm. 1), and combined in Theorem 2.
  - Controls the *confidence* with which the above probabilistic claim holds, i.e., the probability that the inequality regarding $\mathbb{P}(\boldsymbol{x} \text{ is safe})$ is valid under the randomness of our sampling process. Theorem 2 gives the overall confidence:

$$\mathbb{P}\left[\mathbb{P}_{x \sim \mathcal{C}}(\boldsymbol{x} \text{ is safe}) > 1 - (1 + T)\alpha\right] \geq 1 - T\left(\beta + 2(1 - \epsilon)^{M_2}\right) - \beta.$$

  - We use $\beta = 0.0099$; with our default $N, \epsilon, M_2$, and $T = 1$, this lower bound is $\approx 0.98$ (reported in Appendix N).

- $\epsilon$ **(width of the local uncertainty interval in Part 3)**
  - Used in Theorem 1 to define the probability mass of the local distance interval $[B_{\mathrm{F}(\boldsymbol{x})}, A_{\mathrm{F}(\boldsymbol{x})}]$. For each $\boldsymbol{x}$, at least a $1 - 2\epsilon$ fraction of the $M$ local perturbations fall into this interval.
  - The term $2(1 - \epsilon)^{M_2}$ in Theorem 2 bounds the probability that the empirical estimation of this interval fails.
  - We set $\epsilon = 1/200$ in all experiments.

In practice, one needs to select appropriate $\alpha, \beta, \epsilon$ based on the desired probability thresholds and then calculate the sample sizes $N_1, N_2, N, M, M_2$. Below we discuss the relationship between sample sizes and these parameters.

## W.2 SAMPLE SIZE RELATIONSHIPS

The main sample size constraints appear in Theorem 2:

- **Part 1 Sampling:**

$$N_2 \geq \lceil \frac{2\ln(1/\beta)}{\alpha} + 2 + \frac{2\ln(2/\alpha)}{\alpha} \rceil$$

  For fixed $\alpha, \beta$, we have $N_2 = O\left(\frac{1}{\alpha} \log \frac{1}{\beta}\right)$. Importantly, there is **no explicit dependency on the output dimension** $d_L$ in this bound.

- **Part 3 Sampling:** With fixed $\epsilon, M$, the bound on $N$ can be written as

$$N \geq \frac{1}{(1 - 2\epsilon)^M}\left(\frac{1}{\alpha} \ln \frac{1}{\beta}\right),$$

  Thus, for fixed $\epsilon, M$, we again have $N = O\left(\frac{1}{\alpha} \log \frac{1}{\beta}\right)$.

In the experiments, we instantiate these quantities with the following values:

$$N_1 = 30{,}000, \quad N_2 = 5{,}000, \quad N = 3{,}000, \quad M = 10, \quad M_2 = 2{,}000,$$

and $\alpha = \beta = 0.0099$, $\epsilon = 1/200$ (Appendix N). Because the samples from Part 1 are reused in Part 3 (Remark 6), the total number of samples per box is:

$$N_{\text{total}} = N_1 + N_2 + M_2 = 37{,}000$$

This quantity is independent of $d_L$ and forms the basis for the claim "achieving 98% guarantee at 98% confidence with 37,000 samples" in Section 6.

### W.3 NOTATION FOR SAMPLING PARAMETERS

Throughout Proposition 1, Theorem 1–2, the symbols $N_1, N_2, N, M, M_2, A'_i, B'_i, A_m, B_m$ and $C$ are inherited from Algorithms 2 and 4 as follows. In Algorithm 2, $N_1$ denotes the number of samples used to estimate the per-coordinate scale vector $\boldsymbol{v}_{\max}$, and $N_2$ is the number of samples used to compute the smallest scaling factor $c_1$ in the optimization problem equation 1, which yields the output hyper-rectangle $\mathcal{Z}$. In Algorithm 4, $N$ is the number of reference points $\{\boldsymbol{x}^{(i)}\}_{i=1}^N$ drawn from $\mathcal{C}$ in Step One, and $M$ is the number of auxiliary samples $\{\boldsymbol{x}^{(i,j)}\}_{j=1}^M$ drawn for each reference point. For each $i \in [N]$, we define $A'_i = \max_{j \in [M]} \|\mathrm{F}(\boldsymbol{x}^{(i,j)}) - \mathrm{F}(\boldsymbol{x}^{(i)})\|_2$ and $B'_i = \min_{j \in [M]} \|\mathrm{F}(\boldsymbol{x}^{(i,j)}) - \mathrm{F}(\boldsymbol{x}^{(i)})\|_2$, and the constant $C$ is updated as $C \leftarrow \max\{C, B'_i/(A'_i - B'_i)\}$. In Step Two of Algorithm 4, $M_2$ denotes the number of additional samples $\{\boldsymbol{x}^{(i)}\}_{i=1}^{M_2}$ drawn from $\mathcal{C}$ to estimate the distance from a candidate output $\boldsymbol{y}$ to the true output set $\{\mathrm{F}(\boldsymbol{x})\}_{\boldsymbol{x} \in \mathcal{C}}$, with $A_m = \max_{i \in [M_2]} \|\mathrm{F}(\boldsymbol{x}^{(i)}) - \boldsymbol{y}\|_2$ and $B_m = \min_{i \in [M_2]} \|\mathrm{F}(\boldsymbol{x}^{(i)}) - \boldsymbol{y}\|_2$. These quantities are then combined to form the estimator $d_{\min} = \max\{(B_m - C(A_m - B_m))/(1 + 2C), 0\}$ used in Part 3.

## X EXTENDING OUR METHOD

We acknowledge that different architectures and threat models may necessitate distinct verification strategies, making a universal solution challenging. Nonetheless, for many common threats, our proposed method can be adapted by modifying only the encoding in Part 2, while keeping Parts 1 and 3 unchanged. The following describes how to adapt our method to verify two additional threats: class misidentification and false appearances.

### X.1 EXTENDING TO CLASS MISIDENTIFICATION

---
**Algorithm 6** Sound Verification of Class Misidentification
---
**Require:** Constraints $\mathcal{Z} = \mathcal{H} \setminus \mathcal{S}$; ground truth $box_{\mathrm{gt}}$; confidence threshold $\iota$.
**Ensure:** Calculate per-box upper bounds $\{\tau_{\mathrm{mis}}(i)\}_{i=1}^{n_{\boldsymbol{x}}}$ for class misidentification.
1:   $\{\overline{box}_i\}_{i=1}^{n_{\boldsymbol{x}}} \leftarrow \text{CONSTRUCT\_ABSTRACT\_BOX}(\mathcal{Z})$
2:   **for all** $\overline{box}_i \in \{\overline{box}_k\}_{k=1}^{n_{\boldsymbol{x}}}$ **do**
3:       $\tau_{\mathrm{mis}}(i) \leftarrow 0$                    $\triangleright$ Initialize upper bound for misidentification IoU of box $i$
4:       **if** $\forall k \in [n_{\mathrm{cls}}] \setminus \{\mathrm{Class}(box_{\mathrm{gt}})\},\ \underline{p}_{i,\mathrm{Class}(box_{\mathrm{gt}})} \geq \overline{p}_{i,k}$ **then**
5:           **continue**  $\triangleright$ Skip boxes that must match $box_{\mathrm{gt}}$ class (never misclassified w.r.t. this GT)
6:       **end if**
7:       **if** $\overline{c_i} \geq \iota$ **then**                    $\triangleright$ Box $i$ may pass the confidence threshold in some realization
8:           $\tau_{\mathrm{mis}}(i) \leftarrow \text{IOU\_UPPER\_BOUNDS}(\overline{box}_i, box_{\mathrm{gt}})$ $\triangleright$ Worst-case IoU to $box_{\mathrm{gt}}$ when $box_i$ is potentially of a wrong class
9:       **end if**
10: **end for**
11: **return** $\{\tau_{\mathrm{mis}}(i)\}_{i=1}^{n_{\boldsymbol{x}}}$
---

**Formalization of the Property (Bad Event):** Given an input constraint set $\mathcal{C}$, if there exists an input $\boldsymbol{x} \in \mathcal{C}$ such that after processing by the network F and NMS module, the output set $\mathrm{N}(\mathrm{F}(\boldsymbol{x}))$ contains a predicted box $box_i$ that has an IoU $\geq \tau$ with some GT box $box_{\mathrm{gt}}$, but their predicted and GT classes do not match, we consider this a class misidentification. Formally:

$$\exists \boldsymbol{x} \in \mathcal{C},\ \exists box_i \in \mathrm{N}(\mathrm{F}(\boldsymbol{x})),\ \exists box_{\mathrm{gt}} \in \mathcal{G} : \mathbb{I}(\mathrm{Class}(box_i) \neq \mathrm{Class}(box_{\mathrm{gt}})) \cdot \mathrm{IoU}(box_i, box_{\mathrm{gt}}) \geq \tau.$$

We want to prove that class misidentification does **not** occur, which is the negation of the bad event, equivalently written as:

$$\forall \boldsymbol{x} \in \mathcal{C},\ \forall box_i \in \mathrm{N}(\mathrm{F}(\boldsymbol{x})),\ \forall box_{\mathrm{gt}} \in \mathcal{G} : \mathbb{I}(\mathrm{Class}(box_i) \neq \mathrm{Class}(box_{\mathrm{gt}})) \cdot \mathrm{IoU}(box_i, box_{\mathrm{gt}}) < \tau.$$

To this end, we can define a worst-case function:

$$\Phi_{\mathrm{cls}}(\boldsymbol{x}) = \max_{box_i \in \mathrm{N}(\mathrm{F}(\boldsymbol{x})),\, box_{\mathrm{gt}} \in \mathcal{G}} \left( \mathbb{I}(\mathrm{Class}(box_i) \neq \mathrm{Class}(box_{\mathrm{gt}})) \cdot \mathrm{IoU}(box_i, box_{\mathrm{gt}}) \right),$$

The property holds if and only if: $\sup_{\boldsymbol{x} \in \mathcal{C}} \Phi_{\text{cls}}(\boldsymbol{x}) < \tau$.

Let $\mathcal{Z}$ be the over-approximation of the pre-NMS network output range $\{F(\boldsymbol{x})\}_{\boldsymbol{x} \in \mathcal{C}}$ obtained in Part 1. Since our NMS analysis in Part 2 is applied uniformly over all $y \in \mathcal{Z}$, we can compute an upper bound of $\mathbb{I}(\text{Class}(box_i) \neq \text{Class}(box_{\text{gt}})) \cdot \text{IoU}(box_i, box_{\text{gt}})$ on $\mathcal{Z}$ over all possible predicted boxes and GT pairs:

$$U_{\text{cls}} = \sup_{y \in \mathcal{Z},\, box_i \in N(y),\, box_{gt} \in \mathcal{G}} \left( \mathbb{I}(\text{Class}(box_i) \neq \text{Class}(box_{\text{gt}})) \cdot \text{IoU}(box_i, box_{\text{gt}}) \right).$$

If we can prove $U_{\text{cls}} < \tau$, then it is impossible for any $\boldsymbol{x} \in \mathcal{C}$ and any prediction/GT pair to have an IoU $\geq \tau$ and a class mismatch. Thus, we can certify that no class misidentification occurs within $\mathcal{C}$, and the property is verified.

Algorithm 6 shows how to compute per-box upper bounds on the misidentification IoU in Part 2.

## X.2 Extending to False Appearances

---
**Algorithm 7** Sound Verification of False Appearance
---
**Require:** Constraints $\mathcal{Z} = \mathcal{H} \setminus \mathcal{S}$; set of ground truth boxes $\mathcal{G}$; confidence threshold $\iota$.
**Ensure:** Calculate per-box lower bounds $\{\tau_{\text{FA}}(i)\}_{i=1}^{n_x}$ on $\max_{box_{\text{gt}} \in \mathcal{G}} \text{IoU}(box_i, box_{\text{gt}})$.
 1: $\{\overline{box_i}\}_{i=1}^{n_x} \leftarrow \text{CONSTRUCT\_ABSTRACT\_BOX}(\mathcal{Z})$
 2: **for all** $\overline{box_i} \in \{\overline{box_k}\}_{k=1}^{n_x}$ **do**
 3: $\quad$ $\tau_{\text{FA}}(i) \leftarrow 0$ $\quad\quad\quad\quad\quad\quad\quad$ ▷ Initialize lower bound for maximal IoU to GTs of box $i$
 4: $\quad$ **if** $\overline{c_i} < \iota$ **then**
 5: $\quad\quad$ **continue** $\quad$ ▷ Box $i$ can never become a high-confidence detection, ignore it for False Appearance
 6: $\quad$ **end if**
 7: $\quad$ $lb \leftarrow 0$ $\quad\quad\quad\quad\quad\quad\quad\quad\quad\quad\quad$ ▷ Lower bound on $\max_{box_{\text{gt}} \in \mathcal{G}} \text{IoU}(box_i, box_{\text{gt}})$
 8: $\quad$ **for all** $box_{\text{gt}} \in \mathcal{G}$ **do**
 9: $\quad\quad$ $lb_{\text{gt}} \leftarrow \text{IOU\_LOWER\_BOUNDS}(\overline{box_i}, box_{\text{gt}})$
10: $\quad\quad$ $lb \leftarrow \max(lb, lb_{\text{gt}})$ $\quad$ ▷ Aggregate lower bounds to obtain a conservative lower bound of $\max_{box_{\text{gt}}} \text{IoU}$
11: $\quad$ **end for**
12: $\quad$ $\tau_{\text{FA}}(i) \leftarrow lb$
13: **end for**
14: **return** $\{\tau_{\text{FA}}(i)\}_{i=1}^{n_x}$

---

**Formalization of the Property (Bad Event):** Given an input constraint set $\mathcal{C}$, if there exists an input $\boldsymbol{x} \in \mathcal{C}$ and a predicted box $box_i \in N(F(\boldsymbol{x}))$ whose IoU with all GT boxes is less than $\tau$, we consider this a False Appearance:

$$\exists \boldsymbol{x} \in \mathcal{C},\ \exists box_i \in N(F(\boldsymbol{x})),\ \forall box_{\text{gt}} \in \mathcal{G},\ \text{IoU}(box_i, box_{\text{gt}}) < \tau.$$

Equivalently, define the **maximum IoU** for each predicted box with all GT boxes:

$$\text{IoU}_{\max}(box_i, \boldsymbol{x}) = \max_{box_{\text{gt}}} \text{IoU}(box_i, box_{\text{gt}}),$$

The bad event can be written as:

$$\exists \boldsymbol{x} \in \mathcal{C},\ \exists box_i \in N(F(\boldsymbol{x})) :\ \text{IoU}_{\max}(box_i, \boldsymbol{x}) < \tau.$$

We want to prove "no false appearances occur", which is the negation of the bad event:

$$\forall \boldsymbol{x} \in \mathcal{C},\ \forall box_i \in N(F(\boldsymbol{x})) :\ \text{IoU}_{\max}(box_i, \boldsymbol{x}) \geq \tau.$$

We can further define:

$$\Phi_{\text{FA}}(\boldsymbol{x}) = \min_{box_i \in N(F(\boldsymbol{x}))} \text{IoU}_{\max}(box_i, \boldsymbol{x}),$$

and the property holds if and only if $\inf_{\boldsymbol{x} \in \mathcal{C}} \Phi_{\text{FA}}(\boldsymbol{x}) \geq \tau$.

Algorithm 7 shows how to compute per-box lower bounds on the maximum IoU to GT boxes in Part 2.

For any given property, we first formalize the attack and verification objective as described in Section 3 and the process above. Then, as in Section 4 and above, we adapt Part 2 of the algorithm (e.g., by adjusting the MIQCP constraints) based on the specific verification objective.

### X.3 EVALUATION

To assess the effectiveness of our proposed method under diverse noise conditions and threat types, we conduct experiments using three distinct noise models and verify the False Appearance (FA) detection performance of our YOLO11x model. We define a noise tensor $\mathbf{N}_x$ and set the perturbation magnitude to $\varepsilon = 1/255$. The specific noise distributions and their corresponding real-world motivations are outlined below:

- **Uniform**: $\mathbf{N}_x \sim \mathcal{U}(-\varepsilon, \varepsilon)$ (Quantization/Uncertainty).
- **Gaussian**: $\mathbf{N}_x \sim \mathcal{N}(0, (\varepsilon/3)^2)$ (Sensor Readout Noise).
- **Salt-and-Pepper**: $\pm\varepsilon$ impulse noise with $p = 0.05$ (Transmission Faults).

We randomly select 10 images from the COCO validation set and apply each noise model with $\varepsilon = 1/255$ to generate noisy inputs. We then evaluate the FA detection verification performance of our YOLO11x model on these perturbed images. For each input, we draw $10^6$ samples from the corresponding noise distribution to approximate the ground truth. The results are summarized in the table below:

Table 6: FA Detection Verification Performance under Diverse Noise Models

| Model | Noise type | $\text{CRA}_{\text{FA}}$ | $\text{TPR}_{\text{FA}}$ | $\text{TNR}_{\text{FA}}$ | $\text{FPR}_{\text{FA}}$ | $\text{FNR}_{\text{FA}}$ |
|-------|------------|------|------|------|------|------|
| yolo11x | gaussian | 100.00% | 92.31% | 100.00% | 0.00% | 7.69% |
| yolo11x | salt and pepper | 100.00% | 85.71% | 100.00% | 0.00% | 14.29% |
| yolo11x | uniform | 100.00% | 92.31% | 100.00% | 0.00% | 7.69% |

A detection is considered positive if it remains robust under the corresponding noise model and negative otherwise. The results indicate that our method sustains a high Certified Robust Accuracy (CRA) across different noise types, demonstrating that it provides reliable guarantees under diverse real-world noise conditions and threat types. Moreover, the True Positive Rate (TPR) and True Negative Rate (TNR) remain consistently high, while the False Positive Rate (FPR) and False Negative Rate (FNR) stay low, underscoring the method's effectiveness in distinguishing between robust and non-robust detections.

Overall, these results demonstrate that our method remains reliable across heterogeneous noise distributions and diverse threat types. This confirms that the proposed framework is broadly applicable and provides trustworthy robustness guarantees under a wide range of real-world noise conditions.

