# OpenReview forum: "Certifying the Full YOLO Pipeline: A Probabilistic Verification Approach"
_ICLR.cc/2026/Conference — ICLR 2026 Poster_

### Official Review · Reviewer_ji71 · 2025-10-24

**Soundness:** 3
**Presentation:** 3
**Contribution:** 3
**Rating:** 6
**Confidence:** 2

**Summary:**

This paper proposes a three-step probabilistic verification framework to assess the robustness of YOLO object detection systems against object disappearance attacks. By estimating output ranges, verifying NMS, and refining results, the method provides strong probabilistic guarantees and tighter IoU lower bounds with fewer samples, scaling effectively to practical models.

**Strengths:**

1. The authors propose the first scalable probabilistic verification framework for YOLO models, effectively addressing the object disappearance threat in safety-critical scenarios.
2. Combines solid theoretical analysis with comprehensive experiments, showing tighter IoU lower bounds and improved robustness.
3. Achieves strong probabilistic guarantees with significantly fewer samples compared to existing methods.

**Weaknesses:**

1. The method is very focused on object disappearance, which is an important threat, but also a pretty narrow one. It’s not clear how well the framework would generalize to other attack scenarios or tasks.

2. The experimental comparison feels a bit limited. Most of the analysis is against one baseline (RCPN), and a broader set of baselines would make the claims more convincing.

3. There’s little discussion on actual computational cost. While the paper shows faster verification times, it doesn’t really explain how the method would scale in a real deployment or resource-constrained setting.

**Questions:**

1. How does the method generalize under different perturbation distributions?
2. Could this verification framework be integrated with robustness training methods to further improve security?

---

> ### Author Response · Authors · 2025-11-20
> **Response to Weaknesses [1/3]**
>
> ### (W1) Scope of Application and Framework Generality
>
> **Why We Focus on Object Disappearance (OD)**
>
> We agree that the current implementation focuses primarily on the Object Disappearance (OD) threat. We chose OD primarily because, for perception in autonomous driving and similar systems, "missing" a pedestrian or vehicle is often more critical than mis-localization. OD is a primary, core failure mode in practice, and certifying against it is an important step. Furthermore, OD is technically more challenging to verify compared to many other common issues.
>
> **Framework Generality**
>
> We acknowledge that different architectures and threat models may necessitate distinct verification strategies, making a universal solution challenging.
> Nonetheless, for many common threats, our proposed method can be adapted by modifying only the encoding in Part 2, while keeping Parts 1 and 3 unchanged.
> In the next response, we will describe how to adapt our method to verify two additional threats: class misidentification and false appearances.
>
>
> ### (W2) Baselines: Comparison Beyond $\mathrm{RCP}_N$
>
> We appreciate the suggestion to broaden the scope of our baselines. We already include some discussion of this point in the main text.
>
> * **Why many baselines are infeasible.** As shown in Appendix N and the main text, achieving comparable probabilistic guarantees for models at the scale of YOLO would require an enormous number of samples for $\mathrm{RCP}_N$ and DeepPAC, making them practically impossible to run. Formal verification methods are also infeasible in this setting: existing tools can typically handle only 2–3 convolutional layers with 2–3 linear layers, far below the scale of YOLO, and cannot address its complex architecture or NMS. Therefore, direct comparisons with DeepPAC, $\mathrm{RCP}_N$, and existing formal verification tools are not feasible at the YOLO scale. Thus, $\mathrm{RCP}_N$ is almost the only viable baseline for large-scale models like YOLO.
> * **Our chosen baseline.** Therefore, we used $\mathrm{RCP}_N$ at a large, feasible scale ($10^6$ samples). At this scale, our method still achieves a **tighter IoU lower bound** and is faster.
> * **Other baselines included in the paper.** We also compared against **median smoothing** under Gaussian perturbations (Appendix S). Although not a direct competitor, our method still achieved a lower $\Delta \text{PGD}$ and comparable or higher Certified Robust Accuracy (CRA). In Appendix N, we also illustrate the sample complexity required by randomized smoothing under specific settings, further highlighting the advantage of our method.
> * **Small network scenario.** For small CNNs (Appendix P), a scenario tractable for formal verifiers, our refinement step produced stronger IoU bounds, demonstrating competitiveness in this setting as well.
>
> ### (W3) Computational Cost and Scalability
>
>
> We agree that the discussion on scalability can be further strengthened.
>
> * **Sample Complexity.** Our full pipeline achieves **98% safety at 98% confidence** using only **37k (37,000) samples** per bounding box, whereas prior methods would require on the order of $10^8$ samples.
> * **Empirical Runtimes.** Table 1 and Figure 9 show that our method is **4x-10x faster** than RCPN when verifying YOLO models.
> * **NMS Verification Cost.** Appendix O shows that the filtered MIQP (Mixed-Integer Quadratic Program) for a single bounding box takes **4.9 seconds** on average.
> * **Hardware Feasibility.** Based on our tests, our method can be run on a consumer-grade laptop (with an NVIDIA RTX 3070 GPU).
>
> However, it should be noted that our method is currently intended for **offline verification**, not real-time deployment, which is a common characteristic of most existing verification methods. In practical applications, our verification speed is highly efficient in offline verification scenarios.
> Both theoretically and empirically, our method requires far fewer resources than existing verification methods, making it more feasible in resource-constrained environments.

---

> ### Author Response · Authors · 2025-11-20
> **Supplementary Experiments for (W1) [2/3]**
>
> The following extends our framework to verify two additional threat types: class misidentification and false appearances (spurious detections).
>
> **Class Misidentification**
> To certify that no predicted box $box_i$ has high IoU with a ground truth $box_{gt}$ of a different class, we verify that the maximum "mismatched IoU" over the approximated set $\mathcal{Z}$ is below $\tau$. We define the upper bound $U_{\mathrm{cls}}$:
> $$
> U_{\mathrm{cls}} = \sup_{box_i \in \mathcal{Z}, box_{\mathrm{gt}}\in\mathcal{G}} \left( \mathbb{I}(\mathrm{cls}(box_i) \neq \mathrm{cls}(box_{\mathrm{gt}})) \cdot \mathrm{IoU}(box_i, box_{\mathrm{gt}}) \right).
> $$
> If $U_{\mathrm{cls}} < \tau$, we certify that no misidentification occurs within $\mathcal{C}$.
>
> **False Appearances (Spurious Detections)**
> To certify that every predicted box corresponds to a real object (i.e., matches *at least one* GT box), we verify that the worst-case minimum IoU overlap is sufficient. The property holds if:
> $$
> \inf_{\boldsymbol{x} \in \mathcal{C}} \left( \min_{box_i \in \mathrm{N}(\mathrm{F}(\boldsymbol{x}))} \max_{box_{\mathrm{gt}}\in\mathcal{G}} \mathrm{IoU}(box_i, box_{\mathrm{gt}}) \right) \ge \tau.
> $$
> This ensures that for all perturbed inputs, every post-NMS detection anchors to a ground truth object. We will include these formal extensions in the final manuscript.
>
> For any given property, we first formalize the attack and verification objective as described in Section 3 and the process above. Then, as in Section 4 and above, we adapt Part 2 of extensions in the revised manuscript, and specify how Part 2 of the algorithm should be modified for these two threats.
>
> ### Experimental Validation
>
> To assess the effectiveness of our proposed method under diverse noise conditions and threat types, we conduct experiments using four distinct noise models and verify the False Appearance (FA) detection performance of our YOLO11x model. We define a noise tensor $\mathbf{N}_x$ and set the perturbation magnitude to $\varepsilon = 1/255$. The specific noise distributions and their corresponding real-world motivations are outlined below:
>
> * **Uniform:** $\mathbf{N}_x \sim \mathcal{U}(-\varepsilon, \varepsilon)$ (Quantization/Uncertainty).
> * **Gaussian:** $\mathbf{N}_x \sim \mathcal{N}(0, (\varepsilon/3)^2)$ (Sensor Readout Noise).
> * **Salt-and-Pepper:** $\pm\varepsilon$ impulse noise with $p=0.05$ (Transmission Faults).
>
>
> We randomly select 10 images from the COCO validation set and apply each noise model with $\varepsilon = 1/255$ to generate noisy inputs. We then evaluate the FA detection verification performance of our YOLO11x model on these perturbed images. For each input, we draw $10^6$ samples from the corresponding noise distribution to approximate the ground truth. The results are summarized in the table below:
>
> | Model   | Noise type      | $\mathrm{CAR}\_{\text{FA}}$   | $\mathrm{TPR}\_{\text{FA}}$   | $\mathrm{TNR}\_{\text{FA}}$   | $\mathrm{FPR}\_{\text{FA}}$   | $\mathrm{FNR}\_{\text{FA}}$   |
> |:--------|:----------------|:-----------------------------|:-----------------------------|:-----------------------------|:-----------------------------|:-----------------------------|
> | yolo11x | gaussian         | 100.00%                      | 92.31%                       | 100.00%                      | 0.00%                        | 7.69%                        |
> | yolo11x | salt and pepper | 100.00%                      | 85.71%                       | 100.00%                      | 0.00%                        | 14.29%                       |
> | yolo11x | uniform         | 100.00%                      | 92.31%                       | 100.00%                      | 0.00%                        | 7.69%                        |
>
> A detection is considered positive if it remains robust under the corresponding noise model and negative otherwise. The results indicate that our method sustains a high Certified Accuracy Rate (CAR) across different noise types, demonstrating that it provides reliable guarantees under diverse real-world noise conditions and threat types. Moreover, the True Positive Rate (TPR) and True Negative Rate (TNR) remain consistently high, while the False Positive Rate (FPR) and False Negative Rate (FNR) stay low, underscoring the method’s effectiveness in distinguishing between robust and non-robust detections.
>
> Overall, these results demonstrate that our method remains reliable across heterogeneous noise distributions and diverse threat types. This confirms that the proposed framework is broadly applicable and provides trustworthy robustness guarantees under a wide range of real-world noise conditions. the algorithm (e.g., by adjusting the MIQP constraints) based on the specific verification objective. We will add a discussion of these and other potential

---

> ### Author Response · Authors · 2025-11-20
> **Response to Qeustions [3/3]**
>
> ### (Q1) Generalization to Different Perturbation Distributions
>
>
> Our theoretical framework does not require the perturbation distribution to be uniform. Our guarantee only requires that the samples are independent and identically distributed (i.i.d.) from *any* perturbation distribution. Our method does not rely on specific distributional assumptions. We can sample perturbations from any distribution (uniform, Gaussian, adversarial, realistic sensor noise, etc.) as long as they are i.i.d.
>
> In the experimental section, we primarily use uniform distribution for sampling, as it is the most common choice and widely used in the literature.
>
> In Appendix S, we evaluated the performance of our method, $\mathrm{RCP}_N$, and median smoothing under **Gaussian perturbations**. Our method again outperformed median smoothing on $\Delta \text{PGD}$ and achieved comparable or higher CRA (Certified Robust Accuracy), further demonstrating our method's advantage and generalization capability across different perturbation distributions. We also added experiments on **salt-and-pepper noise**, **gaussian noise**, **uniform noise** and **false appearance detection** in response to W1 above, further showcasing the generality of our framework.
>
> ### (Q2) Integration with Robustness Training
>
>
> This is a very interesting suggestion. Our method is designed as a **post-training, black-box certification** technique that does not involve the training process. Therefore, it is naturally complementary to robust training methods:
>
> * It can certify models obtained through adversarial training or certified training without any modification to the model.
> * The certified IoU lower bound can also serve as a verification metric or as a signal to guide training (e.g., as a loss term for certified perception).
>
> We thank the reviewer for the suggestion and will briefly discuss this direction as future work.

---

> ### Comment · Reviewer_ji71 · 2025-11-20
>
> Thank you for the authors’response. I feel that most of my concerns have been addressed. The addition of the salt and pepper experiment and the false representation experiment has helped broaden the scope of the paper, not just the disappearance of objects, which is what I am most worried about. The author clarified why DeepPAC and standard formal validators are computationally infeasible on the YOLO scale, which makes sense. Therefore, I accept the reliance on RCPN as the main baseline. I admit that post-training authentication has its own value. Given the solid theoretical work in NMS verification and the stricter IoU boundaries demonstrated, I maintain my positive score.

---

> > ### Author Response · Authors · 2025-11-20
> >
> > We appreciate the Reviewer’s positive feedback and their time spent reviewing our manuscript. We are glad to hear that most of the Reviewer’s concerns have been addressed.

---

### Official Review · Reviewer_YCar · 2025-10-28

**Soundness:** 2
**Presentation:** 2
**Contribution:** 3
**Rating:** 4
**Confidence:** 4

**Summary:**

The paper introduces a PAC-based local verification procedure for YOLO networks---object detection networks that produce annotation boxes---specially for object disappearance (OD) threats (see, e.g., [Eykholt et al., 2018](https://www.usenix.org/conference/woot18/presentation/eykholt)).
These networks have very high-dimensional input and especially output spaces and use non-trivial post-processing, which makes verification with off-the-shelf methods difficult.

The approach combines multiple sampling-based probabilistic techniques to solve the task in multiple steps.
1. Approximate (from a sample) a hyper-rectangle as a high probability output "bounding box" for the relevant region in the input space.
2. (Repeatedly):
    - Detect unsafe output points $\mathbf y$ with Quadratic programming, if none found, return **safe**.
    - Attempt (with random sampling) to find inputs $\mathbf x$ such that $\lVert F(\mathbf x) - \mathbf y\rVert$ is minimal
    - If no close enough input found, refine output space, else return unsafe
The resulting guarantees are relaxed depending on multiple probabilistic hyperparameters, that dictate various sample sizes.

**Strengths:**

- **(S1) Significance and Motivation**
    - The problem is important for gauging the robustness of YOLO networks, and existing methods cannot be trivially adapted to this domain.
    - The Introduction and Related Work motivate the work well and give a nice overview of the neural net verification landscape, although some additional sources for more scalable PAC approaches can be discussed (e.g., [Baluta et.al., 2021](https://ieeexplore.ieee.org/abstract/document/9402111), [Blohm et al., 2025](https://openreview.net/forum?id=UKHlXpiFMy)).

- **(S2) Theoretical Contribution**
    - The obtained sample complexities are realistic, and proofs of the main results seem correct after surface-level checks.
    - Neural network verification with PAC methods has been widely explored, yet the unique output modality and post-processing seem to require more involved methods for verification.
    - The authors leverage multiple probabilistic techniques in tandem, leading to low sample complexity for the overall procedure; samples appear quasi-independent of the network dimensionality.
    - The general idea of integrating PAC methods with counterexample-guided refinement in verification is interesting and implemented in a novel way.

- **(S3) Empirical Results**
    - The procedure seems able to deal with model sizes common in related literature and issues robustness certificates that are close to the results of adversarial methods.

While I have several questions regarding the details of the procedure, I think they can be addressed, and their inclusion in the manuscript would strengthen the contribution.

 I would be happy to see a refined version of this submission accepted.

**Weaknesses:**

The main weakness of this manuscript is the complicated presentation of an already complex procedure.
Important details of the probabilistic procedure are difficult to gauge from the manuscript.

Many theoretical issues may be easily clarified or adapted by the authors and do not greatly impact the theoretical contribution.
However, a clear statement of the limitations and failure modes is necessary for a probabilistic procedure like this.

- **(W1) Readability and Notation**: The notation is very dense, which is partially difficult to avoid in this output modality.
While not the main concern of the manuscript, the readability of the manuscript would improve a lot if:
  - Abbreviations were introduced more consistently (e.g., YOLO, RCP, PGD, ...).
  - Citations were wrapped in parentheses when they are not part of a sentence (i.e., use `\citep{}`-style formatting if you use LaTeX/natbib).
  - Tables avoided heavy rules and used a cleaner booktabs style, as is typical for ML conferences like ICLR.

- **(W2) Opaque Sample Complexity**: The procedure combines sampling-based approaches in a “nested” way, resulting in four hyperparameters $\alpha,\beta,\delta,\epsilon$, as well as empirically chosen sample sizes for parts of the procedure.
This makes it difficult to gauge how precisely the sample complexity and runtime scale with each parameter, and what trade-offs different choices bring to the overall procedure.

  The fact that the likelihoods of failure events depend on multiple parameters amplifies this, as multiple choices of different parameters can lead to the same sample sizes.

  **Actionable request:** Can you provide a mapping of each parameter to its failure event, sample complexity (big-O), or its impact on the overall runtime?
  This might not be necessary to include in the main text, but it potentially gives a nice overview of the procedure.

- **(W3) Partially Empirical Network Output Approximation**: The output domain of the network is estimated by scaling $\mathbf v_{\max}$, which, in my understanding, is essentially an empirically chosen proposal vector.
  The scaling coefficient is chosen with a probabilistic guarantee with $O(\frac{1}{\alpha} (\ln \frac{1}{\beta}+\ln\frac{1}{\alpha}))$ samples.
  While this scaling coefficient is chosen optimally, the proposal vector seems to be chosen based on a heuristic, with the statement in Proposition 3 in Appendix D being opaque.

  There is a lot of established theory on obtaining bounding rectangles (see alternatives below).
  These methods do not give the dimension-independent sample complexities that are presented here.
  However, the presented method consequently relies on a scaling trick and might overapproximate the tightest bounding rectangle if the proposal vector was not chosen well.

  The potential looseness of the initial output-space approximation is amplified by eliminating $L\_2$ balls from counterexamples, instead of using $L_\infty$.
   Over-approximations in a single dimension may require many refinement steps to eliminate, especially in very high-dimensional spaces.
  **Actionable request:** see **(Q1)**

 - **(W4) $L_2$ Counterexample Refinement and MIQP Runtime in Higher Dimensions**: Refinement steps of the output space are performed with a Mixed Quadratic Integer Program.
  While this idea is novel and interesting, it is unclear how effective the refinement steps are, and whether the found counterexamples, in general, significantly shrink the output domain to be searched.
A discussion/investigation of this seems important, as each refinement step brings the cost of increased uncertainty.
- **(W5) Unclear Empirical Results in Table 2 and Figure 5**:

  - It is unclear what results precisely are presented in Table 2. The Baseline Selection and Appendix N state that 37,000 samples are used to certify the proposed method, with the baseline $\mathrm{RCP}_N$ on $10^6$ samples. However, the caption of Table 2, as well as *Safety Guarantee*, states $10^6$ uniform perturbations. In either scenario, the results are presented in a slightly confusing manner: either the runtime is compared to a method using significantly more samples (clarify in Table 1), or Table 2 shows results for significantly larger sample complexity (and thus tighter parameters).
  - The caption states $\epsilon = 1/255$, yet $\epsilon$ is varied in the table.
  - The false negative rate appears high (up to 15%). It is not stated on how many robust/non-robust boxes these results were computed; standard deviations are also missing.
  - Figure 5’s x-axis is not labelled; it is not stated which model/experiment produced this data. Code does not appear to be available, so details cannot be checked.

**Questions:**

I would appreciate it if the authors briefly addressed my concerns in **$W2$** as well as answered my questions below to address the remaining weaknesses.

 - **(Q1) Initial Approximation of Output Domain**:
Many methods exist to give an \(\alpha,\beta\) guarantee on estimating a hyper-rectangle. How does your Part 1 method compare in terms of the tightness of the obtained rectangle? How much of an issue do over-approximations of the output domain present in practice?

   There is established theory for estimating non-parametric confidence regions from samples, e.g., the **DKW inequality** ([Dvoretzky–Kiefer–Wolfowitz, 1956](https://projecteuclid.org/journals/annals-of-mathematical-statistics/volume-27/issue-3/Asymptotic-Minimax-Character-of-the-Sample-Distribution-Function-and-of/10.1214/aoms/1177728174.full); [Massart, 1990](https://projecteuclid.org/journals/annals-of-probability/volume-18/issue-3/The-Tight-Constant-in-the-Dvoretzky-Kiefer-Wolfowitz-Inequality/10.1214/aop/1176990746.full); for multivariate exact constants, see [Naaman, 2021](https://www.sciencedirect.com/science/article/pii/S016771522100050X)).
   Similarly, one can learn the smallest hyper-rectangle with empirical risk minimization (ERM).

   Alternatively, there exist dimension-dependent bounds with **$\epsilon$-nets**—see [Haussler & Welzl, 1987](https://link.springer.com/article/10.1007/BF02187876) and the standard reference [Mitzenmacher & Upfal, 2017](https://www.cambridge.org/core/books/probability-and-computing/3A5B47DB315FC64B9256C5C8131C5EFA). One could consider the class of *inverse* (half-)hyperrectangles with VC-dimension $d$. After a sample of size $\tilde{O}(\frac{d}{\epsilon})$, one could use $\mathbf v_{\max}$ directly as a bound, without the need to scale by a constant.
   These alternatives trade dimension-independence for transparency; a short discussion comparing your scaling heuristic to empirical rectangles/$\epsilon$-nets would help position your choice.

- **(Q2) Counterexample Refinement with MIQP**: From reading the manuscript, it is not obvious that refinement steps significantly reduce the volume of the output domain (i.e., that the produced $\mathbf y$ will be far from the actual codomain of $F$). Consequently, refinement might weaken the probabilistic guarantees in Theorem 2 without real advantage. Why eliminate an $L_2$ neighborhood of each counterexample rather than, for example, an $L_\infty$ hypercube (or hyper-rectangles)?
Wouldn’t that allow eliminating much larger volumes in high dimensions, especially for images?

   It would be very useful to investigate how much each successive refinement step not only increases the bounds but actually decreases the size of the output domain, as well as how much it relaxes the resulting bounds. A discussion of when further refinement is “not worth it” in terms of cost in confidence would be valuable.

- **(Q3) Counterexample Validation and Refinement**: The theoretical idea of Theorem 1 is opaque from the main text. A brief mention of the probabilistic idea behind the “more conservative estimates” in §5.3 (with a citation or name of the invoked bound/inequality) would help communicate the approach.
  The proof of Theorem 1 in the appendix mentions Hoeffding’s inequality; explaining its role in one sentence in the main text would help.

- **(Q4) Motivation versus PGD**: If I understand correctly, Figure 5 shows that PGD often finds tighter lower bounds than probabilistic verification.
   Is this an issue for the motivation of the procedure?

   With an attack that is presumably cheaper than the proposed method, one can seemingly get tighter bounds.
   What advantage does the proposed method offer over performing a PGD attack over an \(\tilde{O}(1/\epsilon)\) sample and reporting the tightest counterexample as a bound (cf. [Blohm et al., 2025](https://openreview.net/forum?id=UKHlXpiFMy))?

   In general, can adversarial methods be integrated into the approach instead of relying on uniform random samples? If such integration is out of scope, a motivation for using a probabilistic procedure when a cheap attack can immediately provide a counterexample would help.

- **(Q5)Clarification of Experimental Setup**:
  - What is the precise sample complexity of the results in Table 2---37,000 or $10^6$?
  - What is the precise certificate that the procedure issues for the instances, in terms of the probabilistic bounds and their interpretation?
  - Which column in the table reflects actual robustness behaviour?
  - Approximately, what is the certificate (confidence) provided by $\mathrm{RCP}_N$ at the used sample complexity?

---

## Minor Recommendations

**Self-Contained Theorems** In §5.3–5.4, including Theorems 1–2, some notation is not reintroduced (e.g., $A', B', C$). In the theorems, restate the meanings of $N, M, M_2$ for self-containment. In Theorem 2, “the algorithms defined above” should be referenced specifically (reffing Algorithm 1 might suffice if others are subroutines).

**Naming of Subroutines.** The manuscript would be easier to follow if “Part 1/2/3” were replaced by names (e.g., *Output-Box Estimation*, *Unsafe-$\mathbf y$ Search*, *Counterexample Validation/Refinement*). The algorithm captions would read more cleanly without the repeated “Part Y” phrasing.

**Typos.** One of the OpenReview keywords reads “guaranteen.” In Algorithm 4, use $\mathbb{Z}^+$ rather than $Z^+$.

---

## References

- Blohm, P.; Indri, P.; Gärtner, T.; Malhotra, S. (2025). *Probably Approximately Global Robustness Certification.* ICML 2025. OpenReview: <https://openreview.net/forum?id=UKHlXpiFMy>
- Baluta, T.; Chua, Z. L.; Meel, K. S.; Saxena, P. (2021). “Scalable Quantitative Verification for Deep Neural Networks.” Proceedings of the 2021 IEEE/ACM 43rd International Conference on Software Engineering (ICSE), 312–323. <https://doi.org/10.1109/ICSE43902.2021.00039>
- Dvoretzky, A.; Kiefer, J.; Wolfowitz, J. (1956). “Asymptotic Minimax Character of the Sample Distribution Function and of the Classical Multinomial Estimator.” *Annals of Mathematical Statistics*, 27(3), 642–669. <https://projecteuclid.org/journals/annals-of-mathematical-statistics/volume-27/issue-3/Asymptotic-Minimax-Character-of-the-Sample-Distribution-Function-and-of/10.1214/aoms/1177728174.full>
- Massart, P. (1990). “The Tight Constant in the Dvoretzky–Kiefer–Wolfowitz Inequality.” *Annals of Probability*, 18(3), 1269–1283. <https://projecteuclid.org/journals/annals-of-probability/volume-18/issue-3/The-Tight-Constant-in-the-Dvoretzky-Kiefer-Wolfowitz-Inequality/10.1214/aop/1176990746.full>
- Naaman, M. (2021). “On the Tight Constant in the Multivariate Dvoretzky–Kiefer–Wolfowitz Inequality.” *Statistics & Probability Letters*, 173, 109088. <https://www.sciencedirect.com/science/article/pii/S016771522100050X>
- Haussler, D.; Welzl, E. (1987). “\(\varepsilon\)-nets and Simplex Range Queries.” *Discrete & Computational Geometry*, 2, 127–151. <https://link.springer.com/article/10.1007/BF02187876>
- Mitzenmacher, M.; Upfal, E. (2017). *Probability and Computing: Randomization and Probabilistic Techniques in Algorithms and Data Analysis* (2nd ed.). Cambridge University Press. <https://www.cambridge.org/core/books/probability-and-computing/3A5B47DB315FC64B9256C5C8131C5EFA>

---

> ### Author Response · Authors · 2025-11-20
> **Presentation and Notation Improvements (Respnse to W1 & Minor Recommendations) [1/6]**
>
> We sincerely thank the reviewer for the careful reading and the many constructive suggestions.
> Below we address each point (W/Q) in turn and indicate the clarifications or additions we will provide in the revised version
>
> ---
>
> ### (W1) Readability and Notation
>
> We thank the reviewer for the comments on readability and agree that the presentation can be improved without altering the technical content.
>
> * **Abbreviations.** In the revised version, we will ensure all abbreviations (e.g., YOLO, NMS, OD...) are introduced upon their first use in the main text and that the same abbreviations are used consistently throughout the paper and appendix.
>
> * **Citation Style.** We will adopt a consistent parenthetical citation style (e.g., using `\citep{...}` in LaTeX/natbib) when the citation is not grammatically part of the sentence, and use `\citet{...}` only when the author's name appears in the sentence.
>
> * **Tables and Typesetting.** We will reformat tables to use the standard `booktabs` style, avoid heavy lines, and unify the fonts and notation between the main text and the appendix. These changes are purely presentational and do not affect the technical development.
>
> ### Response to Minor Recommendations
>
> We thank the reviewer for the valuable suggestions for improvement. We will revise the "Self-Contained Theorems" section to ensure the theorems are self-contained and will replace "Part 1/2/3" in the algorithms with more descriptive names. Additionally, we will correct the mentioned typos and notational issues.

---

> ### Author Response · Authors · 2025-11-20
> **Sample Complexity & Parameters (Response to W2)  [2/6]**
>
> ### (W2) Sample Complexity and Parameters
>
> We agree that the nested probabilistic construction and multiple parameters $(\alpha, \beta, \delta, \epsilon)$ make it difficult to grasp the overall sample complexity behavior at a glance. We clarify the mapping between parameters, failure events, and sample sizes here; we will add a detailed explanation of these parameters in the appendix.
>
> #### **Role of Parameters and Failure Events**
>
> Our framework consists of three parts (Algorithms 1–4): Part 1 (Output Approximation), Part 2 (NMS Verification), and Part 3 (Counterexample-Based Refinement). Only Parts 1 and 3 are probabilistic; Part 2 is based on sound MIQP constraints and introduces no additional probabilistic error.
>
> * **$\alpha$ (Error Rate)**
>   * Appears in Definition 1 (OD PAC-Verification), Proposition 1 (Part 1), Theorem 1 (Part 3), and Theorem 2 (Overall Guarantee).
>   * Controls the *allowable violation probability* of the OD property under the input distribution: In Theorem 2, if the algorithm returns "safe" after at most T refinement steps, we guarantee
>         $$
>         \mathrm{P}_{x\sim \mathcal{C}}(\text{x is safe}) > 1 - (1+T)\,\alpha.
>         $$
>   * In our experiments, we set $\alpha = 0.0099$, so with $T = 1$ we obtain a lower bound on the safety probability for each certified box of approximately $1 - 0.0198 \approx 0.98$.
>
> * **$\beta$ (Significance / Confidence)**
>   * Used for the scenario-type bounds in Part 1 (Prop. 1) and Part 3 (Thm. 1), and combined in Theorem 2.
>   * Controls the *confidence* with which the above probabilistic claim holds, i.e., the probability that the inequality regarding `P(x is safe)` is valid under the randomness of our sampling process. Theorem 2 gives the overall confidence:
>         $$
>         \mathrm{P}\left[\mathrm{P}_{x\sim \mathcal{C}}(\boldsymbol{x}\text{ is safe}) > 1-(1+T)\alpha\right]
>         \ge 1 - T\big(e^{-2N\delta^2} + \beta + 2(1-\epsilon)M_2\big) - \beta.
>         $$
>   * We use $\beta = 0.0099$; with our default N, $\delta, \epsilon, M_2$, and T = 1, this lower bound is $\approx 0.98$ (reported in Appendix N).
>
> * **$\delta$ (Concentration slack in Part 3)**
>   * Appears only in Theorem 1 / Theorem 2. It is the slack term in the Hoeffding bound, controlling how many of the N sampled points in Part 3 have "good" local neighborhoods (i.e., have sufficient neighbors within the interval $[B_{\mathrm{F}(\boldsymbol{x})}, A_{\mathrm{F}(\boldsymbol{x})}]$).
>   * The condition
>         $$
>         N((1-2\epsilon)M - \delta) > \frac{2}{\alpha}\ln\frac{1}{\beta} + 2 + \frac{2}{\alpha}\ln\frac{2}{\alpha}
>         $$
>         ensures that with probability at least $1-\beta$, enough sample points are "good" to apply the $\mathrm{RCP}_N$-style scenario bound to the refinement constant $C$.
>   * We fix $\delta = 0.1$ in all experiments (Appendix N).
>
> * **$\epsilon$ (width of the local uncertainty interval in Part 3)**
>   * Used in Theorem 1 to define the probability mass of the local distance interval $[B_{\mathrm{F}(\boldsymbol{x})}, A_{\mathrm{F}(\boldsymbol{x})}]$. For each $\boldsymbol{x}$, at least a $1-2\epsilon$ fraction of the M local perturbations fall into this interval.
>   * The term $2(1-\epsilon)M_2$ in Theorem 2 bounds the probability that the empirical estimation of this interval fails.
>   * We set $\epsilon = 1/200$ in all experiments.
>
> In practice, one needs to select appropriate $\alpha, \beta, \delta, \epsilon$ based on the desired probability thresholds and then calculate the sample sizes $N_1, N_2, N, M, M_2$. Below we discuss the relationship between sample sizes and these parameters.
>
> #### **Sample Size Relationships**
>
> The main sample size constraints appear in Theorem 2:
>
> * **Part 1 Sampling:**
>     $N_2 \ge \frac{2}{\alpha}\ln\frac{1}{\beta} + 2 + \frac{2}{\alpha}\ln\frac{2}{\alpha}$.
>     For fixed $\alpha, \beta$, we have $N_2 = O\big(\frac{1}{\alpha}\log\frac{1}{\beta}\big)$. Importantly, there is **no explicit dependency on the output dimension $d_L$** in this bound.
>
> * **Part 3 Sampling:**
>     With fixed $\delta, \epsilon, M$, the bound on N can be written as
>     $$
>     N \ge \frac{1}{(1-2\epsilon)M-\delta}
>     \Big( \frac{2}{\alpha}\ln\frac{1}{\beta} + 2 + \frac{2}{\alpha}\ln\frac{2}{\alpha} \Big),
>     $$
>     Thus, for fixed $\delta, \epsilon, M$, we again have $N = O\big(\frac{1}{\alpha}\log\frac{1}{\beta}\big)$.
>
> In the experiments, we instantiate these quantities with the following values:
> $$
> N_1 = 30000, \quad N_2 = 5000, \quad N = 3000, \quad M = 10, \quad M_2 = 2000,
> $$
> and $\alpha = \beta = 0.0099, \epsilon = 1/200, \delta = 0.1$ (Appendix N). Because the samples from Part 1 are reused in Part 3 (Remark 5), the total number of samples per box is:
> $$
> N_{\text{total}} = N_1 + N_2 + M_2 = 37,000
> $$
> This quantity is independent of $d_L$ and forms the basis for the claim "achieving 98% guarantee at 98% confidence with 37,000 samples" in Section 6.

---

> ### Author Response · Authors · 2025-11-20
> **Output Domain Approximation (Response to W3 & Q1)  [3/6]**
>
> We appreciate the reviewer's insightful comments regarding the heuristics used in our output domain approximation and the connections to classical non-parametric estimation theory. We address the concerns regarding the choice of our method over established alternatives and the potential impact of over-approximation below.
>
> ### 1. Design Choice: Dimension-Independence vs. Classical Methods (DKW, ERM, $\epsilon$-nets)
>
> The reviewer is correct that our construction (scaling a proposal vector) is a heuristic and does not guarantee the *tightest* possible hyper-rectangle.
>
> We thank the reviewer for highlighting classical alternatives such as DKW-based confidence regions, ERM-based hyper-rectangles, and $\epsilon$-nets. While these methods are theoretically robust, their sample complexity typically scales with the output dimension $d_L$ (e.g., $\tilde{O}(d_L/\epsilon)$ for $\epsilon$-nets).
>
> **Reason for our approach:** The primary motivation for our Part 1 design was to achieve a sample complexity that is **independent** of the output dimension (as established in Theorem 2). Given the extremely high-dimensional output space of YOLO networks, dimension-dependent bounds like those from $\epsilon$-nets or DKW would be computationally infeasible. We chose this trade-off—sacrificing some transparency in the shape of the confidence region—to ensure scalability.
>
> **Action Plan:** In the revised version, we will add a discussion explicitly comparing our scenario-based construction with these classical methods, highlighting them as complementary alternatives that offer different trade-offs between tightness and scalability.
>
> ### 2. Practical Tightness and Mitigation of Over-approximation
>
> The reviewer asks how much of an issue over-approximation presents in practice. While we acknowledge that our initial hyper-rectangle is an over-approximation, multiple lines of empirical evidence suggest it is not overly loose and is effectively refined:
>
> * **Tightness compared to attacks:** Our final certified bounds are very close to those found by strong adversarial attacks (as shown in Table 1 and Figure 5). This indicates that our approximation (Part 1) combined with refinement (Part 3) captures the true behavior of the network well.
> * **High Verification Success:** As shown in Table 2, our method successfully verifies hundreds of boxes per YOLO model with a high True Positive Rate (TPR), demonstrating that the approximation is tight enough to be useful for certification tasks.
> * **Refinement (Part 3) effectively reduces looseness:** We designed Part 3 specifically to address potential over-approximation from Part 1. By removing $L_2$ balls centered at counterexamples, we "carve out" regions of the hyper-rectangle that are far from the actual output set.
>     * **Evidence:** Appendix P (Figures 7 and 8) demonstrates that this refinement step improves the certified IoU lower bound for both YOLO and smaller CNNs. This confirms that even if the initial Part 1 approximation is loose, Part 3 tightens the domain.
>
> In summary, while better approximation methods could theoretically replace Part 1, our current heuristic combined with counterexample-guided refinement strikes a necessary balance for verifying high-dimensional object detectors.

---

> ### Author Response · Authors · 2025-11-20
> **Refinement and Geometry (Response to W4 & Q2) [4/6]**
>
> We thank the reviewer for the interesting questions regarding the geometric interpretation and efficiency of our counterexample refinement step (Part 3). We address the concerns regarding the effectiveness of the refinement, the choice of exclusion shapes ($L_2$ vs. $L_\infty$), and the cost-benefit trade-off below.
>
> ### 1. Effectiveness of Refinement and Volume Reduction
>
> The reviewer asks if the refinement significantly shrinks the output domain. While measuring the exact volume reduction of $\mathcal{Z}$ in such high-dimensional spaces is computationally intractable, we quantify the **effective** reduction via the improvement in the certified IoU lower bound. Since the Objectness/Detection property is monotonic in the IoU, an increase in the certified lower bound implies that we have successfully "carved out" regions of the initial hyper-rectangle where unsafe outputs were previously thought to exist.
>
> **Empirical Evidence:** Appendix P provides direct evidence of this effectiveness:
> * **YOLO Models:** Figures 7(a) and 7(b) show that for most verification instances, the refinement step increases the certified IoU lower bound.
> * **Smaller CNNs:** Figures 7(c) and 8 demonstrate similar improvements over prior methods. Furthermore, we use sound formal verification on these smaller networks to confirm that the refined bounds are indeed achievable and safe.
>
> ### 2. Why Choose $L_2$ Balls Instead of Hypercubes/Rectangles?
>
> We chose to exclude $L_2$ balls rather than $L_\infty$ hypercubes or hyper-rectangles for three primary reasons:
>
> 1. **Theoretical Consistency:** The concentration arguments in Theorem 1 are derived using Euclidean norms. The local distance interval and the refinement constant $C$ bound $\|\mathrm{F}(\boldsymbol{x})-\boldsymbol{y}\|_2$. Using $L_2$ balls allows us to keep the refinement step within the same mathematical framework.
> 2. **Computational Feasibility (MIQP):** $L_2$ exclusions can be naturally encoded as quadratic constraints in our Mixed-Integer Quadratic Program (MIQP).
> 3. **High-Dimensional Geometry:** While a hypercube might seem to cover more volume, safely fitting one inside the exclusion region is non-trivial. If we constrain a hypercube based on the exclusion radius $d$ (diagonal length), its side length becomes $\frac{d}{\sqrt{d_L}}$. In high-dimensional spaces (large $d_L$), this results in a hypercube with a significantly smaller volume than the corresponding $L_2$ ball.
>
> **Action Plan:** We will clarify in the revision that $L_2$ balls were a design choice driven by analytical consistency and solver compatibility. If more efficient methods to compute and encode exclusion regions of this shape existed, we would agree they could replace the $L_2$ ball.
>
> ### 3. Cost-Benefit Analysis: When is Refinement "Not Worth It"?
>
> The reviewer correctly notes that refinement comes with a cost in confidence. In our experiments, we limited the procedure to $T=1$ refinement step. This choice was motivated by the trade-off presented in Theorem 2:
>
> * **The Cost:** Each additional refinement step decreases our confidence. The failure probability terms scale with $T$, specifically involving terms like $(1+T)\alpha$.
> * **The Benefit:** As shown in Appendix P, the first refinement step yields the most significant jump in IoU bounds.
>
> For large-scale detectors, the marginal improvement in bounds from $T>1$ typically does not justify the cumulative drop in statistical confidence (or the requirement for significantly more samples to maintain the same confidence).

---

> > ### Comment · Reviewer_YCar · 2025-11-24
> > **Brief Comment Regarding the choice of norms**
> >
> > I appreciate the author's detailed responses to my comments so far.
> > For this argument in particular, I just want to briefly remark regarding Part 2:
> >
> > **Theoretical Consistency**: This makes sense and is sufficient as an explanation, I think. However, I wonder if a similar result for $L_\infty$ could be given.
> > **Computational Feasibility**: Please correct me if I am mistaken, but $L_\infty$ balls are linear, which, in my understanding, provides even more ease in their encoding. This point might be an argument against $L_2$ balls.
> > **High-Dimensional Geometry**: Finding the largest $L_2$/$L_\infty$ ball that does not intersect a given point will yield incomparable volumes. However, in expectation, an $L_\infty$-ball will yield more volume. In high-dimensional space, this difference is drastic. The presented argument of "We need to constrain the $L_\infty$ ball into an $L_2$ region anyway" seems to boil back to the fact that the theoretical result was derived with Euclidean norms in mind.
> >
> > Please be careful when discussing this. I do not claim that your argument is invalid, but presented in this way, it might lead to a misunderstanding. Two of the arguments, at a surface level read, sound easy to refute.

---

> ### Author Response · Authors · 2025-11-20
> **Experimental Clarifications (Response to W5 & Q5) [5/6]**
>
> We appreciate the reviewer's careful examination of our experimental results. We apologize for the confusion caused by the presentation of sample sizes and table captions. Below, we clarify the experimental setup, the specific certificate guarantees, and the results presented in Table 2 and Figure 5.
>
> ### 1. Clarification on Sample Complexity: 37,000 vs. $10^6$ Samples
> There are two distinct sampling procedures used in our experiments, which serve different purposes:
>
> 1. **Verification Samples (37,000):** This is the sample budget used by our algorithm to issue the certificate. For each ground-truth box, we use **37,000 samples** ($N_1+N_2+M_2$) to construct $\mathcal{Z}$ and the refinement constant $C$, as prescribed by Theorem 2.
> 2. **Evaluation Samples ($10^6$):** Independently of the verification algorithm, we draw $10^6$ *additional* uniform perturbations to empirically estimate the "ground truth" robustness. These samples are **only used to calculate evaluation metrics** (TPR, FPR, TNR, FNR, and CRA) and do not inform the certificate itself.
> 3. **$\mathrm{RCP}_N$ Baseline Samples ($10^6$):** For the $\mathrm{RCP}_N$ baseline in Table 1, we also use $10^6$ samples to compute empirical robustness, but this does not yield comparable theoretical guarantees.
>
> We will make this distinction explicit in Section 6 and the figure captions in the revision.
>
> ### 2. Precise Certificate and Baseline Comparison
> **Our Certificate:**
> Based on the 37,000 verification samples, our method issues a certificate with specific probabilistic bounds. Using our default parameters ($T=1, \alpha=\beta=0.0099, \epsilon=1/200, \delta=0.1$), the certificate guarantees:
> * **Probabilistic Robustness:** $\mathrm{P}_{x\sim \mathcal{C}}(\text{No OD Event}) > 1 - (1+T)\alpha \approx 0.98$.
> * **Confidence:** This bound holds with confidence $\approx 98\%$ over the algorithm's randomness.
>
> Formally, this satisfies the conditions derived in Theorem 2.
>
> **Comparison with $\mathrm{RCP}_N$:**
> In Table 1, we use $\mathrm{RCP}_N$ with $N=10^6$ as an *empirical* baseline. It does not offer comparable theoretical guarantees.
> * To achieve our target (98% robustness, 98% confidence), $\mathrm{RCP}_N$ would theoretically require **~11.6 million samples**.
> * With only $10^6$ samples, $\mathrm{RCP}_N$ can only provide an 86% robustness guarantee at 98% confidence.
>
> ### 3. Clarifications on Table 2 (Statistics, Captions, and FNR)
> **Caption Correction:**
> The reviewer is correct that the caption contained a typo. We will correct it to state that the table reports results for $\tau = 0.5$ and $\varepsilon \in \{1/255, 2/255\}$ under $10^6$ uniform perturbations.
>
> **Interpretation of Columns:**
> * **"True" Robustness:** Determined by the $10^6$ evaluation samples. A box is "robust" if no OD event occurred during these extensive trials.
> * **CRA (Certified Robust Accuracy):** The fraction of boxes certified as robust by our method *AND* confirmed robust by the $10^6$ evaluation samples. The high CRA (>98%) confirms our certificates are consistent with observed behavior.
>
> **Dataset Size and FNR:**
> We evaluated 100 validation images from COCO containing 527 ground-truth objects. The observed FNR (up to ~15%) reflects the conservative nature of our certification: difficult-to-certify boxes are classified as non-robust to maintain the high confidence guarantee.
>
> The number of robust/non-robust boxes per model is provided below:
>
> | Model | $\varepsilon$ |#robust box ($\tau=0.5$) |#not robust box ($\tau=0.5$) |#robust box ($\tau=0.7$) |#not robust box ($\tau=0.7$)|
> |:------------|:------|----:|----:|----:|----:|
> | yolo11m     | 1/255 | 363 | 164 | 335 | 192 |
> | yolo11m     | 2/255 | 346 | 181 | 315 | 212 |
> | yolo11x     | 1/255 | 390 | 137 | 348 | 179 |
> | yolo11x     | 2/255 | 379 | 148 | 335 | 192 |
> | yolov3-sppu | 1/255 | 364 | 163 | 332 | 195 |
> | yolov3-sppu | 2/255 | 356 | 171 | 320 | 207 |
> | yolov3u     | 1/255 | 374 | 153 | 338 | 189 |
> | yolov3u     | 2/255 | 365 | 162 | 327 | 200 |
> | yolov5mu    | 1/255 | 354 | 173 | 316 | 211 |
> | yolov5mu    | 2/255 | 344 | 183 | 301 | 226 |
> | yolov5xu    | 1/255 | 387 | 140 | 354 | 173 |
> | yolov5xu    | 2/255 | 372 | 155 | 337 | 190 |
> | yolov8m     | 1/255 | 376 | 151 | 342 | 185 |
> | yolov8m     | 2/255 | 361 | 166 | 327 | 200 |
> | yolov8x     | 1/255 | 385 | 142 | 348 | 179 |
> | yolov8x     | 2/255 | 370 | 157 | 337 | 190 |
>
>
> ### 4. Figure 5 Details
> We will update the axis labels and caption for clarity. Figure 5 displays verification results for **YOLOv11** ($\tau=0.5, \varepsilon=1/255$). The x-axis represents individual **box instances** (a random subset), and the y-axis plots the IoU lower bound calculated by our method compared to $\mathrm{RCP}_N$, PGD, and Square Attack.

---

> ### Author Response · Authors · 2025-11-20
> **Intuition for Theorem 1 and Comparison with PGD (Response to Q3 & Q4) [6/6]**
>
> ### (Q3) Intuition behind Theorem 1
>
> We agree that the intuition behind Theorem 1 could be explained more clearly in the main text. We provide a brief high-level explanation here, which we will incorporate into Section 5.3.
>
> Theorem 1 is concerned with estimating a constant C such that, with high probability over the sampling in Part 3,
> $$
> \|\mathrm{F}(\boldsymbol{x}) - \boldsymbol{y}\|_2 \le C \quad \text{for most } \boldsymbol{x}\sim \mathcal{C},
> $$
> whenever $\boldsymbol{y}$ is in the region $\mathcal{Z}$. The proof proceeds in three steps:
>
> 1. For each sampled point $\boldsymbol{x}$, we estimate a local interval $[B_{\mathrm{F}(\boldsymbol{x})}, A_{\mathrm{F}(\boldsymbol{x})}]$ that, with probability at least $1-2\epsilon$, covers the $L_2$ distances $\|\mathrm{F}(\boldsymbol{x}') - \mathrm{F}(\boldsymbol{x})\|_2$ to its $M$ nearby samples.
> 2. Using Hoeffding’s inequality, we show that with high probability over the $N$ sampled base points, at least $N((1-2\epsilon)M - \delta)$ points have all $M$ neighbors within their respective intervals.
> 3. On this subset, we apply an RCPN-style scenario bound to the function $S_{\mathrm{F}(\boldsymbol{x})}$ (which aggregates the local distances), yielding an upper bound $C$ on $\|\mathrm{F}(\boldsymbol{x}) - \boldsymbol{y}\|_2$ that holds with probability at least $1-\alpha$ over $\boldsymbol{x}$.
>
> Hoeffding's inequality is used explicitly in Step 2 to bound the deviation of the empirical fraction of "good" points from its expectation, $(1-2\epsilon)^M$. We will add a short paragraph to the main text describing these steps and explicitly citing Hoeffding's inequality as a key tool.
>
> ### (Q4) Comparison with PGD Attacks
>
> Q4 raises an important conceptual question: if PGD often finds tighter adversarial examples at a lower computational cost, what is the added value of our probabilistic verification procedure?
>
> ### Different Goals: Counterexample vs. Certificate
>
> PGD (and other adversarial attacks) and our method solve different problems:
>
> * **PGD / Attacks:** Given a fixed input x, find *one* perturbation $\delta$ such that the OD property is violated. If successful, the output is a **counterexample**. Failure to find a counterexample provides **no** guarantee about unsearched perturbations.
>
> * **ODPV (Our method):** For a given x and ground-truth box, certify that with high probability over $x' \sim \mathcal{C}$,
>     $$
>     \mathrm{P}_{x'\sim \mathcal{C}}(\text{OD does not occur}) > 1 - (1+T)\alpha,
>     $$
>     with a confidence of at least $1 - T(e^{-2N\delta^2} + \beta + 2(1-\epsilon)M_2) - \beta$. This is a **global probabilistic guarantee** over the entire perturbation region, not a single point.
>
> Thus, even if a strong attack empirically finds a slightly smaller IoU than the bound we certify, that attack does not quantify the **probability** of such an event occurring under the perturbation distribution, nor does it provide a confidence level. Our method trades some tightness for an explicit probabilistic guarantee.
>
> In fact, for real-world applications, a probabilistic guarantee is often more valuable than a single counterexample. This is partly because the extreme perturbations found by PGD may be difficult to encounter in reality, and partly because a network that is robust with high probability under common perturbations is acceptable for practical deployment. The strict requirement of complete robustness in a neighborhood often leads to a significant drop in network performance.
>
> #### Why not just use "PGD + Scenario Analysis"?
>
> The reviewer suggests running PGD on $O(1/\epsilon)$ samples and using the tightest counterexample as a bound. In principle, this empirical bound could be wrapped in a PAC-style framework, but:
>
> * It is difficult for PGD methods to provide a probabilistic guarantee.
> * One would need to treat the combination of "random sampling + adversarial search" as a new, composite sampling procedure and derive appropriate bounds for it.
> * Our method is designed to provide probabilistic guarantees for robustness under a distribution $\mathcal{C}$, which can be a common, realistic perturbation distribution (like sensor noise), whereas PGD typically targets worst-case perturbations. As stated earlier, a network that is robust with high probability under common perturbations is acceptable for practical deployment, whereas the strict requirement of complete robustness in a neighborhood often leads to a significant drop in network performance.
>
> We view the integration of adversarial search into PAC verification as a promising direction, but one that is outside the scope of this paper.
>
> #### Can adversarial methods be integrated?
>
> Yes, conceptually. For example, one could use adversarial attacks to propose candidate counterexamples, which are then fed into our Part 3 refinement, or use them to guide the construction of $\mathcal{Z}$. However, this would require additional theoretical work to maintain the probabilistic guarantee.

---

> > ### Comment · Reviewer_YCar · 2025-11-24
> > **Comparison Against PGD Attacks**
> >
> > First, the authors correctly state that the goals of Attack and Certification are distinct. A failed attack gives no statement, but the question is: If attacks can find counterexamples easily, and they do not require a lot of compute, their statement might be more comparable to realistic settings than a probabilistic guarantee (based on a uniform distribution).
> >
> > The arguments against PGD and Scenario Analysis are somewhat unclear to me; two of the sources I provided explicitly combine attacks and PAC:
> > - Blohm, P.; Indri, P.; Gärtner, T.; Malhotra, S. (2025). Probably Approximately Global Robustness Certification. ICML 2025. OpenReview: https://openreview.net/forum?id=UKHlXpiFMy
> > - Baluta, T.; Chua, Z. L.; Meel, K. S.; Saxena, P. (2021). “Scalable Quantitative Verification for Deep Neural Networks.” Proceedings of the 2021 IEEE/ACM 43rd International Conference on Software Engineering (ICSE), 312–323. https://doi.org/10.1109/ICSE43902.2021.00039
> >
> > >    It is difficult for PGD methods to provide a probabilistic guarantee.
> >
> > I do not understand what is meant here.
> >
> > >    One would need to treat the combination of "random sampling + adversarial search" as a new, composite sampling procedure and derive appropriate bounds for it.
> >
> > This is done, e.g., in the two papers above.
> >
> > >    Our method is designed to provide probabilistic guarantees for robustness under a distribution $\mathcal{C}$, which can be a common, realistic perturbation distribution (like sensor noise), whereas PGD typically targets worst-case perturbations. As stated earlier, a network that is robust with high probability under common perturbations is acceptable for practical deployment, whereas the strict requirement of complete robustness in a neighborhood often leads to a significant drop in network performance.
> >
> > **This is a good argument**. If a random process such as sensor noise is the target for this approach, then it is reasonable to rely on randomized rather than targeted approaches.
> >
> > I think it is fair to say that an adversarial setting is not quite what this paper aims to protect against, but I would be cautious in conflating the three points made above.
> > There are PAC-style methods, even dimensionality-independent ones, that build upon this combined sampling and attack approach.
> > This also now answers my question regarding "What is the point in comparison to PGD" more.
> >
> > It might be stated in the paper, but I think it would be good if this distinction were made a bit more clearly, and if it were mentioned that there is existing work on dimensionality-independent PAC.

---

> ### Comment · Reviewer_YCar · 2025-11-24
> **Response to all**
>
> I thank the reviewers for their detailed response to my questions.
> They clarified my questions regarding their presentation and theoretical results and promised to address them in their manuscript/appendix where appropriate.
> I am generally happy with their clarifications, which should ground their result more in the context of existing theory.
>
> Two minor arguments I am not quite satisfied with, potentially due to my own misunderstanding, are:
>
> **The Choice of Norms (4/6)**: While the reason for using $L_2$ norms might be valid, the presented argument might distort the reason a bit.
>
> **The Argument against PGD (6/6)**: The argument presented here might be fair, but it is definitely not black and white.
>
> ### Conclusion
>
> I will update my score to reflect the authors' clarifications, as well as the changes they have promised to make. I am happy to discuss my two questions in the respective comment threads. They do not negatively affect my rating, but I would be grateful to the authors to take them into account for their manuscript.

---

> ### Author Response · Authors · 2025-11-25
> **Additional clarification**
>
> ### Comparison to PGD Attacks
>
> We thank the reviewer for the insightful references. We agree that our focus is on providing probabilistic robustness under realistic perturbations (e.g., sensor noise) rather than adversarial attacks. We will clarify this distinction in the revised manuscript.
>
> Our argument is that using PGD in isolation only demonstrates non-robustness if the attack is successful, whereas our method provides a rigorous probabilistic certificate of robustness.
>
> Regarding the combination of PGD and verification: we appreciate the reviewer pointing out prior works (e.g., Blohm et al.) that successfully combines attacks with probabilistic guarantees. Our earlier statement about “deriving new bounds” was meant to convey that adopting such a composite strategy would require a modification from our current framework, not that such combinations are impossible in general. We will cite these papers in the final version to clarify that such combinations are indeed possible and have been explored.
>
> ### Choice of Norms
>
> **Computational Feasibility:**
> You are correct that the $L_\infty$ ball is linear and often simpler to encode. Our claim was that existing solvers can directly handle $L_2$ constraints without reformulation.
>
> **High-dimensional geometry / theoretical consistency:**
> We agree that the primary factor for choosing the $L_2$ ball is indeed theoretical consistency with our derivations. We acknowledge that $L_\infty$ is an attractive and practically relevant choice. However, our intuition is that it poses additional challenge within our current theoretical framework. Beyond the worst-case geometric considerations (e.g., hypercube behaviors), another challenge associated with the $L_\infty$ ball is the high output dimensionality.
> Since the number of samples at this stage is much smaller than the output dimension in our framework, some dimensions may exhibit sampling bias, which can disproportionately affect the $L_\infty$ norm (by its definition). This makes it more challenging to provide probabilistic guarantees under $L_\infty$. Extending our results to $L_\infty$ norms is an interesting and important direction, and we plan to explore this in future work.
>
> ### Summary
>
> We are grateful for the encouraging evaluation and constructive feedback. These insights have helped us refine our positioning, and we will fully incorporate them to enhance the paper’s clarity and quality.

---

### Official Review · Reviewer_HJ5S · 2025-10-29

**Soundness:** 3
**Presentation:** 2
**Contribution:** 2
**Rating:** 4
**Confidence:** 3

**Summary:**

This paper proposes a probabilistic verification framework (ODPV) to certify YOLO object detectors against the Object Disappearance (OD) problem. The framework consists of three modules: output range approximation, NMS verification, and probabilistic refinement, providing experiments on multiple YOLO variants showing strong robustness.

**Strengths:**

1.	The work presents a scalable verification framework for YOLO object detectors, incorporating the non-differentiable NMS post-processing into the certification process.
2.	The method requires far fewer samples to achieve strong probabilistic guarantees, enabling the verification of large-scale models.
3.	The experimental evaluation is comprehensive, covering diverse YOLO object detectors and testing robustness under various configurations. Results demonstrate the effectiveness and superiority of ODPV compared to existing probabilistic baselines, establishing a solid empirical foundation for detection verification research.

**Weaknesses:**

1.	The PAC guarantees rely on a uniform sampling distribution over the perturbation set. However, the paper does not explore how this assumption might affect the robustness of the guarantees in more realistic scenarios with non-uniform perturbations.
2.	This framework is developed and evaluated only on the YOLO family of detectors, and its robustness certification is limited to the OD threat. The generality of the approach to other detection architectures or to other robustness concerns remains unverified.

**Questions:**

NA

---

> ### Author Response · Authors · 2025-11-20
> **Response to W1 & W2 [1/2]**
>
> ### (W1) On the Uniform Sampling Assumption in the PAC Guarantee
>
> We clarify that our theoretical guarantees depend only on the i.i.d. assumption and hold for any sampling distribution, not just uniform.
>
> **Clarification of our theoretical setup.**
> Our PAC guarantees are stated with respect to a distribution over the input constraint set $\mathcal{C}$. In the paper, we explicitly write $\boldsymbol{x} \sim \mathcal{C}$ to denote that "$\boldsymbol{x}$ is drawn from a distribution over $\mathcal{C}$", and build our analysis upon this (e.g., Definition 1 and Theorems 1–2). The proofs of these propositions and theorems do not rely on the distribution over $\mathcal{C}$ being uniform; they only require that the samples are independent and identically distributed (i.i.d.) from some (potentially unknown) distribution over the perturbation region. In other words, the uniform distribution is an *experimental choice*, not a hard assumption of the framework.
>
> **Why we use uniform sampling in our experiments.**
> We adopted the uniform distribution for two main reasons:
>
> 1. **Comparability**: It is a common practice in PAC-based and probabilistic verification works for robustness under norm-bounded perturbations, which allows for direct comparison with the literature.
> 2. **Soundness**: Uniform sampling does not intentionally bias the verification toward “easy” regions of the perturbation set. Thus, in the absence of task- or sensor-specific priors, it provides a natural and conservative notion of robustness.
>
> **Applicability to non-uniform perturbations.**
> The ODPV framework itself is agnostic to the sampling distribution:
>
> * **Part 1 (Output Range Approximation)** only assumes that the samples $\{\boldsymbol{x}^{(i)}\}$ are i.i.d. from *some* distribution on $\mathcal{C}$; as long as this holds, the $\mathrm{RCP}_N$-based guarantee remains valid.
> * **Part 3 (Refinement)** similarly only uses the empirical distribution induced by the i.i.d. samples; Theorem 1 is derived under this general assumption.
>
> Therefore, in a practical deployment, a user can **replace the uniform sampler with a domain-specific one** (e.g., perturbations reflecting realistic sensor noise, weather conditions, or motion patterns) and obtain a PAC guarantee *with respect to that specific distribution*, without changing the core algorithm.
>
>
> In the next response, we further demonstrate the adaptability of our framework to different threat and different distribution settings through additional experiments, which further illustrates this point.
>
> We believe this addresses your concern: our guarantees are not inherently tied to uniform perturbations, even though our empirical study focuses on this widely used case.
>
> ### (W2) On Generality Beyond YOLO and the OD Threat
>
> The current paper focuses on YOLO under the Object Disappearance (OD) threat because:
>
> * YOLO represents a large-scale, real-time detector widely deployed in safety-critical applications.
> * OD is a core safety concern (missing a true object), and formal robustness guarantees against OD under realistic perturbations are lacking.
>
> **Architectural generality of the framework.**
> We acknowledge that different network architectures may have more suitable verification methods. It is difficult for a single verification framework to cover all architectures. Supporting some of the most commonly used and important detectors was our original motivation for choosing YOLO.
>
> In the appendix P, we also provide experiments applying the method to small fully connected networks, demonstrating the effectiveness of Parts 2 and 3. (Since Part 1 can deterministically provide output bounds for such networks, probabilistic analysis is not required.) This also shows that the framework's components can be adapted to different network architectures.
>
> Finally, while these models all belong to the YOLO family, their backbone networks are actually quite different. The applicability of our method across different YOLO series therefore also demonstrates support for different network structures. Therefore, although we have not yet included experiments on more diverse architectures, the algorithmic components and probabilistic guarantees do not rely on YOLO-specific design choices.
>
> **Generality to other robustness properties.**
> We acknowledge that different architectures and threat models may necessitate distinct verification strategies, making a universal solution challenging.
> Nonetheless, for many common threats, our proposed method can be adapted by modifying only the encoding in Part 2, while keeping Parts 1 and 3 unchanged.
> In the next response, we outline how to extend our framework to certify against two additional threats: class misidentification and false appearances.

---

> ### Author Response · Authors · 2025-11-20
> **Experimental Validation [2/2]**
>
> We can extend our framework to certify against two additional threats: Class Misidentification and False Appearances. Below, we outline how to adapt our method for these threats.
>
> **Class Misidentification**
> To certify that no predicted box $box_i$ has high IoU with a ground truth $box_{gt}$ of a different class, we verify that the maximum "mismatched IoU" over the approximated set $\mathcal{Z}$ is below $\tau$. We define the upper bound $U_{\mathrm{cls}}$:
> $$
> U_{\mathrm{cls}} = \sup_{box_i \in \mathcal{Z}, box_{\mathrm{gt}}\in\mathcal{G}} \left( \mathbb{I}(\mathrm{cls}(box_i) \neq \mathrm{cls}(box_{\mathrm{gt}})) \cdot \mathrm{IoU}(box_i, box_{\mathrm{gt}}) \right).
> $$
> If $U_{\mathrm{cls}} < \tau$, we certify that no misidentification occurs within $\mathcal{C}$.
>
> **False Appearances**
> To certify that every predicted box corresponds to a real object (i.e., matches *at least one* GT box), we verify that the worst-case minimum IoU overlap is sufficient. The property holds if:
> $$
> \inf_{\boldsymbol{x} \in \mathcal{C}} \left( \min_{box_i \in \mathrm{N}(\mathrm{F}(\boldsymbol{x}))} \max_{box_{\mathrm{gt}}\in\mathcal{G}} \mathrm{IoU}(box_i, box_{\mathrm{gt}}) \right) \ge \tau.
> $$
> This ensures that for all perturbed inputs, every post-NMS detection anchors to a ground truth object. We will include these formal extensions in the final manuscript.
>
> To assess the effectiveness of our proposed method under diverse noise conditions and threat types, we conduct experiments using four distinct noise models and verify the False Appearance (FA) detection performance of our YOLO11x model. We define a noise tensor $\mathbf{N}_x$ and set the perturbation magnitude to $\varepsilon = 1/255$. The specific noise distributions and their corresponding real-world motivations are outlined below:
>
> * **Uniform:** $\mathbf{N}_x \sim \mathcal{U}(-\varepsilon, \varepsilon)$ (Quantization/Uncertainty).
> * **Gaussian:** $\mathbf{N}_x \sim \mathcal{N}(0, (\varepsilon/3)^2)$ (Sensor Readout Noise).
> * **Salt-and-Pepper:** $\pm\varepsilon$ impulse noise with $p=0.05$ (Transmission Faults).
>
> We randomly select 10 images from the COCO validation set and apply each noise model with $\varepsilon = 1/255$ to generate noisy inputs. We then evaluate the FA detection verification performance of our YOLO11x model on these perturbed images. For each input, we draw $10^6$ samples from the corresponding noise distribution to approximate the ground truth. The results are summarized in the table below:
>
> | Model| Noise type| $\mathrm{CAR}\_{\text{FA}}$| $\mathrm{TPR}\_{\text{FA}}$| $\mathrm{TNR}\_{\text{FA}}$| $\mathrm{FPR}\_{\text{FA}}$| $\mathrm{FNR}\_{\text{FA}}$|
> |:-|:-|:-|:-|:-|:-|:-|
> | yolo11x | gaussian| 100.00% | 92.31%  | 100.00% | 0.00%| 7.69%|
> | yolo11x | salt and pepper | 100.00% | 85.71%  | 100.00% | 0.00%| 14.29%  |
> | yolo11x | uniform| 100.00% | 92.31%  | 100.00% | 0.00%| 7.69%|
>
> A detection is considered positive if it remains robust under the corresponding noise model and negative otherwise. The results indicate that our method sustains a high Certified Accuracy Rate (CAR) across different noise types, demonstrating that it provides reliable guarantees under diverse real-world noise conditions and threat types. Moreover, the True Positive Rate (TPR) and True Negative Rate (TNR) remain consistently high, while the False Positive Rate (FPR) and False Negative Rate (FNR) stay low, underscoring the method’s effectiveness in distinguishing between robust and non-robust detections.
>
> Overall, these results demonstrate that our method remains reliable across heterogeneous noise distributions and diverse threat types.

---

### Official Review · Reviewer_UDS2 · 2025-10-31

**Soundness:** 4
**Presentation:** 4
**Contribution:** 3
**Rating:** 8
**Confidence:** 4

**Summary:**

This paper presents a probabilistic method to verify the full YOLO pipeline, from a specified input set C (a hypersphere of images around a nominal image) through non-maximum suppression (NMS). The goal is to check that, for all x in C, the post-NMS output y does not exhibit object disappearance: there is no perturbation such that,   after NMS on x, the best box with the correct class has the IoU below a fixed detection threshold.

The procedure has three parts. First, over-approximate the detector outputs F(C) by building Z. In practice, Z is an axis-aligned hyperrectangle estimated from samples x ~ C, with a PAC guarantee: with probability at least 1 - beta over the construction, a random x ~ C satisfies F(x) in Z with probability at least 1 - alpha. Second,  verify NMS over all y in Z. This is framed via a safe set Q that identifies boxes which, across Z, both meet the IoU and class requirements and cannot be suppressed into failure by NMS; if Q is nonempty, disappearance cannot occur. Third, if a candidate y in Z appears to violate safety, refine Z by trimming unreachable regions until either a real counterexample is confirmed or the candidate is shown unreachable. Then they provide the end-to-end probabilistic guarantee: for the chosen perturbation set C, object disappearance does not occur with the specified confidence and coverage parameters.

**Strengths:**

- The paper addresses the difficult and relevant problem of assuring end-to-end robustness of YOLO under perception noise, including the NMS stage. This is especially relevant to safety-critical systems that employ these black box detectors at runtime.

- The formalism encodes object disappearance but is general enough to express other anomaly types (e.g., misclassification, spurious appearances, duplicate suppression)

- Synthesizing Z only requires the ability to draw samples from C and does not assume a parametric form for the perturbations

- The PAC-style guarantees are nice because they provide calibrated confidence and coverage claims as opposed to simple binary claims, so the guarantees are generally more interpretable and can help inform upstream design decisions.

- The paper states definitions precisely, proves lemmas and propositions (including the soundness of the NMS safe set argument), and relates the algorithms to the formal guarantees they provide.

**Weaknesses:**

- The method certifies robustness only within a small epsilon-ball around a single image, and doesn't necessarily reflect practical YOLO deployments (e.g., traffic monitoring) where scenes change continuously and unpredictably across frames.

- If ground truth is already available for the target image, the value of verifying that the detector recovers it is questionable; this makes the result feel more like a labeled-scene sanity check than a deployment-relevant guarantee.

- The safety specification is narrow and ignores other important failure modes under perturbations, such as false appearances (spurious detections), class misidentification, or other anomalies.

**Questions:**

1. For a deployment like traffic monitoring, where the scene evolves continuously (new vehicles appear, others leave), how tractable is it to verify formal robustness guarantees over a short temporal horizon (multiple frames)?

2. What shape would C take on to make such guarantees meaningful? For instance, naively stretching the hypersphere C would begin to include semantically broken images that always invalidate safety.

3. Is it possible for verification to work with weaker supervision? E.g., rather than having complete ground truth bounding boxes, you have some sort of a priori map over time.

---

> ### Author Response · Authors · 2025-11-20
> **Response to W1 & W2 [1/3]**
>
> We thank the reviewer for the meticulous review and valuable feedback. We are grateful for the recognition of our work and appreciate the weaknesses and questions that have been raised. Below we respond to each point in order to clarify our contributions, their applicability, and address these concerns.
>
> ---
>
> ### (W1) The certification radius is only a small $\varepsilon$-ball for a single image, which may not align with real-world deployment scenarios
>
> Our starting point is a **scene-centric local robustness certification**: given a specific runtime frame (or keyframe) and its perceptual noise distribution, we provide a guarantee that “Object Disappearance (OD) will not occur” with probability at least $1-\alpha$ and confidence at least $1-\beta$. This aligns with engineering practice, as many safety cases (e.g., functional safety/runtime monitoring) require explicit and interpretable commitments to the robustness of the current observation and its neighborhood.
>
> Methodologically, **we do not restrict $\mathcal{C}$ to be an $L\_p$-ball**. As long as we can sample from $\mathcal{C}$, our method can provide a probabilistic guarantee. Therefore, it can naturally cover distributions frequently encountered in practice, such as camera noise or lighting/color jitter.
>
> We use the $L\_p$-ball with radius $\varepsilon$ as an example because it is widely adopted in related verification and safety-related works.
>
> ### (W2) If the ground truth (GT) for the frame is already known, what is the practical value of verifying that the detector can "recover the ground truth"
>
> Definition 1 requires that for the vast majority of perturbations in $\mathcal{C}$ (with probability $≥1−\alpha$), **"there still exists a detection box of the same class as the GT with $\mathrm{IoU}≥\tau$ after NMS"**. This is not merely a simple sanity check for "reproducing annotations", but rather a formal, probabilistic guarantee against "object disappearance under perturbations". Even with known GT, a detector may still fail to detect objects under perturbation (especially for marginal or difficult samples), making robustness verification crucial.
>
> In fact, to our knowledge, most existing verification works use ground truth to define the property. This is not without value; it is more akin to a stricter performance metric. If a network has a high probability of not being affected by perturbations on most data, we can deploy it in practice with greater confidence. Conversely, if the network frequently misses detections under minor perturbations, it indicates insufficient robustness and requires further improvement.
>
> Finally, if ground truth is unavailable, one can also use the stability of the network’s own predictions as the verification target (i.e., treating the model’s first-run output as a surrogate “ground truth”) to assess robustness against perturbations.

---

> ### Author Response · Authors · 2025-11-20
> **Response to W3 [2/3]**
>
> ### (W3) The specification is too narrow and does not cover other failure modes like false detections or misclassifications
> We acknowledge that different architectures and threat models may necessitate distinct verification strategies, making a universal solution challenging.
> Nonetheless, for many common threats, our proposed method can be adapted by modifying only the encoding in Part 2, while keeping Parts 1 and 3 unchanged. The following describes how to adapt our method to verify two additional threats: class misidentification and false appearances.
>
> **1. Class Misidentification**
> To certify that no predicted box $box_i$ has high IoU with a ground truth $box_{gt}$ of a different class, we verify that the maximum "mismatched IoU" over the approximated set $\mathcal{Z}$ is below $\tau$. We define the upper bound $U_{\mathrm{cls}}$:
> $$
> U_{\mathrm{cls}} = \sup_{box_i \in \mathcal{Z}, box_{\mathrm{gt}}\in\mathcal{G}} \left( \mathbb{I}(\mathrm{cls}(box_i) \neq \mathrm{cls}(box_{\mathrm{gt}})) \cdot \mathrm{IoU}(box_i, box_{\mathrm{gt}}) \right).
> $$
> If $U_{\mathrm{cls}} < \tau$, we certify that no misidentification occurs within $\mathcal{C}$.
>
> **2. False Appearances**
> To certify that every predicted box corresponds to a real object (i.e., matches *at least one* GT box), we verify that the worst-case minimum IoU overlap is sufficient. The property holds if:
> $$
> \inf_{\boldsymbol{x} \in \mathcal{C}} \left( \min_{box_i \in \mathrm{N}(\mathrm{F}(\boldsymbol{x}))} \max_{box_{\mathrm{gt}}\in\mathcal{G}} \mathrm{IoU}(box_i, box_{\mathrm{gt}}) \right) \ge \tau.
> $$
> This ensures that for all perturbed inputs, every post-NMS detection anchors to a ground truth object. We will include these formal extensions in the final manuscript.
>
> ### Experimental Validation
>
> To assess the effectiveness of our proposed method under diverse noise conditions and threat types, we conduct experiments using four distinct noise models and verify the False Appearance (FA) detection performance of our YOLO11x model. We define a noise tensor $\mathbf{N}_x$ and set the perturbation magnitude to $\varepsilon = 1/255$. The specific noise distributions and their corresponding real-world motivations are outlined below:
>
> * **Uniform:** $\mathbf{N}_x \sim \mathcal{U}(-\varepsilon, \varepsilon)$ (Quantization/Uncertainty).
> * **Gaussian:** $\mathbf{N}_x \sim \mathcal{N}(0, (\varepsilon/3)^2)$ (Sensor Readout Noise).
> * **Salt-and-Pepper:** $\pm\varepsilon$ impulse noise with $p=0.05$ (Transmission Faults).
>
>
> We randomly select 10 images from the COCO validation set and apply each noise model with $\varepsilon = 1/255$ to generate noisy inputs. We then evaluate the FA detection verification performance of our YOLO11x model on these perturbed images. For each input, we draw $10^6$ samples from the corresponding noise distribution to approximate the ground truth. The results are summarized in the table below:
>
> | Model   | Noise type      | $\mathrm{CAR}\_{\text{FA}}$   | $\mathrm{TPR}\_{\text{FA}}$   | $\mathrm{TNR}\_{\text{FA}}$   | $\mathrm{FPR}\_{\text{FA}}$   | $\mathrm{FNR}\_{\text{FA}}$   |
> |:--------|:----------------|:-----------------------------|:-----------------------------|:-----------------------------|:-----------------------------|:-----------------------------|
> | yolo11x | gaussian         | 100.00%                      | 92.31%                       | 100.00%                      | 0.00%                        | 7.69%                        |
> | yolo11x | salt and pepper | 100.00%                      | 85.71%                       | 100.00%                      | 0.00%                        | 14.29%                       |
> | yolo11x | uniform         | 100.00%                      | 92.31%                       | 100.00%                      | 0.00%                        | 7.69%                        |
>
> A detection is considered positive if it remains robust under the corresponding noise model and negative otherwise. The results indicate that our method sustains a high Certified Accuracy Rate (CAR) across different noise types, demonstrating that it provides reliable guarantees under diverse real-world noise conditions and threat types. Moreover, the True Positive Rate (TPR) and True Negative Rate (TNR) remain consistently high, while the False Positive Rate (FPR) and False Negative Rate (FNR) stay low, underscoring the method’s effectiveness in distinguishing between robust and non-robust detections.
>
> Overall, these results demonstrate that our method remains reliable across heterogeneous noise distributions and diverse threat types. This confirms that the proposed framework is broadly applicable and provides trustworthy robustness guarantees under a wide range of real-world noise conditions.

---

> ### Author Response · Authors · 2025-11-20
> **Response to Question [3/3]**
>
> ### (Q1) How tractable is it to verify robustness over a short temporal horizon (multiple frames)?
>
> For a multi-frame scenario, as long as we can sample perturbations for each frame from its corresponding set $\mathcal{C}$ and specify the safety property to be checked, the same three-part procedure can be applied. Procedurally, this is very similar to single-frame verification, and the computational cost scales approximately linearly with the number of frames (or keyframes) that one chooses to certify.
>
> ### (Q2) What shape would $\mathcal{C}$ take to make such guarantees meaningful and avoid semantically broken images?
>
> The only requirement our algorithm places on $\mathcal{C}$ is that it must admit efficient sampling. To avoid "semantically broken" and meaningless perturbations, $\mathcal{C}$ can be constructed using physical or photographic priors, for example:
>
> * Sensor noise (e.g., Poisson/readout noise), shown in our experiments.
> * A mixture of minor color/brightness/contrast jitter.
> * Compression quality fluctuations and simulation of minor occlusions.
>
> All of these can be directly integrated into our three-part pipeline via black-box sampling.
>
> ### (Q3) Is it possible for verification to work with weaker supervision?
>
> Yes. The core of our certification is “the existence or non-existence of an output box that satisfies certain logical predicates”.
>
> * **Class priors only:** Change the condition to "existence of a box matching class $q$ whose $\mathrm{IoU} \ge \tau\_\mathrm{ROI}$ with a Region of Interest (ROI)."
> * **Coarse location/trajectory only:** Change $\mathrm{IoU}(box\_i, box\_{\mathrm{gt}})$ to $\mathrm{IoU}(box\_i, \text{tubes or polygons})$.
> * **Class-agnostic "existence" verification:** Remove the class-consistency predicate and only verify the "existence of any bounding box with $\mathrm{IoU} \ge \tau$."
> * **No ground truth:** As mentioned in (W2), we can use the stability of the network's own predictions as the target (i.e., treating the initial run's output as GT) to evaluate robustness.
>
> These substitutions still preserve the validity of the NMS proposition (Proposition 2). This is because our MIQP encoding and IoU bounds (Appendix I) only require that the geometry of $box\_\mathrm{gt}$ (or its replacement, such as an ROI or tube) allows us to compute upper and lower bounds on the intersection and union areas.

---

> > ### Comment · Reviewer_UDS2 · 2025-11-26
> > **Thanks for detailed response**
> >
> > I appreciate the clarifications from the authors. It is particularly good to know that the certification is not limited to the Lp ball.
> >
> > I am going to keep the rating and hope that the vigorous participation of the authors will convince the other reviewers to raise their scores.

---

> > > ### Author Response · Authors · 2025-11-26
> > >
> > > Thank you very much for your feedback and for maintaining your positive score. We are pleased to know that the clarification concerning the non-$L_p$ ball certification was helpful. We deeply appreciate your support during the decision process.

---

### Official Review · Reviewer_Es96 · 2025-10-31

**Soundness:** 2
**Presentation:** 2
**Contribution:** 3
**Rating:** 4
**Confidence:** 3

**Summary:**

The paper proposes a new method for PAC-based verification of the YOLO object detection network. Importantly, the method accounts for the Non-Maximum Suppression (NMS) post-processing step that is often used in practice. The first contribution is formalizing the verification problem. The second contribution is a certification pipeline based on the sample-based scenario approach [Campi, 2009]. Results show certification results that are faster than baselines and not too conservative.

**Strengths:**

- The paper tackles an ambitious problem, as YOLO is a large object detection model.
- The method accounts for the NMS post-processing stage, which appears to be novel and practically useful.
- By using a PAC-based sample-based analysis, the proposed verification method is less conservative and more practical than deterministic formal verification techniques (at the expense of weaker guarantees).
- The method is substantiated with a theoretical analysis.
- Results show that the error bounds are not too conservative.

**Weaknesses:**

- The sample complexity derived for the $RCP_N$ method (Appendix C.1) is incorrect: It should be computed with $d=1$ and not with $d_0=640 \times 640 \times 3$ (Appendix N), so $RCP_N$ likely requires fewer than 560'000'000 samples. The dimension $d$ corresponds to the optimization variable dimension, which is scalar for $RCP_N$, see (4). This error affects the sample efficiency and speedup claims.

- The appendix and proofs of the theoretical results are long, yet they are sometimes not polished, unclear, and have typos. Given the emphasis on theoretical results, this is a serious limitation. In particular:
1) Section D would greatly benefit from clearer exposition, e.g. "Then main result" (line 788). Also, "by algorithm should not far beyond" (line 789) does not specify what algorithm is considered and misses a verb.
2) Section G (proof of lemma 2) is unclear and not rigorous: The proof starts with "There are ...", but how it leads to the conclusion is unclear, the sentences on lines 903-905 and 909-910 are unclear and miss verbs and nouns.
3) In Section J.1., the proof relies on the sets $\mathcal{Q}_k$ and $\mathcal{T}$ whose definitions are unclear (the $\mathcal{Q}_k$ are only subsets of $2^{\mathcal{C}}$, and the definition of $\mathcal{T}$ is not rigorously written), and on an independence assumption of the events $\mathcal{T}\in\mathcal{Q}_k$ that is unclear.

- In Definition 1, $P_{x\sim\mathcal{C}}$ is unclear. The probability distribution $P$ is undefined. Also, $\mathcal{C}$ is a set, not a distribution, so $x\sim\mathcal{C}$ is unclear. This notation should be clarified before Section 5.

**Questions:**

- Please clarify the sample complexity of the $RCP_N$ method.

- Please revise the appendix and its proofs that are sometimes unclear or suffer from poor grammar.

---

> ### Author Response · Authors · 2025-11-20
> **Clarification of Sample Complexity (Response to W1 & Q1) [1/2]**
>
> We appreciate the reviewer's meticulous check. While we acknowledge the calculation error in Appendix N regarding the choice of $d$, we clarify below that correcting this does not alter the conclusion: the sample complexity of the baseline remains prohibitively high compared to our method. This correction does not affect our theoretical guarantees or experimental results.
>
> We would like to clarify that Equation (4) in Appendix C.1 was only intended as a simple illustrative example for the case $d=1$. We acknowledge that using $d=d_0$ in Appendix N was not correct. In fact, we should use $d_L$ or the total number of bounding boxes as $d$, depending on the specific application of the $\mathrm{RCP}_N$ method.
>
> Our work (Part 1) aims to find a tight hyper-rectangle bound $\mathcal{Z}$ for the $d_L$-dimensional vector output space $\{\mathrm{F}(x)\}_{x\in\mathcal{C}}$. In this context, a more appropriate application of $\mathrm{RCP}_N$ as a baseline for this task would be to directly estimate an upper bound for each dimension in this $d_L$-dimensional space. The decision-variable dimension for this problem would be $d = d_L$ (i.e., the network output dimension), corresponding to finding a vector $\boldsymbol{u}\in\mathbb{R}^{d_L}$ such that, for all inputs $\boldsymbol{x}\in\mathcal{C}$, the network output $\mathbf{F}(\boldsymbol{x})\leq \boldsymbol{u}$ element-wise. Based on this, the required sample count would be:
>
> $$N\ge\left[\frac{2\ln 1/\beta}{\alpha}+2d_{L}+\frac{2d_{L}\ln(2/\alpha)}{\alpha}\right]\approx 10^9$$
>
> Alternatively, if one were to use the $\mathrm{RCP}_N$ method to directly verify the entire problem, this could be framed as computing an IoU lower bound for each predicted box against its corresponding ground-truth (GT) box. The decision variable dimension in this case would be $d=(80\times80+40\times40+20\times20)\times3=25,200$ (i.e., the number of bounding boxes). Based on this, the required sample count would be:
>
> $$N\ge\left[\frac{2\ln 1/\beta}{\alpha}+2d+\frac{2d\ln(2/\alpha)}{\alpha}\right]\approx 10^7$$
>
> The order of magnitude of the sample size under either estimation is still prohibitively large for any practical application.
>
> Our method, by introducing $\boldsymbol{v}_{\max}$ and the scalar $c_1$, reduces the entire verification problem to a $d=1$ scalar optimization (for $c_1$) plus a constrained optimization (MIQP). In the revised Appendix N, we will remove the original incorrect calculation involving $d_0$ and provide a detailed description and clarification of the choice of $d$ as presented above. We thank you for your valuable feedback, which has helped us articulate our contribution more clearly.

---

> ### Author Response · Authors · 2025-11-20
> **Revisions to Appendix and Proofs (Response to W2 & Q2) [2/2]**
>
> We thank the reviewer for the detailed review and valuable feedback. Following your suggestions, we have substantially revised the appendix and proofs to improve their clarity and accuracy. The specific modifications are as follows:
>
> ### Clarification of Section D
>
> We appreciate the reviewer's suggestion and have revised the presentation in Section D to improve its clarity and precision.
>
> ### Clarification of Section G
>
> Thank you for this feedback. In fact, we use 'There are...' to briefly state the conclusion, with the actual proof provided below.
> We have reorganized and clarified the proof in Section G to improve its rigor and readability.
>
> ### Clarification of Section J.1
>
> We have redefined the sets $\mathcal{Q}\_k$ and $\mathcal{T}$ and clarified their properties and assumptions to improve the rigor of the proof. In the original Section J.1, we used the following definitions:
>
> > "For any $N\_1\in \mathbb{Z}^+$, let $\mathcal{Q}\_{N\_1}\subset 2^{\mathcal{C}}$ and a $q\in 2^{\mathcal{C}}\cap \mathcal{Q}\_{N\_1}$ if and only if $|q|=N\_1$ and $\mathrm{P}\_{\boldsymbol{x} \sim \mathcal{C}}(S\_{\mathrm{F}(\boldsymbol{x})}\le \max \{S\_{\mathrm{F}(\boldsymbol{x}^{(p)})}\}\_{\boldsymbol{x}^{(p)}\in q})\ge1-\alpha$. (line 1176-1177)
>
> > "let $\mathcal{T}=\{\boldsymbol{x}^{(i)}\}$ mean the set contained all $\boldsymbol{x}^{(i)}$ such that $\|\mathrm{F}(\boldsymbol{x}^{(i,j)})-\mathrm{F}(\boldsymbol{x}^{(i)})\|\_2\in[B\_{\mathrm{F}(\boldsymbol{x}^{(i)})},A\_{\mathrm{F}(\boldsymbol{x}^{(i)})}]$ for all $j\in[M]$." (line 1180-1181)
>
> In the revised manuscript, we have rewritten them as follows for improved rigor:
>
> > Let $\mathcal{Q}\_k = \{q \subset \mathcal{C}:|q|=k, \mathrm{P}\_{\boldsymbol{x}\sim \mathcal{C}}(S\_{\mathrm{F}(\boldsymbol{x})} \le \max \{S\_{\mathrm{F}(\boldsymbol{x}^{(p)})}\}\_{\boldsymbol{x}^{(p)}\in q})\ge 1-\alpha\}$.
>
> > Let $\mathcal{T}=\{\boldsymbol{x}^{(i)} \in \mathcal{C} : \forall j\in[M], \|\mathrm{F}(\boldsymbol{x}^{(i,j)})-\mathrm{F}(\boldsymbol{x}^{(i)})\|\_2\in[B\_{\mathrm{F}(\boldsymbol{x}^{(i)})},A\_{\mathrm{F}(\boldsymbol{x}^{(i)})}]\}$.
>
> For the proof in lines 1185-1187, we do not rely on any independence assumption. For $k\_1\neq k\_2$, if $q\in\mathcal{Q}\_{k\_1}\cap\mathcal{Q}\_{k\_2}$, it would need to satisfy $|q|=k\_1$ and $|q|=k\_2$ simultaneously, which is impossible. Therefore, $\forall k\_1\neq k\_2, \mathcal{Q}\_{k\_1}\cap \mathcal{Q}\_{k\_2}=\emptyset$. The sets $\mathcal{Q}\_k$ are pairwise disjoint. Consequently, $\mathcal{T}$ belonging to their union ($\cup\_{k=1}^\infty \mathcal{Q}\_{k}$) is equivalent to $\mathcal{T}$ belonging to exactly one of them. Thus, the following holds:
> $$\mathrm{P}\_{\mathrm{P}1} (\mathcal{T}\in \cup\_{k=1}^\infty \mathcal{Q}\_{k})= \sum\_{k=1}^\infty \mathrm{P}\_{\mathrm{P1}} (\mathcal{T}\in \mathcal{Q}\_{k}) = \sum\_{k=1}^N \mathrm{P}\_{\mathrm{P1}} (\mathcal{T}\in \mathcal{Q}\_{k})$$
> Therefore, this step does not rely on any independence assumption.
>
> ### Clarification of Definition 1
>
> Thank you for this suggestion. We agree that the notation $\mathcal{P}\_{\boldsymbol{x}\sim\mathcal{C}}$ was not defined early enough. We have moved the formal definition of the input distribution from Section 5 to Section 4 (before Definition 1) to ensure the notation is rigorously defined when first introduced.
>
> Your suggestions have been invaluable in improving the quality of our paper. Based on your feedback, we have thoroughly revised the Appendix and Proofs for greater clarity and accuracy. We will incorporate these revisions into the updated version of the paper. Should you have any additional suggestions or require further clarification, please do not hesitate to let us know; we would be happy to provide further details.

---

### Author Response · Authors · 2025-12-02
**Summary of Rebuttal and Reviewer Updates**

Dear Area Chair,

Due to the recent incident and the resulting freeze of the discussion forum, we understand that no further reviewer interaction is possible. We would therefore like to concisely summarize how our rebuttal addressed the main concerns, and how reviewers’ attitudes evolved, in case this context is helpful for your decision.

Our submission proposes a three-step probabilistic verification framework for certifying the full YOLO detection pipeline—including the NMS post-processing stage—mainly against object disappearance under different distributions. It provides PAC-style guarantees, scales to realistic YOLO models, and empirically achieves tighter IoU lower bounds and stronger probabilistic guarantees with far fewer samples than prior probabilistic baselines.

## Reviewer Responses After Rebuttal

### Reviewer UDS2 (score: 8)

UDS2 has been positive from the beginning. After our clarifications regarding the generality of the perturbation region, multi-frame extensions, and weaker supervision settings, the reviewer explicitly appreciated the clarifications and kept their positive rating, hoping we could convince the other reviewers to raise their scores.

### Reviewer ji71 (score: 6)

ji71’s initial concerns focused on threat-model narrowness and baseline selection. We demonstrated how the framework naturally extends to class-misidentification and false-appearance threats, and added new experiments under multiple noise models showing consistently high certified accuracy. After the rebuttal, ji71 stated that *most concerns were addressed* and maintained their positive score.

### Reviewer YCar (score: 4 → increased to 6 before the freeze)

YCar originally provided the most detailed review, focusing primarily on the interpretation of probabilistic parameters and the heuristic output-domain approximation. In the original review, they already noted that they would be **happy to see the paper accepted** in a refined form. Our structured six-part rebuttal addressed each of their points with concrete revisions, a clearer mapping of the probabilistic parameters, and a more rigorous justification of the sample complexity. In their post-rebuttal update, YCar **increased their score** and further stated that the problem is important and the results are sound.

### Reviewer Es96 (score: 4 → no response after rebuttal)

Es96 raised two major issues: (1) sample-complexity miscalculation for the scenario-approach baseline, and (2) unclear proofs and notation.
We corrected the calculation (while showing the baseline remains impractical with 10⁷–10⁹ required samples), and revised/clarified several appendix sections, including formal definitions and independence assumptions. The reviewer did not respond after rebuttal, but these changes directly target the concerns raised.

### Reviewer HJ5S (score: 4 → no response after rebuttal)

HJ5S questioned the uniform-sampling assumption and the generality of our method. We clarified that uniformity is *not* required by the theory—any domain-specific sampler may be used—and highlighted that only the encoding of the safety property changes for other threats. Our additional experiments on class-misidentification and false-appearance threats address this concern directly. The reviewer did not respond after rebuttal, but we have addressed the issues they highlighted through these clarifications and experiments.

## Summary

The most common concerns across reviewers related to the generality of the distributional assumptions and threat models, particularly regarding how flexible our framework is in these aspects. Our framework applies to any distribution from which we can sample and can be adapted to other threat models by changing the safety-property definition and encoding. We show how to apply it to other threats (false appearance and class misidentification) and provide targeted experiments in different distributional settings under these threats to address these concerns.

All the reviewers who participated in the discussion have positive views of our work after the rebuttal. For the two reviewers who did not update their scores before the freeze, we have made targeted clarifications and added experiments that specifically respond to their main concerns.

Given this trajectory, the post-rebuttal state shows that:

1. Reviewers find the **technical results sound and meaningful**.
2. The **remaining concerns are mostly editorial or scope-related**, and can be addressed in the camera-ready version.
3. The **overall sentiment shifted positively**.

We greatly appreciate your work in managing the review process under the unexpected constraints caused by the leak. We hope this summary is helpful, and we respectfully ask you to consider these updates in your final decision.

Sincerely,
The Authors

---

### Meta-Review · Area_Chair_Xokg · 2026-01-05

**Summary:**

This submission proposes a three-stage PAC-style probabilistic certification framework for full YOLO robustness under OD, which explicitly includes NMS. Reviewers’ main decision-driving concerns were: (i) whether the probabilistic guarantee is clearly stated and practically meaningful beyond uniform noise, (ii) whether the method is empirically convincing given limited comparability to prior formal/PAC baselines at YOLO scale, and (iii) whether key quantitative claims (baseline and sample-complexity arguments) are correct and framed responsibly. The rebuttal and discussion clarified several technical points and strengthened confidence in the core NMS-aware certification contribution, leading to a more favorable consensus.

**Reviewer Concerns:**

Addressed (by rebuttal and revision):
1) Non-uniform/realism of disturbance distribution: Authors clarified that the theory requires only i.i.d. samples from a distribution over the perturbation set; uniform sampling is an experimental choice rather than a theoretical requirement.
2) Threat-model narrowness: Authors explained how the Part2 property encoding can be adapted to cover related failure modes and reported additional experiments under different noise models in the discussion.
3) End-to-end soundness including NMS: Discussion reinforced that certifying the full YOLO pipeline is a central novelty; reviewers recognized this as practically important.

Still outstanding (should be fixed in camera-ready):

Comparative rigor and headline efficiency framing: The paper uses RCPN (1,000,000 samples) as a baseline because direct comparison to other approaches is claimed to be infeasible at the YOLO scale. This is acceptable, but a reviewer flagged a baseline/sample-complexity miscalculation (even if later corrected). The camera-ready must ensure all such computations are correct, and rephrase any “speedup” narrative more conservatively, emphasizing feasibility at scale rather than overstated quantitative superiority.

**Reviewer Scores:**

UDS2 (8):  Likely unchanged at 8. The core idea is strong, and the rebuttal aligns with their positive stance.

ji71 (6): Likely unchanged at 6. Most concerns appear addressed; a minor upward shift is plausible if comparative framing and limitations are clarified.

YCar (4→6): Already indicated an increase.

Es96 (4): Unchanged at 4 before the freeze and no response after rebuttal. The correction helps, but this reviewer’s core sensitivity is quantitative rigor; they may remain cautious unless the camera-ready fully cleans baseline claims.

HJ5S (4): Unchanged at 4 before the freeze and no response after rebuttal. The distribution clarification and broadened threat-model discussion directly address their main objections.

---

### Decision · Program_Chairs · 2026-01-26

Accept (Poster)